# Predictive models for the selection of thermally tolerant corals based on offspring survival

K. M. Quigley [1✉] & M. J. H. van Oppen [1,2]

Finding coral reefs resilient to climate warming is challenging given the large spatial scale of reef ecosystems. Methods are needed to predict the location of corals with heritable tolerance to high temperatures. Here, we combine Great Barrier Reef-scale remote sensing with breeding experiments that estimate larval and juvenile coral survival under exposure to high temperatures. Using reproductive corals collected from the northern and central Great Barrier Reef, we develop forecasting models to locate reefs harbouring corals capable of producing offspring with increased heat tolerance of an additional 3.4° heating weeks (~3 °C). Our findings predict hundreds of reefs (~7.5%) may be home to corals that have high and heritable heat-tolerance in habitats with high daily and annual temperature ranges and historically variable heat stress. The locations identified represent targets for protection and consideration as a source of corals for use in restoration of degraded reefs given their potential to resist climate change impacts and repopulate reefs with tolerant offspring.

[1] Australian Institute of Marine Science, Townsville, QLD, Australia. [2] School of BioSciences, The University of Melbourne, Parkville, VIC, Australia. ✉email: katemarie.quigley@my.jcu.edu.au

Human-driven climate warming is pushing coral reefs globally to the brink of extinction, predominantly due to the impacts of summer heat waves that are increasing in frequency and magnitude. Like other ectotherms[1], many of the coral species for which physiological tolerance is known have narrow thermal ranges. When temperatures surpass the coral's upper or lower thermal tolerance limits, the endosymbiotic relationship between the coral host and its photosynthetic dinoflagellates (family Symbiodiniaceae) breaks down and a large percentage of Symbiodiniaceae cells are lost from the host tissues, a process known as bleaching[2]. Bleaching deprives the coral of access to symbiont photosynthate, its essential carbon resource, and leads to coral starvation and potentially death[3]. Models predicting future sea surface temperatures forecast that warming of the tropical oceans will continue[4].

The occurrence and severity of mass coral bleaching are strongly linked to the magnitude and duration of summer temperature anomalies, although smaller scale bleaching in winter months may also occur[5]. Past exposure of corals to high temperatures over short (single generation, acclimation) and longer ecological timescales (multiple generations, adaptation) shapes corals' bleaching responses and survival during thermal anomalies. Although it is assumed that low latitude corals are more heat-tolerant, resilient reefs of tolerance are increasingly being found in other locations. Signatures of acclimation and adaptation have been detected globally, including in Singapore[6], the Great Barrier Reef (GBR)[7], and the Persian Gulf[8]. In some of these locations, acquired increases in heat tolerance have since been shown across diverse species (e.g. *Coelastrea aspera*[9]) and regions (e.g. the Persian Gulf[10]). A range of dependent explanatory variables, including warming intensity, frequency of anomalies, and variability in daily, monthly, and yearly temperature profiles on reefs have been used to explain patterns of bleaching in adult corals[11]. These metrics provide some predictive power in explaining why bleaching has occurred in particular locations[12] or in predicting where and when bleaching will or will not occur in the future[13].

Heritable, genetic variation underpins adaptive potential[14]. Genetic variation of the coral host has been linked to tolerance to higher temperatures that persists from the adult into the larval[15] and juvenile[16] phase, although this notion is based on only a small subset of coral species and findings from these studies may not be generalizable across species and genera. Finding adult corals with heat tolerance alleles is challenging given the large spatial scales of coral reef ecosystems. Moreover, confounding effects of the environment (i.e. genetic environmental covariance, $V_E$)[17] or corals' Symbiodiniaceae symbionts can result in different phenotypes at different ontogenic stages, meaning that adult responses to heat are not always indicative of offspring performance[16]. No method exists that incorporates the heritability of heat tolerance to predict where heat-resistant adult brood stock is located.

In this work, we address this issue by developing a forecasting framework ('intrinsic resistance' models) that predicts conditions for the occurrence of adult colonies that exhibit a high tolerance to heat and a high heritability of heat tolerance. Machine learning gradient boosted models combined with remotely-sensed environmental parameters identifies locations on the GBR predicted to be home to such corals and that represent important targets for conservation planning.

## Results and discussion

**Quantitative genetic breeding experiments**. Data on the survival and heritability of heat tolerance were collected from controlled quantitative genetic experiments conducted in the National Sea Simulator at the Australian Institute of Marine Science for the construction of intrinsic resistance models. Gravid *Acropora tenuis*

colonies were sourced from three (CU-Curd, LS-Long Sandy, SB-Sand Bank 7) and two reefs (BK-Backnumbers, DR-Davies) in the far northern and central GBR, respectively, encompassing a 23–33 °C of the 1.5× interquartile range of the mean annual temperature gradient (>6° of latitude, ~900 km, Fig. 1a). Purebred and hybrid offspring were produced by combining egg and sperm pools with sperm and egg pools from the same colonies (purebreds: parental colonies sourced from one reef) or from different colonies sourced from different reefs (hybrids: parental colonies from two reefs) (Fig. 1b, see "Methods", Supplementary Table 1). Source reefs varied widely in their environmental temperature profiles calculated from remotely sensed data (Fig. 1c–h). This design, therefore, allowed for: (1) the prediction of where heat-tolerant brood stock may be located, and (2) the assessment of the possible value of intraspecific hybridization aimed at breeding heat tolerance genes into corals adapted to cooler sites.

The resulting aposymbiotic larvae were derived from 25 crosses. Larvae from each cross were then settled to produce juveniles (Fig. 1b), and subsequently inoculated with one of three Symbiodiniaceae species or a mixture of free-living symbionts within sediments. Specifically, this involved wild-type *Cladocopium goreaui* WT10 (WT1 in[18]), the heat-evolved ("Selected Strain" SS1 *sensu*[18]; both WT10 and SS1 were derived from the same monoclonal culture of a wild-type *C. goreaui*), *Durusdinium trenchii*, and the free-living mixture. *C. goreaui* is the most common symbiont associated across a range of Cnidaria and *D. trenchii* is also common and a comparatively thermally tolerant species[19]. Establishment of symbiosis with the symbiont strain in the inoculum was confirmed via high-throughput ITS2 Illumina sequencing and dominant communities confirmed with RNAseq mapping of Symbiodiniaceae transcriptomes (Supplementary Fig. 1). The free-living mixture found within the sediments was collected from the warmest reef (CU) and is presumably warm-adapted. The influence of these symbionts in driving coral temperature tolerance in select coral species like *A. millepora* and *A. spathulata* is unequivocal for traits such as bleaching[18] and survival[16,20,21] and is further quantified here. Larvae and juveniles were both exposed to 27 °C (the approximate mean annual temperature of the five reef sites where reproductive adults were collected, Fig. 1c; this was used as a non-stress temperature) and elevated temperature treatments (larvae: 35.5 °C for 56 h; juveniles: 32 °C for 58 days, Fig. 2b, see "Methods" for justification of treatment temperatures).

Mean survival of all the crosses was high in the 27 °C temperature treatment (Fig. 2a, b; larvae: mean 96.2% ± 0.4 Standard Error, median = 100%, Supplementary Figs. 2–4), although variable, particularly for the juveniles (mean 82.02% ± 1.8, median = 95.8%, Supplementary Fig. 2). Survival under the elevated temperature, however, varied widely across the 25 crosses (Fig. 2a, b), with winners and losers at both the larval and juvenile stages across the four Symbiodiniaceae treatments (Fig. 2a, b, larval cross comparisons Wilcoxon test $P.adj = 7.1e-07-0.72$, Supplementary Figs. 2 and 3 and results). For example, under heat stress, larvae from the hybrid cross LSxSB exhibited 86.8% survival [±2.86] whilst larvae from the purebred cross LSxLS exhibited only 9.1% survival [±3.0] (Fig. 2a). Although variability was high, survival responses from these laboratory tests were consistent with patterns seen in the field when a subset of crosses were outplanted onto the reef (e.g. 75–100% survival across all treatments for SBxLS juveniles[22]). Under heat stress, juveniles from crosses produced from mothers from the warmest, inshore reef in the far north (CU) generally survived the best, although the top surviving crosses were hybrids with maternal or paternal contribution from a central GBR reef. We acknowledge that a particular parent could produce particularly vigorous or inviable offspring, influencing whether a "reef" appears to confer high or low survival given the importance of parent identity on offspring survival[15,23].

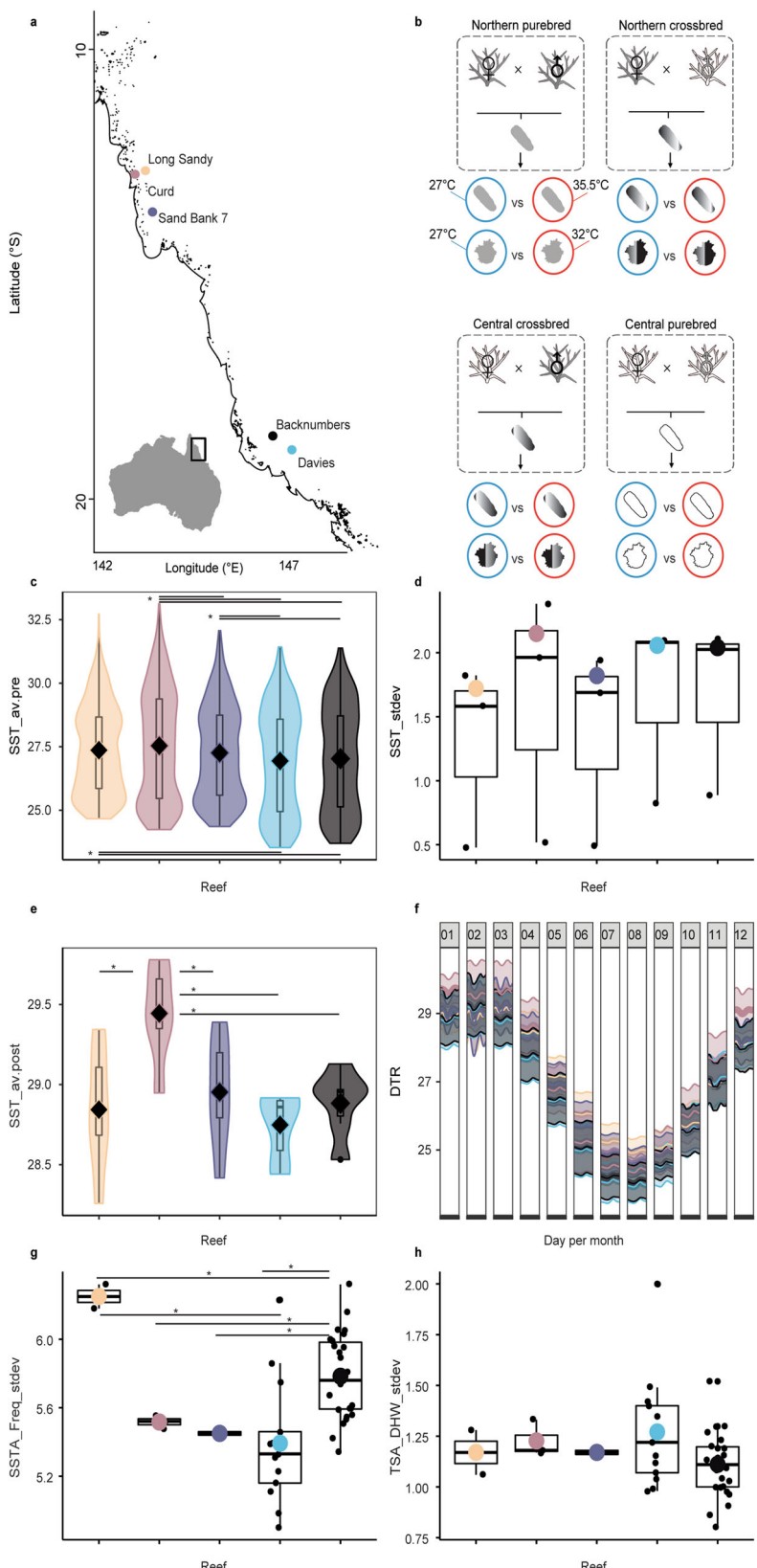

**Influence of variable symbiont communities on coral heat tolerance**. Symbiodiniaceae are taxonomically diverse with many functionally cryptic species[24]. Fitness differences have been attributed to variation in photosynthetic mechanisms and the regulation of specific genes[18] and symbiont community structure and diversity[25]. Juveniles from each cross were exposed to the same four symbiont treatments at both temperatures (at 27 and 32 °C: wild-type *Cladocopium goreaui* WT10, heat-evolved *C. goreaui* SS1, *Durusdinium trenchii*, and the free-living mixture). At the 27 °C treatment, the survival when averaged over all the crosses was high (mean 82.02% ± 1.8, median = 95.8%, Supplementary Figs. 2 and 4). When averaged by symbiont treatment,

**Fig. 1 Great Barrier Reef-scale remote sensing derived metrics and experimental design for heat-stress survival data produced from selective breeding experiments.** Sites where reproductive *Acropora tenuis* colonies were collected across five reefs spanning >6° of latitude from the Great Barrier Reef (inset **a**). The most northern reefs are shown in "warm" colours (red/orange) and the central reefs are in "cooler" colours (black/blue) (**a**). Breeding design used to create purebred and crossbred hybrid offspring (larvae and settled juveniles) encompassing various genetic backgrounds (**b**). Purebred (purebred northern: dark grey, purebred central: white) and crossbred offspring (northern mom x central dad: dark grey to white gradation, central mom x northern dad: white to dark grey gradation) were produced. These included larvae and juveniles exposed to mean (27 °C: blue) and elevated temperatures (32 °C or 35.5 °C: red). Temperature metrics were calculated from global and Great Barrier Reef specific remotely sensed data used as model parameter coefficients (**a**, **c–h**, definitions in Supplementary Table 2). Briefly, parameters in °C describe the following: Latitude **a** is the number of degrees south of the equator. SST_av.pre: average sea surface temperature between 2014 and 2016 before mass bleaching (**c**), SST_stdev: standard deviation of average daily sea surface temperatures (**d**), SST_av.post: average sea surface temperature between 2013 and 2018 including mass bleaching (**e**), DTR: daily temperature range (**f**), SSTA_Freq_stdev: frequency of standard deviation of SST anomalies (**g**), TSA_DHW_stdev: standard deviation of thermal stress anomaly calculated in degree heating weeks (**h**). Horizontal lines and asterisk denote significantly different (*P* < 0.05) comparisons in temperature parameters between reefs using the non-parametric, two-sided Wilcox tests (**c**, **d**, **e**, **g**, **h**). Boxplots include the median values (centre lines), upper and lower quartiles (box limits), 1.5x interquartile range (whiskers), and outliers (points). Derived statistics presented as box and violin plots are defined as independent observations of *n* = daily SST recordings averaged over the year (**c–e**), or *n* = weekly standard deviation of SSTA or TSA (**g**, **h**). Exact *P*-values are as follows, from top to bottom of each panel: 0.049, 3.6e−07, 1.5e−05, 0.0027, 0.023, 7.4e−05, 0.001 (**c**), 0.015, 0.026, 0.0022, 0.015 (**e**), 0.0008, 0.032, 0.051, 0.038, 0.055 (**g**).

mean survival at 27 °C was between 75.4% [±4.5]–87.1% [±2.3] (Fig. 2b). Juveniles infected with SS1 survived approximately on-par with juveniles infected with *Durusdinium* at both temperatures (27 °C: 82.5% [±3.5] vs. 82.4% [±3.3] and 32 °C: 76.6% [±4.0] vs. 77.2% [±3.3]). Juveniles infected with *C. goreaui* had the highest survival at 27 °C (87.1% [±2.3 SE]), whereas juveniles exposed to the mixed sediment community exhibited an average survival of 75.4% [±4.5 SE] (Fig. 2b). Only juveniles with *C. goreaui* performed significantly worse (*P* = 0.001) at the elevated temperature compared to the other symbiont treatments (*P* = 0.15–0.65), indicating that juveniles in both the SS1 and the sediment treatments performed as well as the "stress tolerant" *D. trenchii* at 32 °C (Fig. 2c). Further, when averaged across all hybrid and purebred crosses, survival at elevated temperatures only differed significantly from those at 27 °C for the *C. goreaui* treatment, this suggests the conferral of increased survival of juvenile corals at elevated temperatures by the heat-evolved *C. goreaui* strain SS1 compared to the wild-type *C. goreaui* (Fig. 2b), similar to the tolerance provided by *Durusdinium* and the sediment community (also see Supplementary results). At the level of individual purebred and hybrid crosses, there were no significant differences in survival between 27 and 32 °C at the juvenile stage when examined within each symbiont treatment (all juvenile cross comparisons Wilcoxon test *P.adj* > 0.05, Supplementary Fig. 4). However, cross identity significantly explained variability in juvenile survival shown through comparisons between models with and without cross as a random factor (AIC without cross: 101102.3 compared to AIC with cross: 8808.2; log-likelihood test *P* = 2.2e−16). Therefore, although differences between individual crosses were not significant (likely due to high stringency of tests) the overall cross identity was shown as a significant factor through log-likelihoods, suggestive of biological differences between crosses.

High variability in survival at both temperatures across the various host genomic backgrounds coupled with the SS1 treatment suggests interactive effects between host and symbiont genotypes[26], with juvenile offspring from mothers sourced from cooler, central reefs exhibiting particularly high survival. Previous experiments demonstrated elevated heat tolerance in aposymbiotic larvae of mothers sourced from the northern GBR[15]. Here we expand this understanding to include increased heat tolerance from central GBR mothers, and confirm previously shown[16] interactive effects that host and symbiont combinations have in provisioning heat tolerance. Interactive effects were less prominent in juveniles infected with *D. trenchii*, where the same crosses survived well at 27 and 32 °C treatments (CUxLS, DRxBK, and CUxDR, Supplementary Fig. 4). At 27 °C, juveniles infected with

*C. goreaui* were the best survivors at on average >60% across all the population crosses. However, these host-symbiont pairings suffered the highest mortality at the elevated temperature treatment, highlighting the fragility of this widespread symbiotic partnership.

Fitness trade-offs in the thermal optima of organisms are well known[27], where performance at one end of the temperature range is often sacrificed for improved performance at the opposite end. Juveniles infected with a highly diverse consortium of Symbiodiniaceae (385 Amplicon Sequence Variants, Supplementary Fig. 1, characterized via high-throughput ITS2 Illumina sequencing) sourced from the hot, inshore CU sediments, conferred the highest survival (although non-significant) at elevated temperatures to their juvenile hosts (77.7% [±3.5]) whilst suffering the lowest mean survival at 27 °C (75.4% [±4.5], Fig. 2b, c; comparison between temperatures not significantly different, *P* = 0.72). Although juveniles were exposed to the same sample of sediments originally, uptake of symbionts from the sediments by host juveniles differed in the 27 and 32 °C treatments. Symbiont communities in juveniles at 27 °C were predominantly made up of taxa from *Cladocopium* and *Fugacium*, whereas juveniles exposed to heat were dominated by these taxa in addition to the more cryptic "I" clade (see further discussion of *Fugacium* and "I" in Supplementary results). Survival under heat stress was associated with 6- and 8-fold greater abundances of I4 and C15 and 25- and 22-fold less C15g and C12 compared to surviving juveniles in the 27 °C treatment when exposed to sediments (DESeq2 Benjamini–Hochberg multiple test correction *P.adj* < 0.05). Our results reveal a potentially highly locally-adapted[28] symbiont community (given the differential abundance and performance of juveniles at 27 and 32 °C temperatures) with the ability to maintain symbiosis (i.e., corals did not bleach and die) with juveniles at elevated temperature. This was predominately driven by *Durusdinium*, C15 and previously uncharacterized taxa from clade "I"[19]. The improved performance of sediment-exposed juveniles at elevated compared to 27 °C may be due to differences between cultured and free-living symbionts or due to the novel symbiont taxa sourced from this warm reef and may represent an untapped source of adaptive diversity for corals.

**Characterizing reef habitats using remote sensing.** Experimental studies have long sought to understand the environmental factors that influence bleaching responses[13,29] by decomposing environmental data into metrics that capture latitude, degree heating weeks cumulative heat stress (DHW[30]), high-frequency

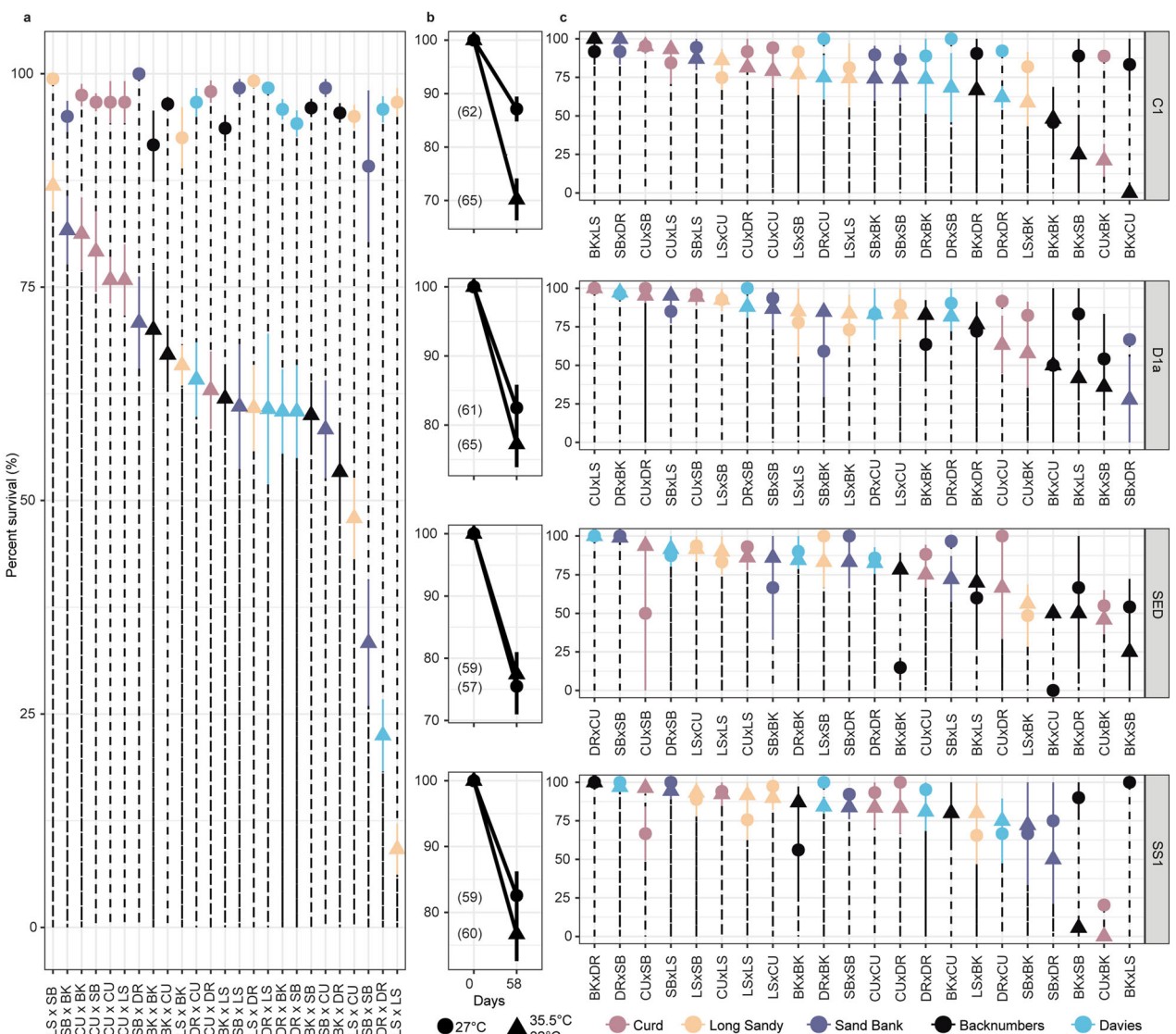

**Fig. 2 Survival of purebred and hybrid coral larvae and juveniles infected with a range of symbionts and subjected to heat stress.** Mean percent (%) survival ± Standard Error bars [SE] of larval (**a**) and juvenile crosses at mean (circles: 27 °C) and elevated (triangles: juveniles: 32 or larvae: 35.5 °C) temperatures across four juvenile-symbiont treatments (**b**, **c**). Cross labels follow the maternal coral and then paternal coral from the five reef locations (Fig. 1a). For example, SBxDR describes a cross composed of eggs from Sand Bank and sperm from Davies reef. Colours correspond to where the maternal corals were sourced (Fig. 1a). Juveniles were infected with four Symbiodiniaceae treatments listed along the vertical facets (**b**), including wild-type *Cladocopium goreaui* W10 (C1), *Durusdinium trenchii* (D1a), the free-living wild-caught mixture found within the sediments collected from the warmest reef (Curd), and the heat-evolved "selected strain" derived from *C. goreaui* (SS1). The mean survival at the final timepoint [±SE] for mean and elevated temperature treatments are shown for aposymbiotic larvae and for each of the four symbiont treatments averaged over all juvenile crosses at the two treatment temperatures (**b**) or separated by cross (**c**). Crosses for each symbiont treatment are ordered from the highest survival at elevated temperatures to the lowest survival at elevated temperatures. A summary of sample sizes can be found in Supplementary Table 3 and Supplementary Data 1. The asterisk in panel b denotes a significant difference between mean and elevated survival (*P* = 0.001). Differences in survival were assessed using the non-parametric, two-sided Wilcox test and were p-adjusted for multiple pairwise tests using the "Bonferroni" method.

short-duration daily SST variation[31], daily rate of temperature increase (DTR[32,33]), pre-heat stress (thermal history[33,34]) and the acute presence/absence of heat stress[30]. Although these metrics are potentially powerful tools to forecast stressful events, ground-truthing against in situ data has revealed relatively poor resolving power at small spatial scales (~10% of variance[12]) with improved performance over larger regions[7].

On average, during recent extreme bleaching events, corals on the GBR experienced between 8 to 16 DHW, with increasing DHW observed over time, as shown by the 2016 event relative to bleaching in 1998 and 2002[35]. When converted to DHW, the additional heat

tolerance acquired by some of the juveniles using selective breeding practices measured here equates to 2.85–3.4 DHW of additional tolerance per reef (Supplementary Fig. 5), an on average boost of 2.6–3.08 °C (mean 2.9 °C ± 0.08; Supplementary results of juvenile survival) above the expected summertime maximum temperatures for each reef for the survivors of these experiments.

The low resolving power of these aforementioned multivariate environmental models coupled with the possible disjunct between adult and offspring responses to heat (likely driven by $V_E$ and symbionts), thus requires a different approach to predicting where heat-tolerant corals occur in the wild. To achieve this, survival

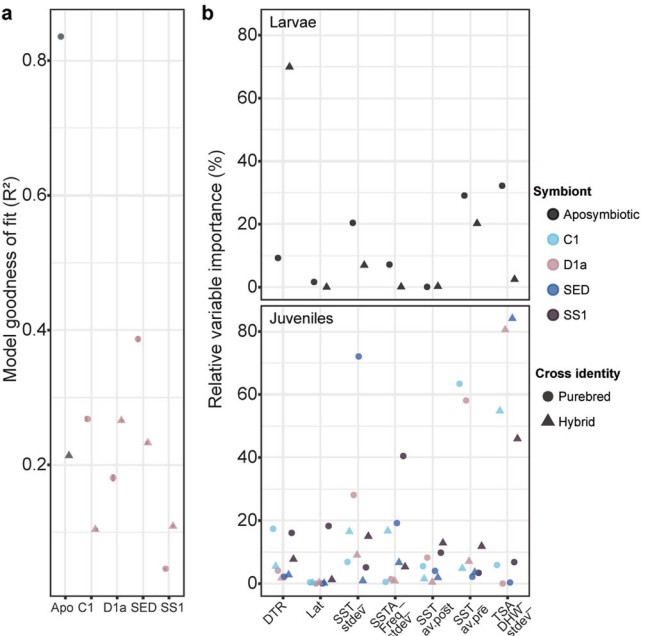

**Fig. 3 Assessment of Gradient Boosted Model (GBM) performance across all coral and symbiont combinations.** The abbreviations follow- Apo: Aposymbiotic, C1: *Cladocopium goreaui*, D1a: *Durusdinium trenchii*, SED: Sediments, SS1: Selected Strain (**a**) ($R^2$ = via Boosted Regression Trees). Relative importance of model parameter coefficients produced from GBMs in predicting larval and juvenile survival under heat (**b**). Colours correspond to different symbiont treatments and shapes correspond to reproductive cross identity. The abbreviations follow- Lat: Latitude, SST_av.pre: average sea surface temperature between 2014 to 2016 before mass bleaching, SST_stdev: standard deviation of average daily sea surface temperatures, SST_av.post: average sea surface temperature between 2013 to 2018 including mass bleaching, DTR: daily temperature range, SSTA_Freq_stdev: frequency of standard deviation of SST anomalies, TSA_DHW_stdev: standard deviation of thermal stress anomaly calculated in degree heating weeks.

analyses discussed above were combined with remotely sensed data (Fig. 1c–h) within a machine learning gradient boosted framework using a sequential ensemble approach to determine the relative importance of environmental parameters in explaining offspring survival given the reefs of their parental stock (Fig. 3a, b). These "intrinsic resistance models", represented as polynomial equations (Fig. 4a–d), therefore provide the framework for the identification of locations where corals with high heat tolerance and heritability are likely to occur. To construct these models, global (CoRTAD[36]) and modelled GBR-specific (eReefs[37]) Sea Surface Temperatures were used (full information on temporal and spatial resolution of this data in Supplementary Table 2; see "Methods" for derivations of seven environmental predictors calculated from remotely sensed data). These metrics included: Daily Temperature Range (DTR *sensu*[30]), Latitude (LAT), Standard Deviation of the daily mean annual temperature (SST_stdev *sensu*[11]), standard deviation of SSTA frequency over the entire time period anomalies (SSTA_Freq_stdev *sensu*[11]), average annual SST (SST_av.post), average annual SST pre- mass bleaching in 2016 (SST_av.pre), and the Standard Deviation of the Thermal Stress Anomaly Degree Heating Week (TSA_DHW_stdev *sensu*[11]). Potential limitations of this approach may stem from the range of data used to inform model predictions, including the year range of remote sensing data and the potential for recent temperature anomalies to bias responses. To minimize potential sources of bias, we calculated average annual temperatures both

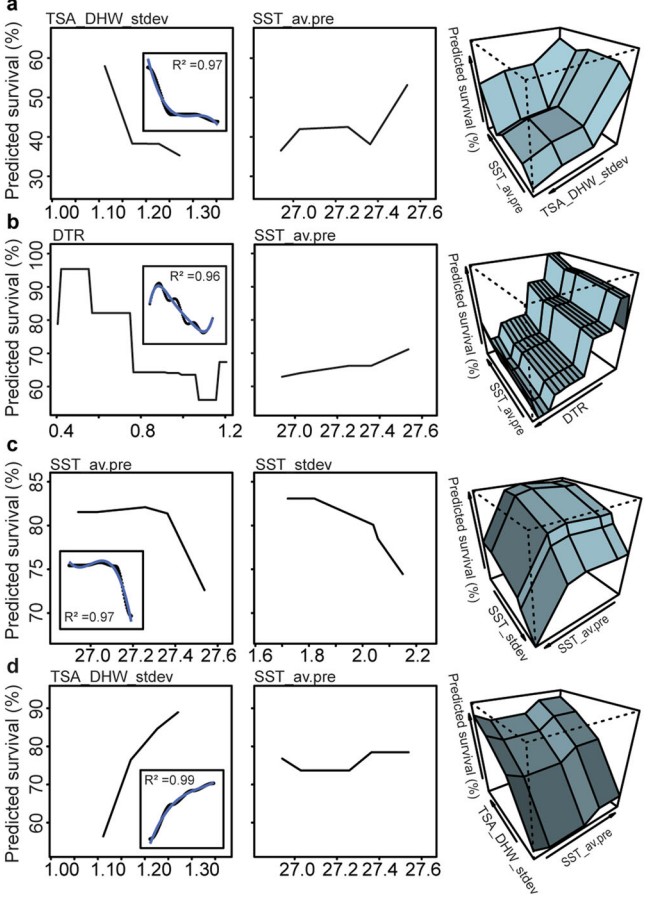

**Fig. 4 Forecasting models developed to locate reefs harbouring corals capable of producing heat-tolerant offspring.** Gradient Boosted Model (GBM) partial plots of the most important parameter coefficients for aposymbiotic larvae (**a**, **b**) and juveniles infected with *Durusdinium trenchii* (**c**, **d**). Parameters for each plot are listed in the top right-hand corner, demonstrating the predicted percent (%) survival (*y*-axis) for each model. 3-D plots show the interactive effect of each parameter estimates per life-stage and symbiont combination for predicting percent survival (blue shading to more easily show contours). Resulting intrinsic resistance models (insets, **a**–**d**) produced from GBM partial plots were used to calculate environmental conditions for optimal survival and represent the most important predictors with a corresponding goodness of fit value ($R^2$, blue line). Intrinsic resistance models capitalize on the heritability of heat tolerance, symbiotic-state, and adult environmental conditions. The abbreviations follow- SST_av.pre: average sea surface temperature between 2014 and 2016 before mass bleaching, DTR: daily temperature range, TSA_DHW_stdev: standard deviation of thermal stress anomaly calculated in degree heating weeks, and SST_stdev: standard deviation of average daily sea surface temperatures.

with (SST_av.post) and without (SST_av.pre) recent temperature anomalies. Year ranges were chosen to include both conditions before and after mass bleaching, and fall within the range of time for *Acropora tenuis* colonies to reach reproductive maturity (~ 4–5 years[38]). Satellite data were collected from a depth of about 5 m, within the upper depth range of where this species tends to be located on the reef (1–8 m[39]).

**Environmental predictors of heritable heat tolerance.** The most influential predictors of survival under elevated temperatures varied between larval and juvenile life-stage and, when ranked, followed: TSA_DHW_stdev ($R^2$ = 31.3% [±10.5]), SST_av.pre

(20.3% ± [7.2]), SST_stdev (18.1% ± [6.5]), DTR (13.7% [±6.5]), SSTA_Freq_stdev (9.8% [±4]), SST_av.post (4.5% [±1.4]), and not Latitude (2.2% [±1.8]) (Fig. 3a, Boosted Regression Trees (BRTs)). Latitude was the poorest predictor of coral heat tolerance in both purebred and hybrid offspring, which conflicts with the earlier paradigm[40,41] that warmer reefs closer to the equator produce more heat-tolerant corals[41,42], and that heat-tolerant corals are restricted to low latitudes. This aligns with recent global estimates demonstrating that latitude was the poorest of 18 environmental predictors of coral bleaching resistance[11], highlighting that the selection for heat-tolerant brood stock is not as simple as selecting individuals from warm, equatorial reefs (i.e. northern GBR).

Survival was best explained in purebred aposymbiotic larvae ($R^2 = 83.5\%$) and least explained in hybrid larvae ($R^2 = 21\%$) or juveniles infected with heat-evolved SS1 ($R^2 = 4$–$10\%$) (Fig. 3a). Purebred aposymbiotic larvae are composed of a genomic architecture shaped by the local selection of climatological pressures of only one reef and are not influenced by symbionts, and may therefore result in heat tolerance behaviour that can be more precisely modelled. Alternatively, environmental data for hybrid crosses only incorporated information from the maternal parent (full justification of design in "Methods"), potentially diminishing the explanatory power of the models. Crossing corals from different reefs with divergent genomes resulted in hybrid offspring with increased genetic diversity[16,43], which is apparent in our data. The variance in survival in aposymbiotic purebred larva was strongly driven by TSA_DHW_stdev (32.2% relative importance of total $R^2 = 83.5\%$) compared to hybrid larvae. Survival of purebred juveniles was also better predicted than that of hybrid offspring (21.5% vs. 17.8%) (Fig. 3a, b). Although heat tolerance may be shaped by other factors, including the bacterial community[44], the strong interplay between host and Symbiodiniaceae genotypes[45] is reflected in the variable influence in environmental predictor contribution across these experimental treatments. The incorporation of hybrid offspring into these models serves to understand the optimal sourcing of parental brood stock to produce hybrids given their potential relevance for applied conservation practices for enhanced heat tolerance[16]. Some level of admixture between northern and central GBR regions occurs in *A.tenuis*[46], suggesting interpopulation hybrids are likely produced in the wild, although they are likely relatively uncommon events over short time scales[47]. Finally, the relative contribution of each environmental predictors in explaining the overall model variance varied by Symbiodiniaceae taxon (Fig. 3b). Hence, the early life-stage data presented here coupled with previous studies on adult corals[34,48–50] integrates the contribution that in situ temperature variability has on the heat tolerance of all coral life stages (larvae, juveniles, and adults). When combined with global correlation analyses of adult bleaching patterns[11], our findings confirm that greater SST variability reduces the odds of coral mortality under heat stress.

**Models of intrinsic resistance**. To develop intrinsic resistance models, four host-symbiont combinations (purebred and hybrid aposymbiotic larvae and juveniles infected with *D. trenchii*) that exhibited optimal predictive power and high overall survival under heat stress (see Supplementary for full justification and "Methods") were used to calculate environmental conditions that would result in high survival (>90%) under heat stress. This was done through the construction of GBM model prediction partial plots (Fig. 4a–d and insets, $R^2 = 0.96$–$0.99$) and resulted in 3rd–5th order polynomial equations (Supplementary results). Solving these equations (Fig. 4a–d) thereby allowed us to resolve reef locations along the GBR where genetically predisposed heat-tolerant corals should

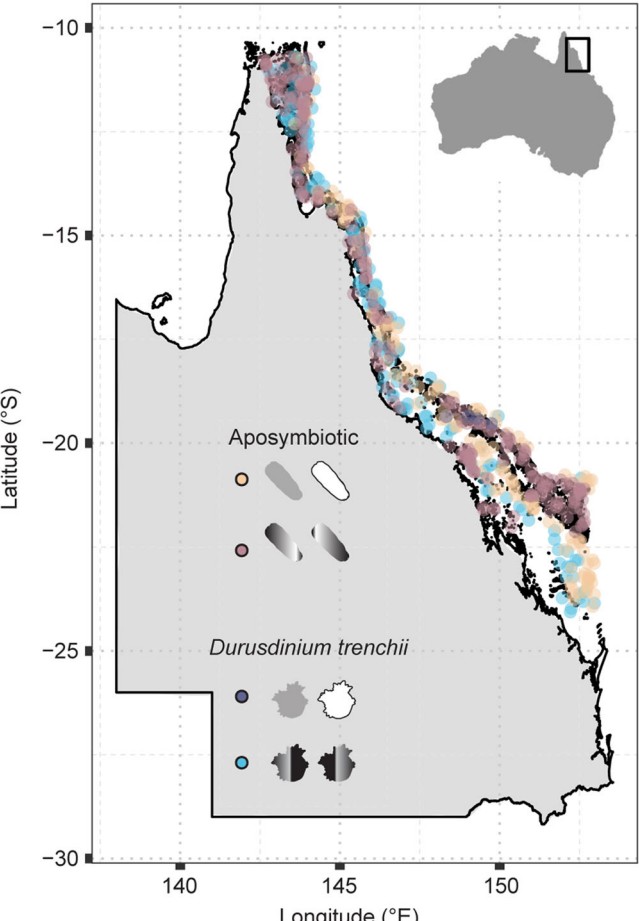

**Fig. 5 Intrinsic resistance models identify hundreds of potential reefs as predicted sources of heat-tolerant adult corals able to produce offspring of optimal survival under heat stress.** Model outputs are presented across four genetic and symbiotic combinations: (tan) purebred, aposymbiotic larvae, (pink) hybrid, aposymbiotic larvae, (purple) purebred juveniles infected with *D. trenchii* (D1a), (blue) hybrid juveniles infected with *D. trenchii*. Reef locations across the Great Barrier Reef (°South to °East) were calculated from intrinsic resistance models. Reef habitat environments were best characterized by the standard deviation of the thermal stress anomaly in degree heating weeks, daily temperature range, and average annual sea surface temperature.

exist, in particular, tolerant maternal genotypes (via maternal genetic or environmental effects) (Fig. 5). In the search for heat-tolerant corals, many reef habitats and large latitudinal clines[15] to semi-isolated backreef pools[51] have been surveyed. Hundreds of reefs were identified here as potential predicted sources of heat-tolerant corals also exhibiting high heritability for heat tolerance (Fig. 5). Of the approximately 3255 reefs within the GBR marine park boundaries, in total, approximately 251 unique reefs or closely adjacent areas (~7.5% of total reefs and potentially non-reefal locations) were identified as potential sources of heat tolerant parental stock for producing larval and juvenile corals capable of surviving temperatures from 32 to 35.5 °C.

The development of novel reef restoration approaches includes the use of assisted evolution[52]; including the increase of heat tolerance of corals through selective breeding[16] and the directed evolution of their symbionts[18]. Hybridization facilitates adaptation by promoting the introgression of genes for high temperature tolerance into the genomic background of corals in cooler environments, thereby preparing those populations for future warming[53]. The outcrossing of populations has rendered

improved fitness relative to control populations in a variety of organisms[53] by increasing genetic diversity in offspring in genomic regions associated with heat tolerance[16]. It should be noted that one key limitation of this work is its reliance on data produced from a single coral species, *Acropora tenuis*. It is therefore unclear how generalizable these models are to other coral taxa. However, as data are generated from the reproductive crosses of novel species in future years of work, these models can be easily updated to incorporate taxa that represent different ecological and evolutionary life-histories. Another limitation of this study is that this approach does not fully capture the variance in heat-tolerance across different coral genotypes or their symbionts within reefs. A poignant example of this can be seen within a single colony, in which there is variability in heat tolerance among Symbiodiniaceae species and strains that dominate shaded vs. exposed sides[54]. Finally, the multitude of factors influencing the heritability of heat tolerance makes ascribing drivers of tolerance challenging[15,17,51]. For example, holobiont tolerance is influenced by the algal and bacterial symbionts, and the acclimatory history and potential of the host and symbionts (reviewed in[55]). Although physiological experiments are useful for investigating if tolerance can be transferred between generations, further work is needed to confirm if this is maintained in natural environments. Therefore, given the multitude of different factors influencing heat tolerance, it is possible that this design may not fully capture the adaptive potential of a single reef that can therefore be extrapolated to the entirety of the GBR.

To assess the influence of individual corals or specific reefs on the BRT models and ultimately, intrinsic resistance equations, we compared our Cross Validation method (Repeated k-fold CV), which takes random subsets of larvae taken from across all the study reefs, against three other CV methods (Bootstrap, k-fold, and Leave One Out), and as well as sequential reef exclusion (Fig. 6). There was no evidence of overfitting from our choice of random repeated k-fold CV, and the percent variation in the data explained using each CV method was almost equivalent (model goodness of fit $R^2 = 0.837–0.835$), as were the resulting predicated intrinsic resistance models (Fig. 6a, b). When reefs were excluded sequentially from each dataset (larval purebreds, larval hybrids, juvenile D1 purebreds, juvenile D1 hybrids) and compared to each of the final models with all reefs included (Fig. 6c–f), the inclusion or exclusion of any single reef did not drive the resulting models. Instead, the models were influenced by each reef and were dependent on the life-stage and symbiotic state. Specifically, purebred larval responses were mainly influenced by Davies and Backnumbers, the larval hybrids by Long Sandy, juvenile D1 purebreds by Curd, whereas juvenile D1 hybrids were not influenced by any one particular reef compared to the others. This comparison between our BRT analysis, which CV removes observations at random (Fig. 6 "All reefs"), versus when whole reefs are left out (Fig. 6, all other reefs) highlights the importance of including as many reefs as possible in the BRT models but suggests that not one reef influenced the overall model construction more than others. It is important to note that the ability to extrapolate to all GBR reefs is likely limited by the relatively small number of reefs ($n = 5$) used in this study. Although all four methods of cross-validation did not show any sign of overfitting (including the drop-one method implemented across all reefs), and the models do well in predicting survival when all reefs are included (Fig. 6a, b and Supplementary Fig. 10a, b), the ability of the models to predict when whole reefs are removed may be limited given when a whole reef is omitted from the analysis, the model predictions of survival are quite different in some cases (Fig. 6 and Supplementary Fig. 10c). This would be expected given the exclusion of whole factors from the

dataset and suggests the importance of collecting data from more reefs to generate precise predictions about reefs where we do not yet have data. Although the predictive power of our models does not seem to be biased towards any particular reef, we acknowledge that it is hard to capture the full diversity of biological and biophysical parameters of all the >3000 reefs with the only five reefs measured here. This is important because we recognize that any differences between these analyses due to the removal of one reef would imply that the ability to infer temperature tolerance on reefs where data have not been collected yet (i.e. naïve reefs) could vary depending on the data input into the model. Hence, the model's ability to predict the situation on reefs not included in the fitting is potentially limited. The flexibility of the BRT framework however will easily incorporate new coral species, new reefs, and new experimental data, which can be used in the future to optimize these methods.

Our intrinsic resistance models improve the usefulness of the information gathered from global searches by combining off-spring phenotypes and remote sensing to identify environmental conditions that result in high heat tolerance and the transgenerational transfer of survival under heat stress. These environmental conditions include high average annual temperatures (SST_av.pre = 26.69 °C) typified by high daily variability (DTR ~ 0.515 °C) or highly variable thermal Stress Anomalies (TSA_DHW_stdev = 1.081–1.319). Identified sites spanned the GBR and contained a high density of reefs in offshore sites, including southern offshore areas, central offshore, and/or far northern region. This signature of heat tolerance spread across the latitude and longitude of the GBR may result from balancing broad-scale selective pressures versus local-scale disturbances and resource partitioning. This wide distribution of potential refugium reefs implies that heat-tolerant brood stock may also encompass individuals with the ability to survive at lower temperatures.

The number of reefs estimated to have corals that would be potential producers of heat-tolerant offspring is encouraging given the current high incidences of coral mortality. The existence of potentially heat-tolerant genotypes throughout the GBR may explain the patchy nature of bleaching-related mortality observed during previous bleaching events[35]. Environmental history may also influence bleaching tolerance, for instance, comparisons of bleaching responses between the 2016 and 2017 relative differences in DHW showed some reefs that were predicted to bleach did not, a phenomenon termed "ecological memory"[56]. This ecological memory may be explained by locally variable environmental conditions across reefs (e.g. flushing), changes in symbiont communities, or shifts in heat tolerance due to the differential mortality of corals. Hence, further research should include model validation through the collection of corals at reefs predicted by the models to assess whether these parent corals do produce tolerant offspring as predicted.

Latitude was a poor predictor of offspring survival under elevated temperatures, indicating that heat resistant corals may occur across the length of the GBR and caution against the widely held assumption that only corals in the northern GBR are comparatively heat-tolerant. The widespread occurrence of heat-tolerant coral genotypes predicted by our models suggests the possibility that natural adaptation is greater than previously thought given the relatively small spatial scale over which adaptive alleles must disperse to rescue poorly heat-adapted coral populations, and assisted gene flow over small spatial scales may have value for reef conservation and restoration. Further work is needed across more coral species from a range of divergent lineages to assess if the conclusions drawn here for *Acropora tenuis* can be generalized. Given the size of the GBR, intrinsic resistance models could be applied to facilitate the selection of brood stock aimed at producing large numbers of heat-tolerant

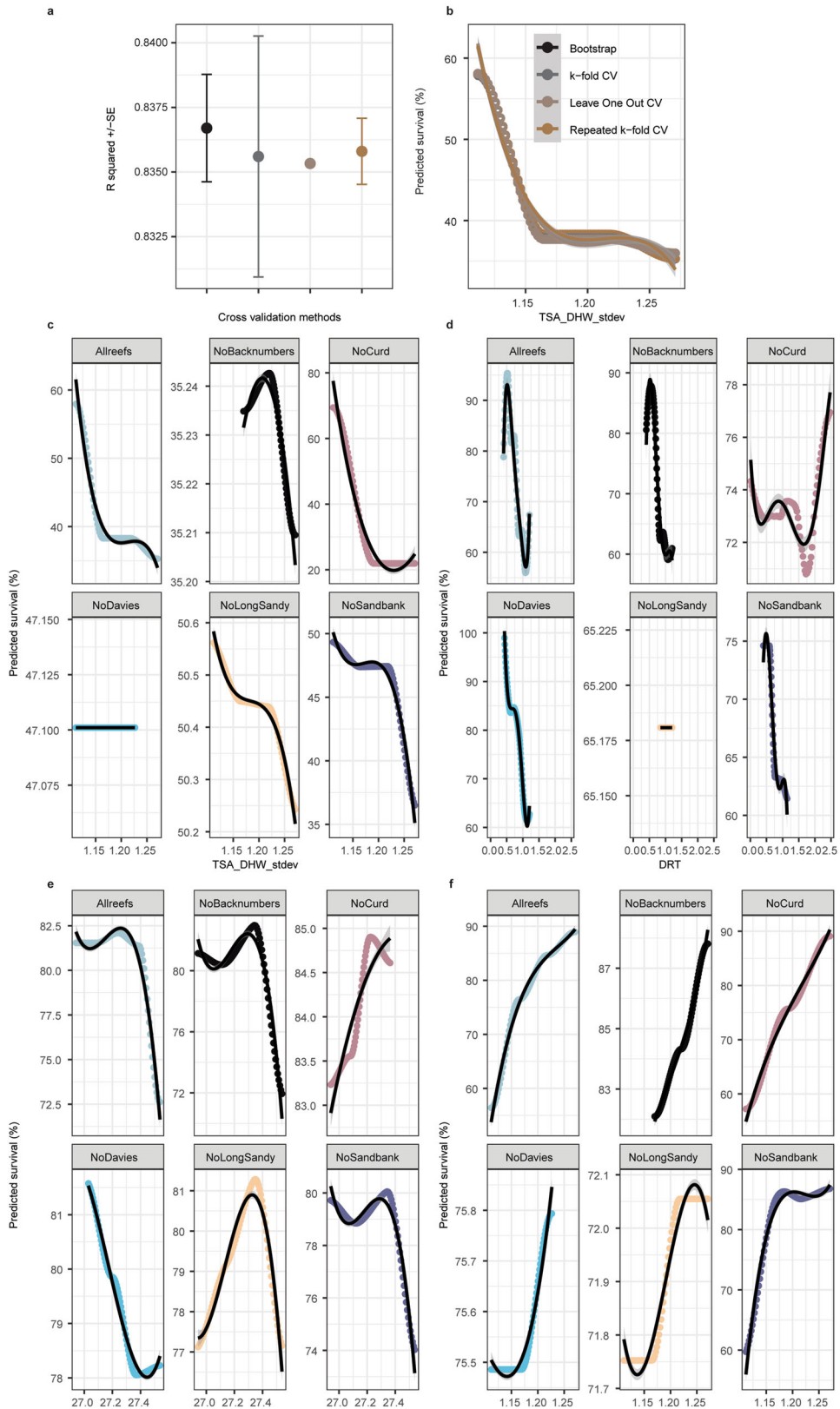

**Fig. 6 Assessment of the impact of various Cross Validation (CV) methods on model construction.** Three CV methods were compared to the chosen Repeated k-fold CV, including Bootstrap, k-fold, and Leave One Out (mean $R^2$ ± Standard Error bars [SE]). Derived statistics are defined as $n =$ independent $R^2$ estimates for each of the 3000 model runs per CV method (**a**) using a subset of the data (purebred larval crosses) to inspect the resulting impact on the intrinsic resistance models (**b**). Further validation of predicted percent (%) survival included dropping whole reefs from each of the datasets (**c** - purebred larva, **d** - hybrid larva, **e** - purebred D1 juveniles, **f** - hybrid D1 juveniles). Please note the differences in y-axis values between groups of panels. Y-axis values within each analysis group was consistent but were allowed to vary across groups to facilitate comparisons between predicted versus actual survival estimates. Colours correspond to different CV methods or where the maternal corals were sourced (**c**–**f**).

offspring for the restoration of damaged reefs. This method will aid in resilience planning[57] and is a powerful approach for targeted species conservation and the management of protected areas.

## Methods

**Coral spawning and larval rearing**. The research work was done with *Acropora tenuis* and complies with all relevant ethical regulations (Great Barrier Reef Marine Park Authority (permit number G18/41667.1). Compared to other species previously used for selective breeding (i.e. *Acropora spathulata*[16]), it has a higher propensity for successful settlement to known cues[58], which is essential for large-scale juvenile experimentation (see below for information on larval settlement). Gravid *Acropora tenuis* colonies were collected from three reefs in the far north of the Great Barrier Reef (Long Sandy, Curd, and Sand Bank 7) and three reefs in the central GBR (Backnumbers and Davies) (Fig. 1a). Ten gravid colonies per reef were collected from each of Long Sandy, Curd, Sand Bank 7, and Davies, and 15 gravid colonies from Backnumbers. Not all colonies per population spawned, and the total number of maternal and paternal colonies used per cross is listed in Supplementary Table 1. Colonies were isolated at dusk, from the 26th to the 29th of November 2018, into individual containers. Gamete bundles were released between 1800 and 1930 h and collected from the water surface. Eggs and sperm of each colony were separated through a 120 μm sieve and washed three times using 0.2 μm filtered seawater (FSW). Spawning, fertilization, and rearing followed established methods[59]. Briefly, purebred and hybrid offspring were produced by pooling eggs and sperm, respectively, from colonies collected from each reef and combining egg and sperm pools with sperm and egg pools from the same colonies (purebreds: parental colonies sourced from one reef) or from colonies sourced from different reefs (hybrids: parental colonies from two reefs) (Fig. 1b and Supplementary Table 1). Eggs were fertilized between 2100 and 2200 h by adding equal numbers of eggs from one parental colony with equal concentration of sperm from a separate parent colony. Hence all genotypes contributed equal numbers of eggs and sperm to each population cross. Sperm was diluted to $1 \times 10^6$ sperm cells per litre following counts using an automated sperm counter (Computer-Assisted Semen Analysis-CASA equipment). Fertilization success was verified by visually inspecting embryos every hour for three hours under magnification. Embryos and developing larvae were maintained in 15 L rearing cones (1 larva mL$^{-1}$). All potential cross combinations successfully produced larvae.

Seven, 6, 8, 7, 7 colonies from BK, DR, CU, LS, SB contributed eggs to the experiment, respectively, and 11, 12, 16, 15, and 14 colonies from those reefs contributed sperm to the experiment, but not all colonies contributed sperm or eggs to all relevant crosses involving that reef (Supplementary Table 3). To determine if differences in the number of parents per cross contributed to differences in survival in the larval and juvenile life-stages, Pearson correlation coefficients ($R^2$) and corresponding significance values were calculated (Supplementary Fig. 6). The number of parents explained little variation in either larval or juvenile survival ($R^2 = 0.0036$–$0.082$, $P = 8e^{-5}$–$0.18$). To account for any potential influence of the reef of origin of egg and sperm donors used in each cross, survival models (see below), incorporated both maternal and paternal identity as fixed effects.

**Larval and juvenile heat stress experiments**. Larvae and juvenile crosses were both exposed to 27 °C and elevated temperature treatments (35.5 and 32 °C for larvae and juveniles, respectively). The temperature treatments were chosen to: 1) facilitate comparison with previous selective breeding experiments in which aposymbiotic selectively bred larvae were subjected the same level of extreme heat (35.5 °C[15]). Experimental conditions for symbiotic juveniles were chosen to match conditions experienced by symbiotic adults in the wild, again using established methods to facilitate comparison[60]. Briefly, maximum seas surface temperatures (SST) were calculated for reefs from which breeding corals were sourced. For example, maximum SST from reefs used in heat stress tests of selectively breed juveniles was ~31 °C (Fig. 1a[60]), and a similar value was therefore chosen as a maximum experimental temperature. Similarly, average maximum temperatures here were calculated, and were on average warmer, with maximum average SST slightly above 32.5 °C (Fig. 1c, this study), so the elevated temperature was set to 32 °C. Finally, the total DHW experienced on reefs varied across the GBR and in 1998, 2002, and 2016, spanning from 1 to up to 15 DHW, with a majority from 8 to 16[61]. To roughly approximate the bleaching related conditions of adults in the wild, an experimental temperature and length of 32 °C for 58 days (~8 weeks) approximated the range of DHW experienced on reefs in 2016, where juveniles from each reef experienced a range of up to 24.7 DHW over the experiment covering responses to current as well as projected future increases in heat stress (Supplementary Fig. 5). A summary of the experimental design for larvae and juveniles is provided in Fig. 1b. Samples sizes per treatment are given in Supplementary Table 3 and Supplementary Data 1.

Larvae were reared in 15 L cones stocked at densities of one larva/mL with constant flow through in the National Sea Simulator (SeaSim). Aposymbiotic larval heat stress was assessed using floating net-wells[62]. Net-wells are light-weight plastic containers with one end open (top) and closed by a fine micron-mesh (bottom), originally used to strain cells. Individual, separated net-wells were placed into floating holders such that the well, and therefore larvae contained with it, were suspended below the surface of the water. Four days post spawning and fertilization, 20 larvae were aliquoted into floating net-wells, which were each separated in distinct well replicates ($n$ = minimum of six replicate wells) for each population cross. Water temperatures in the elevated treatment were ramped 0.5 °C/h until temperatures reached 35.5 °C. Larvae were subjected to a 27 °C and an elevated temperature treatment for 56 h. The 27 °C treatment represents the approximate average temperature for the five reef locations used in this study and was used as a non-stress treatment.

To quantify the interaction of host genetic variation with symbiont genotype in determining juvenile survival under heat, coral larvae were exposed to autoclaved crustose coralline (CCA) algae as a settlement cue to trigger metamorphosis and settlement. A subset of the coral larvae was settled into three wells each ($n$ = three replicates per cross) within sterile six-well plates using autoclaved CCA added to each well. In juveniles, exposure to heat occurred once symbiosis was established in which juveniles were subjected to 58 days of either 27 °C or the elevated temperature stress treatment. Once juveniles were settled, cultured symbionts from *Cladocopium goreaui* (C1, SCF 055–01.10), *Durusdinium trenchii* (D1a, SCF082), and the heat-evolved strain of *Cladocopium goreaui* (SCF055-01.01, as per best results from[18]; SS1 - "Selected Strain"), were added to each well for a final infection density of $10^5$ cells per ml and were allowed to infect juveniles. The sediment treatment was therefore a choice treatment, whereas the other treatments were not (only a single algal strain was provided). To assess the resulting symbiont communities in coral juveniles, which are difficult to assign taxonomically with only light microscopy and whose growth may have been influenced by culture conditions[63], different batches of single coral juveniles were assessed using two independent next generation sequencing methods (amplicon and RNAseq) to justify the efficacy of infection of each cultured strain (see below and Supplementary results). We acknowledge that long-term culturing may have changed these strains physiologically/genetically in comparison to fresh isolates. However, we are interested in the heat tolerance of these specific cultured strains, given their potential use in coral reef restoration initiatives. Infection included two rounds of dosing (23/12 and 26/12), following established methods described in[60]. Simultaneously with the cultured symbiont infection, replicate plates of aposymbiotic settled juveniles for each cross were added to each of the three replicate tanks with sediments collected from Curd reef. Hence, exposure time to symbionts was the same for all Symbiodiniaceae treatments. Sediments had been flown down to the SeaSim in conjunction with the spawning corals and had been maintained with gentle aeration and constant filtered FSW flow.

**Experimental tank set-up**. One to two six-well plates were added to each tank, giving three to six wells of juveniles per tank replicate. There were three replicate tanks at 27 °C and three replicate tanks at elevated temperature, ramped to 32 °C at 0.5 °C interval increases per hour per symbiont treatment, totalling three replicate tanks at each temperature for *C. goreaui*, *D. trenchii*, SS1, and the sediment treatment for 24 total tanks. LED lights were affixed above each tank, providing illumination set at ~171 PAR from 10:00 to 18:00 (12:12 day: night light cycle), with a 2 h ramp. All plates with juvenile corals were floated vertically in tanks.

Juvenile survival, bleaching and growth were assessed from photos that were taken using a Nikon D810 with a Nikon AF-S 60 mm *f*/2.8 G Micro ED Lens with four Ikelite DS160 Strobes, in which all images included a scale bar and mini coral bleaching colour-reference card[64]. Eight timepoints of photos were taken to capture bleaching and mortality of juveniles, started one day prior to when heating started (T1: 30/12–T8: 25/2). Images were analyzed using ImageJ and only single juveniles were retained for analyses (i.e. no juvenile aggregations formed by larvae settling next to each other). Survival was scored as a binary outcome ("2" = alive or "1" = dead).

**Survival statistical analyses**. All analyses were run in R (v. 3.6.0). Larval survival was based on the number of remaining larvae per replicate well at the final sampling time point. The final time point was chosen as approximately the time needed for which 50% of the larval crosses would reach 50% survival at the elevated temperature treatment. Survival was quantified as the average percentage of surviving larvae across each of the replicate wells as compared to the initial number of larvae added per well (Supplementary Data 1).

Replicate, single juveniles per cross were tracked through time using photographs, allowing for individual-based survival analysis to be undertaken using the "survival" package[65] (Supplementary Fig. 7 for juvenile survival experimental design). Individual juveniles were assessed from photographs as either "alive" or "dead" at the final timepoint compared to initial photographs. This data was input into survival models. Survival models were performed using Kaplan–Meier methods using the survfit function; with symbiont treatment, temperature, tank replicates, cross, maternal reef, and paternal reef identities accounted for as fixed effects. As per standard output of survival models, the output included the number of censored individuals per fixed effect treatment and the total number of events (i.e. total alive individuals left at the final timepoint). Percentage survival and standard error for replicate juveniles were calculated per treatment by dividing the number of juveniles (n.event) by the total number of potential individuals to die given initial numbers (n.event + n.censor). Replication included replicate individual juveniles, within replicate wells, replicate plates, within replicate tanks

per symbiont by each temperature combination (Supplementary Fig. 7 and Supplementary Data 2).

Larval and juvenile survival (amongst and between crosses) and between larvae and juveniles were assessed using the non-parametric, two-sided, p-adjusted for multiple pairwise tests with multiple grouping variables value using the "Bonferroni" method with the Wilcox.test from the package "ggpubr"[66]. This non-parametric test does not assume normality in the data. These tests included the comparison of the mean values between each group of crosses at 27 and 35.5 °C (two independent groups) to assess if they were statistically different at the final timepoint (Supplementary Figs. 3 and 4). This includes comparisons within and between larval and juvenile crosses at both temperatures, as well as between the larval and juveniles' stages (Supplementary Fig. 8). To verify these results and calculate the variance due to fixed and random effects, a generalized linear mixed effects model was run using a negative binomial distribution in lme4[67]. The interactive effect of temperature and symbiont treatment were fixed effects and replicate tank, and cross were set as random effects; without an intercept. These results corroborated those from the Wilcox.test (Fig. 2b). Model selection was performed with AIC and the log-likelihood ratio tests using the "anova" function (Supplementary Table 4 and results).

**Molecular identification of symbionts**. For amplicon Miseq and RNAseq sequencing of Symbiodiniaceae taxa in coral juveniles, individual juveniles were sampled on day 58 from the 27 °C and elevated temperature treatments, preserved in 100% ethanol or *RNAlater* and stored at −20 °C until DNA and RNA extraction. DNA was extracted using a modified Gloor and Ingles extraction method and amplified using the following primers: ITS2alg-F (5′-TCGTCGGCAGC GTCA-GATGTGTATAAGAGACAGGTGAATTGCAGAACTCCGTG) and ITS2alg-R (3′-TTCGTATATTC ATTCGGCCTCCGACAGAGAATATGTGTAGAGGCTCGG GTGCTCTG-5′)[64,68]. Amplicon data was analyzed using a modified DADA2 pipeline for Symbiodiniaceae that is published[64,68] and fully available via Github https://github.com/LaserKate/MontiSymTransgen. The resolution of taxonomic issues with ITS2 are well known (reviewed in[69]). The DADA2 pipeline incorporates multiple methods to account for this; including base-pair quality scores incorporated into the learned error models, the DADA2 high resolution sequence variance inference algorithm, the removal of substitution and indel errors, and most importantly, the RDP's naive Bayesian classifier to assign taxonomy. The database to assign taxonomy was updated from GeoSymbio[70] to the more recent[71]. DESeq2[72] was used to calculate significantly differentially abundant ASVs within the juveniles exposed to the sediment treatment between the two temperature treatments. Adjustments for multiple comparisons were then applied using the Benjamini–Hochberg multiple test correction which calculates the upper limits of the False Discovery Rate and then re-estimates and penalizes relevant *p*-values. Amplicon sequencing was performed on the following number of individual juveniles: n = 12 (*C. goreaui*), 7 (*D. trenchii*), 30 (SED), 11 (SS1). As a secondary confirmation of the taxonomic identities of symbionts in each treatment, additional biological replicates of coral juveniles per treatment (n = 29 (*C. goreaui*), 38 (*D. trenchii*), 43 (SED), 36 (SS1)) were sequenced using RNAseq and Symbiodiniaceae transcriptomes were mapped to these reads using standard tag-based methods[73].

**Remotely sensed data**. Remotely sensed data were extracted from two data sources. First, derived metrics like TSA_DHW_stdev were sourced from the Coral Reef Temperature Anomaly Database (CoRTAD) (v.6). This is a global database of weekly, 4 km resolution Sea Surface Temperature (SST) and related Thermal Stress Metrics running from 1982-01-02 to 2019-12-27. The remaining metrics were calculated from 1 km eReefs models from −2.35 m depth[37] (see description of each metric in Supplementary Table 2). The use of remote-sensing tools to assess the spatial distribution of heat tolerance traits or bleaching resistance traits facilitates the rapid linkage of environmental data and coral phenotypes. Often, in situ logger measurements are used for this purpose. However, over larger spatial scales, like the GBR, this is challenging and would necessitate the use of remotely sensed data. Therefore, eReefs data was used to bypass the need for in situ logger data, where the eReefs data have been groundtruthed against real time in situ measurements and were found to be within 0.5 °C[37].

**Remotely sensed metrics statistical analyses**. Statistical differences in environmental Sea Surface Temperature metrics (Supplementary Table 2 for descriptions) were performed using two-sided, Wilcox.tests as described above.

**Boosted regression trees**. The relative importance of these temperature metrics as explanatory variables was assessed using Classification and Regression tree methods, specifically BRTs[74]. BRTs utilize machine-learning (ML) methods to produce aggregations of single-tree methods. A sequential ensemble approach was used to determine which predictors best explained larval and juvenile survival under heat stress by constructing BRT predictive models. The strength of these models lies in their ability to maximize predictions through the building, selection, and pruning of different models to maximize predictive power. BRTs were implemented as Stochastic Gradient Boosting machine models (GBMs) in the R package "caret"[75], used for non-linear regression and classification.

Another strength of GBMs is their ability to integrate and predict across multiple variables[74]. BRTs were chosen over other ML methods as they outperform other methods like Random Forest when predictor data encompass missing values (NAs)[74]. The relative importance of different predictors is quantified through assessments of performance calculated by the number of times a predictor is used to split tress. R-squared ($R^2$) was used to rank the usefulness of each model by the proportion of variance explained in survival per environmental variable within each GBM.

GBMs were calculated within the R statistical framework using the following packages: "caret"[75], "rlang"[76], "gbm"[77], "rsample"[78], and "plotmo"[79]. Base R's random number generator (RNG) was set to the following: "Mersenne-Twister", "Inversion", "Rounding." Cross validation (CV) is important during model evaluation, often when data are limited. There are a number of CV methods, including K-fold cross-validation, leave-one-out cross-validation and bootstrapping. These methods can be used to test model performance on a subset of the data (random or selected) to evaluate model performance, even with small datasets and helps to determine the number of trees appropriate that balances overfitting versus explanatory power. To avoid overfitting the optimum number of tress, which is of particular concern for small datasets, K-fold cross-validation was performed with a 10-fold CV using a "random" search of the hyper-parameter space. During this process, the data was split into 10 randomly chosen testing subsets and was repeated to make 10 training sets, meaning that CV involves randomly splitting the data into $10 \times 10$ ($nt$[74]) random splits and repeated, with each step randomly omitting one test set for validation and optimal $nt$. The use of 10 folds is commonly used and recommended in method guidelines[74]. To test the appropriateness of this parameter choice on our data, further diagnostics were run for each model to assess if CV adequately balances the squared error loss versus the number of trees (Supplementary Fig. 9), in which the optimal number of trees should reflect where there is a decline in prediction error to a minimum value (dashed line) compared to independent training test estimates (solid line). These figures suggests that 10 randomly chosen subsets of the data (error on the test data) performed very similarly to the training dataset, with the optimum number of interactions (trees) designated in grey. The minimum error values should be chosen before the point of increase, which represents where model overfitting may start to occur. Furthermore, we also tested if our method of CV using random partitioning influenced the downstream predictions of the models, if the model's ability to predict was influenced (as measured by $R^2$) or influenced by overfitting. To do this, we tested our CV method (Repeated k-fold CV) against three other CV methods (Bootstrap, k-fold, and Leave One Out) using a subset of the data (purebred larval crosses) to inspect the resulting impact on predicted survival models (Supplementary Fig. 6a, b).

BRTs were built for purebred and hybrid crosses separately at both life-stages (larvae and juveniles). BRTs for purebred crosses were built using environmental covariates sourced from a single reef, i.e. crosses in which both parents were sourced from Curd reef incorporated environmental data from only Curd reef. Alternatively, hybrid crosses, by definition, have parents sourced from two reefs. In these cases, the environmental data used in BRTs were provided only for the maternal parent. For example, the environmental data produced for hybrid offspring from a Curd mother and Backnumbers father followed environmental data from Curd reef. Therefore, these models by design will identify source reefs for either maternal or paternal corals for purebred offspring, or only maternal corals for hybrid offspring. This decision was based on the higher contribution by the maternal parent in influencing key traits including survival, settlement, and symbiosis[59] and heat tolerance[15]. Juvenile data were further divided by the type of symbiont infection (*C. goreaui*, *D. trenchii*, SS1, sediments). The following levels were treated as statistically independent observations given the measurements were taken directly at these levels: each replicate well of pooled larvae and each single juvenile. Further, whilst there were only five "maternal" reefs, this dataset consisted of 25 unique cross combinations when these five maternal reefs were crossed with the same five paternal reefs, increasing the predictive power of the models given there were 3.5x more combinations of reefs compared to the seven predictor environmental variables.

Model performance was also validated against a range of parameter values. This included a grid-search cross validation step that was also incorporated to prevent overfitting of BRTs. First, a range of hyperparameters were tested using "tuneGrid" in caret to explore the trade-offs between overfitting and variance explained in the model. The parameters tested included: shrinkage, interaction depth, minimum number of nodes, and bag fraction. Various combinations of these parameters ($11 \times 11$ combinations over 3000 interactions each using 99% of the data to train against) were tested and assessed against overfitting diagnostics visualized with the plotres function from the package "plotmo."

The final model parameters were chosen from the tuneGrid iterations using the minimum RMSE score and diagnostic plots. In assessments based on diagnostic plots, 5/10 calculated model hyperparameter combinations that resulted the lowest min_RSME value were chosen, whereas the other five required further tuning. The model of larval purebreds, where the depth.interaction of 3 was used instead of 1. The diagnostic plot for *C. goreaui* purebreds suggested that the best combination of parameters that balanced between squared error loss and minimum number of trees included the following: shrinkage (0.01), interaction.depth (5), n.minosbsinnode (10) in contrast to the suggested: 0.3, 1, and 15. In the model of sediment purebred juveniles, an interaction depth of 1 (instead of 5), shrinkage of

0.3 (instead of 0.1), and n.minosbsinnode of 15 (instead of 10) was used. The model of sediment hybrid juveniles, a n.minosbsinnode of 15 (instead of 10) was used. The purebred *D. trenchii* model was also changed from 15 to 10 for n.minosbsinnode. Overall, the number of trees to use (n.trees) was based on the recommended value from diagnostic plots produced from the plotres function ("predict n.trees"). Models were run on the High-Performance Computing node at the Australian Institute of Marine Science.

**Intrinsic resistance models**. Partial dependence functions (i.e., marginal effects) were calculated using the package "pdp"[80]. Non-linear models were fit to these predictive GBM outputs. To improve predictive accuracy, multiple non-linear trends were fit, including models incorporating 3–5 polynomial parameters. Polynomial regression functions were then calculated such the relationship between the independent variable ($x$ = percent (%) larval or juvenile survival) and the dependent variable ($y$ = environmental co-variates) modelled as an $n$th degree polynomial of $x$. The maximal level of survival and its corresponding environmental value were then calculated for each polynomial equation.

**Reporting summary**. Further information on research design is available in the Nature Research Reporting Summary linked to this article.

## Data availability

Data are available from K. Quigley's Github (https://github.com/LaserKate/AGF18_MachineLearning.git) with the following identifier: https://doi.org/10.5281/zenodo.6100726[81]

Amplicon sequencing data of the Symbiodiniaceae ITS2 region data is available on the Sequence Read Archive (BioProject PRJNA720058). Source Data are provided as a Source Data file. Source data are provided with this paper.

## Code availability

The code used in this current study is available from K. Quigley's Github: https://github.com/LaserKate/AGF18_MachineLearning.git with the following identifier: https://doi.org/10.5281/zenodo.6100726[81].

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

## Acknowledgements

We would like to thank the Traditional Owners from whose Sea-Country these colonies were collected, in particular the Lama Lama Traditional Owners from Far North Queensland. We would like to thank GBRLegacy, Ramaciotti Centre for Genomics at the University of New South Wales and Korak Saha at NOAA and Marc Hammerton at the Australian Institute of Marine Science with their assistance in re-formatting the CoR-TAD global satellite data and eReefs data. Laboratory animals were collected and cared for under institutional guidelines and permit number G18/41667.1. Funding was provided by the Australian Institute of Marine Science to KMQ. The development of SS01 was supported by funding from Paul G. Allen Family Foundation to MJHvO. We acknowledge the Australian Research Council Laureate Fellowship FL180100036 to MJHvO.

## Author contributions

The study was conceived by K.Q. Data was gathered, analyzed, and interpreted and the first draft was written by K.Q. Editing multiple versions and writing the final draft M.v.O. and K.Q.

## Competing interests

The authors declare no competing interests.
