## [Peer Review File · Nature Communications]

Reviewers' Comments:

Reviewer #1:

Remarks to the Author:

This paper represents a huge undertaking by the authors testing coral (*A. tenuis*) larval and recruit thermal tolerance from parental colonies that were collected from a wide range of latitudes across the GBR. The authors also included the effects of different algal symbionts on recruit thermal tolerance. They then mined satellite data to characterize thermal environments across the GBR and used these data to predict sites that might harbor thermal tolerant corals. It is clear that a massive amount of work went into this study and I commend the authors on these efforts. I will state outright that I do not feel qualified to address the modeling components of this work and hope that one of the other reviewers will be able to comment on this component. I will therefore focus my review on the thermal tolerance components and symbiont typing. Obviously it is up to the editor if they decide to move forward with this publication, however I hope that these recommendations can help the authors make an even stronger paper moving forward.

*Authorship: This study clearly involved a lot of moving parts and analyses. I was surprised there were only two authors. I am not making any assumptions, but I encourage careful reflection on who contributed to the success of the study.

The paper was generally well-written so I have instead focused on more broad issues.

Comments:

1. How do you account for mortality in control conditions? While this mortality is not that high in the larvae under control conditions, it looks equally high in recruits under both thermal conditions. It is not clear from the methods or supps how survival differences were calculated (cox-proportional hazards?) either. Overall I am really unclear how the experiment on recruit thermal tolerance answers these questions given the equally high mortality in the control conditions (except in those hosting C1).

2. Did you look at the correlation of larval thermal tolerance to recruit thermal tolerance? If not, this seems like a really important result either way! Does a cross who is more thermally tolerant as a larvae predict who is thermally tolerant as a recruit? This is especially important info for restoration and I would love to see this result incorporated in a revision. It could be that this analysis was already performed, however I did not find the information easily.

3. Data and code availability upon request is not an acceptable format. Please provide all data and code publicly to ensure transparency of all analysis and reproducibility.

4. Looking at the supplements it is not clear if there were culture replicates. How can you be sure that the response is not due to jar effects? At minimum this needs to be discussed.

5. Have the ITS2 data been put on SRA? There is no project number. The github page that is cited points to a different study. Please include the scripts and data used for this study.

6. Amplicon sequencing of the sediments- same comments here. Readers need to be able to see the code. Also F and I present is strange. Has this been seen before? I elaborate on this issue below.

7. Figure 1:

A. The colored text is too light on my version. Also the color scheme is not immediately obvious. Seems like one color scheme for "northern reefs" and another for "central reefs" makes the most sense here since this is how you made your crosses.

B. In the crossing design, I feel the like grey female color looks black- contrast not obvious on my version. Second, the color schemes of the larvae and recruits do not make sense. In the top left panel everything should be black (based on colors of male/female symbols), top right should be black to grey, bottom left should be grey to black and then fully grey. I think this could be fixed just by fixing the male/female symbols.

F. This figure is trying to showcase the DTR, but the 12 panels are not clear- is this lumping data

by every 2 hours? I am not immediately able to see the differences from this plot- most sites look the same, can you plot them all together? It is hard to compare them in this visualization.

D, G, H: I think these data would be more effective as violin plots (as others) or boxplots on the front of all data points? Single points do not tell the whole story. Also are any of these significantly different from each other?

8. Figure 2: It seems odd that the survival between the 27 and 32 was only really different for the corals infected with C1. How can this be the case? Seems like all treatments resulted in significant mortality in recruits and this seems like a major caveat to the study (This is the same comment as point 1). The grey colored larvae and recruit makes it seem like these plots are just corals from one region, consider changing color.

9. Figure 4: I find using the same color scheme as your sites very confusing in this figure.

10. Fig S2: I think something very strange is going on with the ITS2 data. Have you tried seeing if your ASVs that are mapping to the I and F are actually host contamination? We have seen this before when mapping to the GeoSymBio database, which I assume is what you're doing- although it is never explicitly stated (just referred to an old github). I recommend taking these data through SymPortal to see if you still see this I and F association. We have seen similarly strange data and when we reanalyzed in SymPortal these strange lineages disappear and they were actually associated with host contamination. An alternative is to include host sequences in the GeoSymBio database when assigning the ASVs.

11. Fig S2: I also find it strange that an ASV is shared between C1 and D1a. What does this mean? Contamination?

Reviewer #2:

Remarks to the Author:

This manuscript by Quigley and van Oppen look at how a coral from 2 different geographical locations and 5 sites, when crossed (Pure and hybrid) together with different symbiont species show differential tolerance and how using seawater temperature data in various ways and combining it with the data from physiology can be used to identify potential tolerant parent source in the natural environment. This is an interesting undertaking by the authors as to how selective breeding using stress tolerant parents might help in assisted sustenance of corals into the future. However, I cannot completely understand the logic behind using seawater temperature data to identify locations where corals are present and may or may not be exposed to high temperature and as a result may or may not be able to produce resistant offsprings as we cannot predict that coral physiology to stress is constant through time

While the main experimental work has no problem, there are many places in the text, I find difficult to understand. This is either due to misleading sentences or not so clear explanations or descriptions. Hence, I think that this manuscript needs to be revised for more clarity

Why authors use *Acropora tenuis* and not *A. millepora* or *A. spathulate* (Mol Ecol paper by same authors with almost similar experiment use *A. spathulata* instead) or coral from other genera? It is necessary to explain the selection of this particular species. And I cannot see this anywhere in the text. I feel that coral research has become biased towards very few coral species and generalizing the physiological attributes of those few coral species to all coral species in general. Hence, it is necessary to justify use of this particular species, not only for this piece of work, but also because same authors have not justified the use of this species in their previous publications as well as need to switch between 2-3 species of *Acropora*

Abstract

Line 31 – not at all true. In fact, there are many studies that have and are showing otherwise. It is just that our knowledge on the physiological tolerance mechanisms of every coral species is scarce.

Line 32 – sorry, this statement is little bold. Can reef corals by themselves transmit tolerance to high temperatures – to corals that are present in locations where they are not tolerant to high temperature, is this what you mean or the basis of this work?

Line 33 – 35 – again sorry, this statement is little bold and without strong support– If feel it is not wise to make a statement that selective breeding might help restoration efforts, when in this case we don't have any data on whether selective breeding to achieve assisted adaptation will work and also it has not been systematically tested.

Main Text

Line 49 – this is nothing but generalizing a complex system. It is not true, not all corals have narrow thermal ranges. If you want to be specific to some species or species groups, it is fine. Or only those coral species for which there are data on physiological tolerance, majority do show narrow thermal ranges.

Line 50 – also lower thermal tolerance limits. In past couple of years, winter bleaching has picked up in many locations

Line 56 – same as line 50 – authors need to consider that it is not just the problem of summer temperature anomalies anymore

Line 68-69 – this is not at all true. We are not able to and/or cannot find heat tolerant corals because coral reef researchers have focused on less than 20 coral species and most of the time less than 10 species to do repeated physiological tolerance studies. All the results on physiological experiments are skewed towards species that are, either dominant, easily accessible or just because they have been used by previous researchers.

Line 71- 72 – again this is a generalization. The previous study you cite, is based on limited number of crosses and moreover the coral species used in the study is *Acropora spathulata* and in this study it is *Acropora tenuis*, why not use one same species for every experiment so that everything is clarified at the end. What I mean is, if we keep switching species in every study, then there is no way we can get a complete picture on one species, this I say hoping authors are not assuming that all coral species have same or even similar physiological response. And there is no mention (in Material and Methods) of how many colonies from each location were used in the crossing experiments.

Also, the authors did not try this on multiple species across different genera. So, this does not necessarily reflect upon the response of adult and offspring performance in all coral species.

Line 100 -102 – I think that unless authors can show that all corals or all *Acropora* species have same physiological response, it is not fair to assume so. In Reference 2 (Material and Methods), the coral used is *Acropora millepora* and in Reference 3 it is *A. spathulata*. I don't agree basing the success of selective breeding or other assisted methods based on just one or two coral species. As you obscurely mention in discussion about this aspect (line 2430, which I think is hidden from the readers, especially journalists and normal people, and this has every possibility of misinterpreting your actual message.

Comments on Methods (submitted as supplementary information)

While there are many things explained in detail, I have difficulty in understanding some aspects. What exactly are floating net-wells? You refer to Meyer et al 2009, but there is no mention of floating net-wells in their paper.

Since you have a very complicated experimental design, it needs to be explained efficiently, because if not, only you can repeat this experiment on other coral species and no one else can. Figure 1B is very simplistic and not at all helpful to understand your experiment.

Why there is no statistical analysis on any of your data?

You use cultured Symbiodiniaceae in your experiments, you need to justify their efficacy since those in long-time culture are no more similar to those in the natural environment or in symbiosis in terms of their physiology.

Also, see "Limitations of Using Cultured Algae to Study Cnidarian-Algal Symbioses and Suggestions for Future Studies" by Maruyama and Weis 2020 - <https://onlinelibrary.wiley.com/doi/10.1111/jpy.13102>

Line 103 -109 – This seems, but obvious random effects at any given point of time. Corals from warm locations do better sometimes and those from not so warm do better some times, and random crosses have better response sometimes. So, if you repeat this experiment, will you see the same cross response as you see here, or you think it might change...

Line 113-117 – I don't understand this, how come you say that lab-evolved strain is better? From figure 1C the survival and moreover their response is not on par with other Symbiodiniaceae. I see a significant difference in the response and as usual *Durusdinium trenchii* fares better. I would like to see statistical analysis done on your data, at least simple statistics to show if there is any significant difference within and between treatments across your crosses as well as juveniles and larvae

And the, in your supplementary file (predictive models) you say "the overall survival of the C1 treatment at elevated temperature was lower than other treatments (Fig. 2). SS and heat-tolerant D1a provided an almost equivalent heat protection to juveniles at 32°C although the explanatory power of the..."

Line 132 onwards – I don't understand this. Is this right? – you have different Symbiodiniaceae composition in your control and high temperature treatments? If so, why? And if not, then please excuse me, I cannot understand. Maybe you can describe it in a way it can be understood easily?

Line 162 – This is outdated and based on the response of 1 coral species. Corals do tolerate more than 1-1.5 °C – for example, *Coelastrea aspera*, corals in the Persian and Arabian Gulf and many other places...I think you use that the study by Berkelmans et al just to make your point here about selective breeding mechanism.

Line 251 – so how will you find heat-tolerant colonies in the natural environment? I am not convinced from this work that you can find them. Also, I am bit confused. Is there any result or did you do any work on stress resistance or sensitivity of aposymbiotic larvae and juveniles? Because how do you know if the tolerance you see in your experiment is because of tolerant parents or just because of the experiment effect using cultured symbionts? Coral response, mainly to temperature stress, is relative to time and space. It also depends on acquired thermal breadths over time and on the thermal thresholds of individual coral species. Moreover, as authors mention in lines 279-284, there are so many other factors that might influence how a coral colony will react to its environment. So, a coral colony in a particular location may be not bleach this year but might bleach next year. Or once considered temperature stress resistant *Porites lutea*, is not so much resistant anymore as seen by its reported bleaching from various locations in past couple of years.

Even in Plaumbi et al, they discuss "Thus, acclimatization alone cannot be expected to completely overcome the threat to corals from widespread bleaching events, especially if the onset of high-temperature stress is abrupt and sustained. In this regard, the tempo and severity of heat anomalies will be critical for effective coral acclimatization" in addition to that the microbiome and other factors

What I want to say is, we cannot be sure that to identify potential parents across latitudes that

might help us make resistant juveniles in the laboratory.

Line 255 onwards – I don't understand this. You used mainly cultured symbionts and one modified symbiont to make stress resistant combinations. How this same as presence of heat tolerant adults in the natural environment? Figure 4 title says "Intrinsic resistance models identify hundreds of potential reefs as predicted sources of heat-tolerant adult corals able to produce offspring of optimal survival under heat stress", this itself is contradictory to this work, because there is little chance for those adults in the natural environment to cross with each other – for example parents from north with those in central GBR. did Dixon et al and Palumbi et al conduct their experiment to search for heat-tolerant corals? Dixon et al, similar to this work, perform crosses using parents from different latitudes to show again, that it is possible to get tolerant juveniles by crossing parents, and Palumbi et al just look at how corals between warm and normal pool respond to temperature stress in one location in American Samoa.

Line 290-292 – I am not convinced that lab-evolved strain confers increased survival to juvenile corals (see my previous comment). Need more statistical analysis and clear data presentation. Also as mentioned before, you need to discuss about using cultured or modified symbiont in terms of their physiology and long-term association with corals

Reviewer #3:

Remarks to the Author:

Review of NCOMMS #20-48044-T

Overall

This is an excellent, ground-breaking manuscript and I recommend publication following a minor revision. The authors report out on a monumental effort to increase heat tolerance via selective breeding/novel host-symbiont associations. The authors use these results to model where corals may be sourced that have inherent heat tolerance, as well as the environmental conditions of reefs that correlate with elevated heat tolerance. This is incredibly important and I hope to see it published soon.

The authors might consider changing their title (see comments below). I think there are plenty of examples of heat tolerant corals that aren't from low latitudes.

Also, the discussion seems to run out of steam and go on too long. Please tighten it up and organize it better.

Specific

1)Line 31. I don't think it's a common assumption that only low latitude corals are heat tolerant. Reword. Maybe say it is assumed low latitude corals are more heat-tolerant or something along those lines. I just think that anyone who knows anything about coral heat tolerance does not assume that only low latitude corals are heat tolerant. I'd also suggest changing the title so as not to dwell on the low latitude aspect of the study. I think its worth discussing, but I don't find that aspect of the work the most important.

2)Reference 2 is not a good reference to describe bleaching.

3)Reference 5 reports on bleaching in Singapore, not Palau

4)Line 82. Change to 23-33oC

5)Line 93: Insert "of" in between mixture and free-living

6)Line 188: Start new paragraph here

Reviewer #4:

Remarks to the Author:

I really like what Quigley and van Oppen are trying to do here. My problem is that I can't really assess the validity of the conclusions because I cannot work out some important sample size

information. How many parent colonies were collected from each reef, and how many of those (at each reef) provided sperm or eggs or both? This is important for working out how good is the evidence that the reef effects identified are actually effects of the reef, or whether they may reflect vagaries of which genotype was sampled. The follow up to this question is then how was this incorporated into the analysis? Say reef A had corals A1 and A2, and reef B had corals B1 and B2. If A1 x B1 and A2 x B2, then all the larvae and juveniles come from two sets of parents. So is each set of parents treated as a random effect nested within reef, and then all the corresponding juveniles or larvae are the individual observations nested within the parent? If not, how do the authors know they are not quantifying parent affects that may or may not reflect what holds generally on the reef that the parent is from?

A similar argument applies to the boosted regression tree analysis. With only 5 reefs, is it really possible to distinguish the predictive power of so many environmental variables? Along the same lines, it seems like the only way to cross-validate with 10 folds would be to treat each observation of each larva or juvenile as statistically independent. Again, I would want to be reassured that the nested structure of the data is accounted for in the analysis. It is hard for me to imagine how data from 5 reefs could be used to generalize to the whole Great Barrier Reef in a regression model with so many variables. I could be wrong, but I would want to know how the authors' approach to the analysis accounts for these issues.

REVIEWER COMMENTS -RESPONSES

Title Predictive models for the selection of thermally tolerant corals

Authors K. M. Quigley^{1*}, M. J. H. van Oppen^{1,2}

Reviewer #1 (Remarks to the Author):

Review-Comment	Comment by Reviewer	Response
1.1	This paper represents a huge undertaking by the authors testing coral (A. tenuis) larval and recruit thermal tolerance from parental colonies that were collected from a wide range of latitudes across the GBR. The authors also included the effects of different algal symbionts on recruit thermal tolerance. They then mined satellite data to characterize thermal environments across the GBR and used these data to predict sites that might harbor thermal tolerant corals. It is clear that a massive amount of work went into this study and I commend the authors on these efforts. I will state outright that I do not feel qualified to address the modeling components of this work and hope that one of the other reviewers will be able to comment on this component. I will therefore focus my review on the thermal tolerance components and symbiont typing. Obviously it is up to the editor if they decide to move forward with this publication, however I hope that these recommendations can help the authors make an even stronger paper moving forward.	We appreciate the recognition by Reviewer 1 for the large amount of experimental and modelling work presented in this paper. We thank Reviewer 1 for their detailed comments on the thermal tolerance components and symbiont typing, it has significantly improved the quality and rigour of the work.
1.2	*Authorship: This study clearly involved a lot of moving parts and analyses. I was surprised there were only two authors. I am not making any assumptions, but I encourage careful reflection on who contributed to the success of the study.	We acknowledge that this is a large body of work. Other people involved in the non-modelling components of this work are co-authors on a complementary publication that has recently been released (Quigley et al. 2021; doi: 10.3389/fmars.2021.636177) which used a subset of these crosses in a field experiment. These people are also cited in the Acknowledgement section of this current paper.
1.3	The paper was generally well-written so I have instead focused on more broad issues.	We thank Reviewer 1 for their broad-scale comments and for their belief that the paper is generally well-written.

1.4	1. A. How do you account for mortality in control conditions? While this mortality is not that high in the larvae under control conditions, it looks equally high in recruits under both thermal conditions. B. It is not clear from the methods or supps how survival differences were calculated (cox-proportional hazards?) either. Overall I am really unclear how the experiment on recruit thermal tolerance answers these questions given the equally high mortality in the control conditions (except in those hosting C1).	A. Survival was not significantly different between control and hot treatments between juvenile families. This means that, at the family level, there is no effect of heat in the mortality data. There was only a significant difference in mortality between heat vs ambient treatments when averaged across all juvenile families infected with C1 (Figure 2c). Further information, statistical tests, and Figures have been added to the Supplementary Material regarding this and the mortality experienced in the control treatment of the juvenile experiment. Supplementary, Lines 7-23: “Juvenile survival. Survival at ambient and elevated temperatures varied both within and between the 25 crosses of larvae (Fig. 2a, Supplementary Table 1, Supplementary Fig. 2) and juveniles infected with the four Symbiodiniaceae treatments (Fig. 2b, Supplementary Fig. 3). Specifically, within each of the families and treatments, there was no significant difference in survival between the two temperature treatments for juveniles placed in the sediment treatment (Supplementary Fig. 3; Wilcoxon test $p = 0.15 - 1$), the D1a treatment ($p = 0.33 - 1$), SS1 ($p = 0.0059-1$), or the C1 treatment ($p = 0.081 - 1$). The lack of significant differences in survival was likely due to high variability in this metric at both temperatures. Elevated mortality of juveniles, even at “control” conditions is common in corals and other organisms with mass-spawning reproductive strategies. For example, there is a well-documented drop in survival in the first few months of juvenile life, generally accounting for around >30-99% mortality (reviewed in ¹). Hence, we propose that this mortality at control temperatures is expected, especially given the long timeframe of this experiment (58 days) covering this early window of juvenile life. Survival at both temperatures were incorporated into the Boosted Regression Tree models, allowing the machine learning algorithm to learn from patterns in survival of both control and elevated temperature datasets and thereby incorporates this variability.” B. Further information has been added to the Methods to explain how survival was calculated per time-point for larvae: Methods, Lines 392-393: “Survival was quantified as the average percentage of surviving larvae across each of the replicate wells as compared to the initial number of larvae added per well (Supplementary Table 3).” For juvenile survival, which was based on individual data, survival was calculated using Kaplan-Meier. This additional information has been added to the Methods: Methods, Lines 433-439:
------------	---	--

		“Survival statistical analyses. Replicate, single juveniles per cross were tracked through time using photographs, allowing for individual-based survival analysis to be undertaken using the survfit function from the “survival” package ⁶⁵. This was performed using Kaplan-Meier methods; with symbiont type, temperature, tank replicates, cross, maternal and paternal identities accounted for as fixed effects. Percentage survival and standard error for replicate juveniles were calculated per treatment. Replication included replicate individual juveniles, within replicate wells, replicate plates, within replicate tanks per symbiont by each temperature combination (Supplementary Table 4).”
1.5	2. Did you look at the correlation of larval thermal tolerance to recruit thermal tolerance? If not, this seems like a really important result either way! Does a cross who is more thermally tolerant as a larvae predict who is thermally tolerant as a recruit? This is especially important info for restoration and I would love to see this result incorporated in a revision. It could be that this analysis was already performed, however I did not find the information easily.	This additional analysis was performed and has now been added to the Supplementary. We did not add to the main results for the following reason: Supplementary, Lines 33-36: “It should be noted that these life-stages were measured for different lengths of time, under different symbiont states (symbiotic and aposymbiotic) and at different elevated temperature treatments, so comparisons as such should be taken with caution.” Supplementary, Lines 31-45: “Comparison of survival between larvae and juveniles. To assess if more thermally tolerant larvae grew into more thermally tolerant juveniles, survival at both life-stages was compared at the final timepoint for each life-stage. It should be noted that these life-stages were measured for different lengths of time, under different symbiont states (symbiotic and aposymbiotic) and at different elevated temperature treatments, so comparisons as such should be taken with caution. At the control temperature (27°C), only three crosses differed significantly in survival, in which BKxBK, CUxBK, LSxBK larvae survived better than their juvenile counterparts ($p = 0.00042 - 0.047$; Supplementary Fig. 4). In the elevated temperature treatments (35.5 or 32°C for larvae and juveniles, respectively), most of the crosses survived better as juveniles compared to as larvae (~50% of crosses, potentially due to lower heat), including juveniles from crosses CUxDR, CUxLS, CUxSB ($p = 0.011 - 0.0088$), DRxBK, DRxDR, DRxSB ($p = 0.00087 - 0.0024$), LSxCU, LSxLS ($p = 0.00092 - 0.001$), and SBxLS and SBxSB ($p = 0.0014 - 0.011$). Only 15% of cross comparisons resulted in larvae that survived better compared to juveniles, including BKxCU, BKxSB, CUxBk ($p = 0.00062 - 0.023$).”
1.6	3. Data and code availability upon request is not an acceptable format. Please provide all data and code publicly to ensure transparency of all analysis and reproducibility.	These resources are now available. The following statements have been added as well. The code for the Symbiodiniaceae analysis was already available via referenced published work in Quigley et al. 2019 Scientific Reports (https://www.nature.com/articles/s41598-019-50045-y)

		Data availability Data are available from K. Quigley’s Github (https://github.com/LaserKate/AGF18_MachineLearning.git). Amplicon sequencing data of the Symbiodiniaceae ITS2 region data is available on the Sequence Read Archive (BioProject PRJNA720058). Code availability The code used in this current study is available from K. Quigley’s Github: https://github.com/LaserKate/AGF18_MachineLearning.git
1.7	4. Looking at the supplements it is not clear if there were culture replicates. How can you be sure that the response is not due to jar effects? At minimum this needs to be discussed.	The level of replication has been made clearer in the Methods for both life-stages: Methods, Lines 383-388: “Aposymbiotic larval heat stress was assessed using floating net-wells⁶². Net-wells are light-weight plastic containers with one end open (top) and closed by a fine micron-mesh (bottom), originally used to strain cells. Individual, separated net-wells are placed into floating holders such that the well, and therefore larvae contained with it, were suspended below the surface of the water. Four days post spawning and fertilization, 20 larvae were aliquoted into floating net-wells into separated well replicates (n = minimum of six replicate wells) for each population cross.” Methods, Lines 396-398: “Coral larvae from a subset of crosses were settled into three wells each (n = three replicates per cross) within sterile six-well plates using autoclaved CCA added to each well.”
1.8	5. Have the ITS2 data been put on SRA? There is no project number. The github page that is cited points to a different study. Please include the scripts and data used for this study.	These resources are now available. The following statements have been added as well. Data availability Data are available from K. Quigley’s Github (https://github.com/LaserKate/AGF18_MachineLearning.git). Amplicon sequencing data of the Symbiodiniaceae ITS2 region data is available on the Sequence Read Archive (BioProject PRJNA720058). Code availability The code used in this current study is available from K. Quigley’s Github: https://github.com/LaserKate/AGF18_MachineLearning.git
1.9	6. A. Amplicon sequencing of the sediments- same comments here. Readers need to be able to see the code.	A. Please see comment/response and 1.6/1.8. All code and data were available and is now clearer. B. Further information on F and I have been added to the Main and Supplementary text:

	B. Also F and I present is strange. Has this been seen before? I elaborate on this issue below.	Main, Lines 154-157: “Symbiont communities in juveniles at the control temperature were predominantly made up of taxa from Cladocopium and Fugacium, whereas juveniles exposed to heat were dominated by these taxa in addition to the more cryptic “I” clade (see further discussion of Fugacium and “I” in Supplementary).” Supplementary, Lines 137-150: “After variance normalization, symbiont communities within the control treatment were generally dominated by Cladocopium and Fugacium, whilst the juveniles sequenced from the elevated temperature treatment had more individual juveniles with Durusdinium and the hitherto uncharacterized clade “I” (Supplementary Fig. 1). A high diversity of Symbiodiniaceae taxa in coral juveniles has been previously reported, including the occurrence of Fugacium and “I”, recovered from juvenile samples in the wild ³. Moreover, the occurrence of these taxa has been detected through the use of two different pipelines (OTU variant calling via USEARCH clustering ⁴, adapted for Symbiodiniaceae specifically ⁵ using custom databases) as well as with ASV calling (DADA2 ⁶, adapted for Symbiodiniaceae ⁷). RNAseq of n = 43 different juvenile samples confirmed the mixed assemblage of symbionts from these genera (Quigley and Strader, in-review). Therefore, given the detection of these taxa in coral juveniles of this species previously from other sources and using other methods, we suggest that these ASVs do not represent spurious variants or host contamination.”
1.10	7. Figure 1: A. The colored text is too light on my version. Also the color scheme is not immediately obvious. Seems like one color scheme for “northern reefs” and another for “central reefs” makes the most sense here since this is how you made your crosses. B. In the crossing design, I feel the like grey female color looks black- contrast not obvious on my version. Second, the color schemes of the larvae and recruits do not make sense. In the top left panel everything should be black (based on colors of male/female symbols), top right should be black to grey, bottom left should be grey to black and then fully grey. I think this could be fixed just by fixing the male/female symbols. F. This figure is trying to showcase the DTR, but the 12 panels are not clear- is this lumping data by every 2 hours? I am not immediately able to see the	A. This has been changed. It should also be noted that this was also raised by Reviewer 2, in comment 14. The coloured text has now been changed to black text to aid in contrast. This colour scheme has been further explained in the Figure legend: “The most northern reefs are shown in “warm” colours (red/orange) and the central reefs are in “cooler” colours (black/blue).” B. This has been changed to be described as “dark grey” and “white” now. Further labels have been added to also improve clarity. “Breeding design used to create purebred and crossbred hybrid offspring (larvae and settled juveniles) encompassing various genetic backgrounds (b). Purebred (purebred northern: dark grey, purebred central: white) and crossbred offspring (northern mom x central dad: dark grey to white gradation, central mom x northern dad: white to dark grey gradation) were produced.”

	differences from this plot- most sites look the same, can you plot them all together? It is hard to compare them in this visualization. D, G, H: I think these data would be more effective as violin plots (as others) or boxplots on the front of all data points? Single points do not tell the whole story. Also are any of these significantly different from each other?	Further, the figure has been edited so that “dark grey” and “white” have been made easier to distinguish, including modifying the colours as suggested (in central purebreds for example). F. This has been changed. All reefs have now been plotted together as suggested instead of faceted separately. D,G,H: This has been changed. These three figures have been changed to boxplots as suggested. These statistical tests have been added to the Supplementary and summarized as asterisks on Figure 1. Supplementary Lines 54 – 68: “Non-parametric Wilcoxon tests were used to determine if there were statistically significant differences in the mean value of predictors (Fig. 1c-h) across each reef (pairwise comparisons, $p < 0.05$ denoted by asterisks in Fig. 1c-h). SST_av.pre varied significantly between Curd and Sand Bank 7 (Fig. 1c; $p = 0.049$), Davies ($p = 3.6e-7$), and Backnumbers ($p = 1.5e-5$). SST_av.pre did not vary between Long Sandy and Curd or Sand Bank 7 ($p = 0.39$ and 0.17) but it did vary with Davies and Backnumbers ($p = 0.001$ and $7.4e-5$). Sand Bank 7 also varied with Davies and Backnumbers ($p = 0.0027$ and 0.023). Davies and Backnumbers did not differ significantly in SST_av.pre ($p = 0.34$). SST_stdev did not differ significantly across any reef comparisons (Fig. 1d; non-parametric Wilcoxon test all $p = 0.38 - 1$). SST_av.post differed significantly between Curd reef and Long Sandy ($p = 0.02$), Sand Bank 7, ($p = 0.031$), Davies ($p = 0.0051$) and Backnumbers ($p = 0.02$), but not between any of the other reef combinations (Fig. 1e). SSTA_Freq_stdev differed significantly between Backnumbers and Curd ($p = 0.038$), Davies ($p = 8e-4$), Sand Bank 7 ($p = 0.055$), Long Sandy ($p = 0.032$) and Davies compared to Long Sandy ($p = 0.051$, Fig. 1g). TSA_DHW_stdev did not vary significantly between any reef combinations ($p = 0.087 - 1$, Fig. 1h).”
1.11	8. a. Figure 2: It seems odd that the survival between the 27 and 32 was only really different for the corals infected with C1. How can this be the case? Seems like all treatments resulted in significant mortality in recruits and this seems like a major caveat to the study (This is the same comment as point 1). b. The grey colored larvae and recruit makes it seem like these plots are just corals from one region, consider changing color.	a. There was substantial variability in survival at both temperatures at the family level, likely due to low sample size and high mortality normal at this life-stage regardless of treatment. Moreover, survival was not significantly different between control and hot treatments between juvenile families. A figure and statistical analysis and results have been added into the Supplementary: Supplementary Lines 7-23: “Juvenile survival. Survival at ambient and elevated temperatures varied both within and between the 25 crosses of larvae (Fig. 2a, Supplementary Table 1, Supplementary Fig. 2) and juveniles

		infected with the four Symbiodiniaceae treatments (Fig. 2b, Supplementary Fig. 3). Specifically, within each of the families and treatments, there was no significant difference in survival between the two temperature treatments for juveniles placed in the sediment treatment (Supplementary Fig. 3; Wilcoxon test $p = 0.15 - 1$), the D1a treatment ($p = 0.33 - 1$), SS1 ($p = 0.0059-1$), or the C1 treatment ($p = 0.081 - 1$). The lack of significant differences in survival was likely due to high variability in this metric at both temperatures. Elevated mortality of juveniles, even at “control” conditions is common in corals and other organisms with mass-spawning reproductive strategies. For example, there is a well-documented drop in survival in the first few months of juvenile life, generally accounting for around >30-99% mortality (reviewed in ¹). Hence, we propose that this mortality at control temperatures is expected, especially given the long timeframe of this experiment (58 days) covering this early window of juvenile life. Survival at both temperatures were incorporated into the Boosted Regression Tree models, allowing the machine learning algorithm to learn from patterns in survival of both control and elevated temperature datasets and thereby incorporates this variability.” b. The grey coloured larvae and recruit have been deleted.
1.12	9. Figure 4: I find using the same color scheme as your sites very confusing in this figure.	This has been changed. The colour scheme has been changed by increasing the “transparency” of the colour scheme for this figure. Setting “transparency” much lower differentiates the colour scheme from the other figures (which I believe is what Reviewer 1 wants). This was to avoid any confusion that they signify the same metrics as the other figures. It also allows to keep for consistency across figures and makes it easier to identify the reef polygons below the points.
1.13	10. Fig S2: I think something very strange is going on with the ITS2 data. Have you tried seeing if your ASVs that are mapping to the I and F are actually host contamination? We have seen this before when mapping to the GeoSymBio database, which I assume is what you’re doing- although it is never explicitly stated (just referred to an old github). I recommend taking these data through SymPortal to see if you still see this I and F association. We have seen similarly strange data and when we reanalyzed in SymPortal these strange lineages disappear and they were actually associated with host contamination. An alternative is to	Further novel sequencing and analysis has been done to confirm symbiont community composition (see Comment-Response 1.14). Further justification for these communities has also been added: These taxa have previously been documented in Acropora tenuis juvenile exposed to Symbiodiniaceae in the wild (e.g. Quigley et al. 2017). During the publication processes, our DADA2 symbiont pipeline was reviewed by Ben Hume, using his SymPortal pipeline, and was found to be equivalent. Further information has been added to the Supplementary. Specifically: Supplementary Lines 137-150:

	include host sequences in the GeoSymbio database when assigning the ASVs.	“After variance normalization, symbiont communities within the control treatment were generally dominated by Cladocopium and Fugacium , whilst the juveniles sequenced from the elevated temperature treatment had more individual juveniles with Durisdinium and the hitherto uncharacterized clade “I” (Supplementary Fig. 1). A high diversity of Symbiodiniaceae taxa in coral juveniles has been previously reported, including the occurrence of Fugacium and “I”, recovered from juvenile samples in the wild ³ . Moreover, the occurrence of these taxa has been detected through the use of two different pipelines (OTU variant calling via USEARCH clustering ⁴ , adapted for Symbiodiniaceae specifically ⁵ using custom databases) as well as with ASV calling (DADA2 ⁶ , adapted for Symbiodiniaceae ⁷). RNAseq of n = 43 different juvenile samples confirmed the mixed assemblage of symbionts from these genera (Quigley and Strader, in-review). Therefore, given the detection of these taxa in coral juveniles of this species previously from other sources and using other methods, we suggest that these ASVs do not represent spurious variants or host contamination.”
1.14	11. Fig S2: I also find it strange that an ASV is shared between C1 and D1a. What does this mean? Contamination?	Additional independent sequencing of different individuals has been done using RNAseq to confirm the dominant symbionts per treatment. This information has been added to the Methods and Results: Method Lines 464 - 467: “As a secondary confirmation of the taxonomic identities of symbionts in each treatment, additional biological replicates of coral juveniles per treatment (n= 29 (C. goreau), 38 (D. trenchii), 43 (SED), 36 (SS1)) were sequenced using RNAseq and Symbiodiniaceae transcriptomes were mapped to these reads using standard tag-based methods ⁷⁰ .” Result Lines 92-94: “Establishment of symbiosis with the symbiont strain in the inoculum was confirmed via high-throughput ITS2 Illumina sequencing and dominant communities confirmed with RNAseq mapping of Symbiodiniaceae transcriptomes (Supplementary Fig. 1).”

Reviewer #2 (Remarks to the Author):

2.1	This manuscript by Quigley and van Oppen look at how a coral from 2 different geographical locations and 5 sites, when crossed (Pure and hybrid) together with different symbiont species show differential tolerance and how using seawater temperature data in various ways and combining it with the	We are very pleased to see that Reviewer 2 thinks that the combination of physiological experiments, satellite data, and modelling is an “interesting undertaking.” To further improve the understanding on how seawater temperature data can be used to identify locations where corals are presented, we have added an addition section further justifying our study goals:
-----	---	--

	data from physiology can be used to identify potential tolerant parent source in the natural environment. This is an interesting undertaking by the authors as to how selective breeding using stress tolerant parents might help in assisted sustenance of corals into the future. However, I cannot completely understand the logic behind using seawater temperature data to identify locations where corals are present and may or may not be exposed to high temperature and as a result may or may not be able to produce resistant offspring as we cannot predict that coral physiology to stress is constant through time	Result Lines 285 - 292: “Finally, the multitude of factors influencing the heritability of heat tolerance makes ascribing drivers of tolerance challenging⁵³. For example, holobiont tolerance is influenced by the algal and bacterial symbionts, and the acclimatory history and potential of the host and symbionts (reviewed in⁵⁴). However, the operationalization of increasing heat tolerance via genetic adaptation in at least some coral species is possible given the strong heritability of this trait (for theory see¹⁷, for quantification in corals see^{16,55}). Given the current difficulty in finding heat tolerant individuals, our method represents a step-change improvement in detection.”
2.2	While the main experimental work has no problem, there are many places in the text, I find difficult to understand. This is either due to misleading sentences or not so clear explanations or descriptions. Hence, I think that this manuscript needs to be revised for more clarity	We are thankful to read that Reviewer 2 thinks “the main experimental work has no problem.” The helpful comments from the four Reviewers have significantly improved the clarity of the text throughout this manuscript. We refer the Reviewer to the track-changed and finalized version of this manuscript for the full scope of edits made during this review process.
2.3	Why authors use Acropora tenuis and not A. millepora or A. spathulate (Mol Ecol paper by same authors with almost similar experiment use A. spathulata instead) or coral from other genera? It is necessary to explain the selection of this particular species. And I cannot see this anywhere in the text. I feel that coral research has become biased towards very few coral species and generalizing the physiological attributes of those few coral species to all coral species in general. Hence, it is necessary to justify use of this particular species, not only for this piece of work, but also because same authors have not justified the use of this species in their previous publications as well as need to switch between 2-3 species of Acropora	This justification has been added to the Supplementary. Methods Lines 341-344: “This species was chosen, compared to other species previously used for selective breeding (i.e. Acropora spathulata¹⁶), due to their higher propensity for successful settlement to known cues⁵⁸, which is essential for large-scale juvenile experimentation (see below for information on larval settlement).”
2.4 Abstract	Line 31 – not at all true. In fact, there are many studies that have and are showing otherwise. It is just that our knowledge on the physiological tolerance mechanisms of every coral	This sentence has been changed and this point has been acknowledged further down: Abstract 17 -19:

	species is scarce.	“Finding coral reefs resilient to climate warming is challenging given the large spatial scale of reef ecosystems.” Introduction 45-49: “Although it is assumed that low latitude corals are more heat-tolerant, resilient reefs of tolerance are increasingly being found. Signatures of acclimation and adaptation have been detected globally, including in Singapore ⁶, the Great Barrier Reef (GBR) ⁷ and the Persian Gulf ⁸. In some of these locations, acquired heat tolerance has since been shown across diverse species (e.g. Coelastrea aspera ⁹) and regions (e.g. the Persian Gulf ¹⁰).” Reviewer 3 also commented on this sentence. This sentence has been changed per Reviewer 3’s suggestion in Abstract Lines 18-19: “Methods are needed to predict the location of corals with heritable tolerance to high temperatures.”
2.5	Line 32 – sorry, this statement is little bold. Can reef corals by themselves transmit tolerance to high temperatures – to corals that are present in locations where they are not tolerant to high temperature, is this what you mean or the basis of this work?	This sentence has been changed as suggested to be clearer about the goals of this work: The original sentence was: “Methods to predict the geographic location of reef corals that can transmit tolerance to high temperatures and then estimate the value of intraspecific hybridisation to breed heat tolerant corals could direct long-term protection and selective breeding for restoration efforts, but such approaches are currently non-existent.” The edited sentence is: “Methods are needed to predict the location of corals with heritable tolerance to high temperatures.”
2.6	Line 33 – 35 – again sorry, this statement is little bold and without strong support– If feel it is not wise to make a statement that selective breeding might help restoration efforts, when in this case we don’t have any data on whether selective breeding to achieve assisted adaptation will work and also it has not been systematically tested.	This specific part of the sentence has been deleted as suggested by Reviewer 2: The original sentence was: “Methods to predict the geographic location of reef corals that can transmit tolerance to high temperatures and then estimate the value of intraspecific hybridisation to breed heat tolerant corals could direct long-term protection and selective breeding for restoration efforts, but such approaches are currently non-existent.” The edited sentence is: “Methods are needed to predict the location of corals with heritable tolerance to high temperatures.”
2.7 Main Text	Line 49 – this is nothing but generalizing a complex system. It is not true, not all corals have narrow thermal	This has been reworded: Introduction Line 32-33:

	ranges. If you want to be specific to some species or species groups, it is fine. Or only those coral species for which there are data on physiological tolerance, majority do show narrow thermal ranges.	“Like many ectotherms ¹, many of the coral species for which physiological tolerance is known have narrow thermal ranges.”
2.8	Line 50 – also lower thermal tolerance limits. In past couple of years, winter bleaching has picked up in many locations	Very good point. This has been added: Introduction Lines 34 -37: “When temperatures surpass the coral’s upper or lower thermal tolerance limits, the endosymbiotic relationship between the coral host and its photosynthetic dinoflagellates (Family Symbiodiniaceae) breaks down and a large percentage of Symbiodiniaceae cells are lost from the host tissues, a process known as bleaching ².”
2.9	Line 56 – same as line 50 – authors need to consider that it is not just the problem of summer temperature anomalies anymore	This has been added: Introduction 41-43: “The occurrence and severity of mass coral bleaching is strongly linked to the magnitude and duration of summer temperature anomalies, although smaller scale bleaching in winter months may also occur (Hoegh-Guldberg and Fine 2004).”
2.10	Line 68-69 – this is not at all true. We are not able to and/or cannot find heat tolerant corals because coral reef researchers have focused on less than 20 coral species and most of the time less than 10 species to do repeated physiological tolerance studies. All the results on physiological experiments are skewed towards species that are, either dominant, easily accessible or just because they have been used by previous researchers.	This generalization and added: Introduction Lines 55- 57: “Genetic variation in some populations of the coral host is associated with tolerance to higher temperatures that persists from the adult into the larval ¹⁵ and juvenile ¹⁶ phase in a few studies of a limited number of coral species.” Introduction Lines 62- 64: “It is important to note that assessments of genetic variation and comparisons between early life-history and adult responses have only been undertaken in a small subset of coral species and findings from these studies may not be generalizable across species and genera.”
2.11	Line 71- 72 – a. again this is a generalization. The previous study you cite, is based on limited number of crosses and b. moreover the coral species used in the study is Acropora spathulata and in this study it is Acropora tenuis, why not use one same species for every experiment so that everything is clarified at the end. What I mean is, if we keep switching	a. and d. This generalization has been qualified as above: Introduction Lines 55- 57: “Genetic variation in some populations of the coral host is associated with tolerance to higher temperatures that persists from the adult into the larval ¹⁵ and juvenile ¹⁶ phase in a few studies of a limited number of coral species.” Introduction Lines 62- 64:

	species in every study, then there is no way we can get a complete picture on one species, this I say hoping authors are not assuming that all coral species have same or even similar physiological response. c. And there is no mention (in Material and Methods) of how many colonies from each location were used in the crossing experiments. d. Also, the authors did not try this on multiple species across different genera. So, this does not necessarily reflect upon the response of adult and offspring performance in all coral species.	“It is important to note that assessments of genetic variation and comparisons between early life-history and adult responses have only been undertaken in a small subset of coral species and findings from these studies may not be generalizable across species and genera.” b. Please see Comment-Response 2.3 concerning justification on why A. tenuis and not A. spathulata was used in this current study. c. This information and an additional Supplementary Table has been added to the Methods Lines 337-340: “Ten gravid colonies per reef were collected from Long Sandy, Curd, Sand Bank 7, and Davies, and 15 gravid colonies from Backnumbers. Not all colonies per population spawned, and the total number of maternal and paternal colonies used per cross is listed in Supplementary Table 1.”
2.12	Line 100 -102 – a. I think that unless authors can show that all corals or all Acropora species have same physiological response, it is not fair to assume so. In Reference 2 (Material and Methods), the coral used is Acropora millepora and in Reference 3 it is A. spathulata. I don’t agree basing the success of selective breeding or other assisted methods based on just one or two coral species. b. As you obscurely mention in discussion about this aspect (line 2430, which I think is hidden from the readers, especially journalists and normal people, and this has every possibility of misinterpreting your actual message.	This has been further qualified: a. Lines 96 – 98: “The influence of these symbionts in driving coral temperature tolerance in select coral species like A. millepora and A. spathulata, is unequivocal for traits such as bleaching¹⁸ and survival^{16,19,20} and is further quantified here.” b. Lines 326 – 328: This sentence (“One key limitation of this work is its reliance on data produced from a single coral species, Acropora tenuis.”) has been re-iterated in the Conclusion so as not to be misconstrued as hidden. “Further work is needed across more coral species from a range of divergent lineages (e.g. Goniastrea, Isopora) to assess if the conclusions drawn here for Acropora tenuis can be generalized.”
2.13 Comments on Methods (submitted as supplementary information)	While there are many things explained in detail, I have difficulty in understanding some aspects. What exactly are floating net-wells? You refer to Meyer et al 2009, but there is no mention of floating net-wells in their paper.	This has been elaborated on in the Methods: Lines 382-388: “Aposymbiotic larval heat stress was assessed using floating net-wells⁶². Net-wells are light-weight plastic containers with one end open (top) and closed by a fine micron-mesh (bottom), originally used to strain cells. Individual, separated net-wells are placed into floating holders such that the well, and therefore larvae contained with it, were suspended below the surface of the water. Four days post spawning and fertilization, 20 larvae were aliquoted into

		floating net-wells into separated well replicates (n = minimum of six replicate wells) for each population cross.”
2.14	Since you have a very complicated experimental design, it needs to be explained efficiently, because if not, only you can repeat this experiment on other coral species and no one else can. Figure 1B is very simplistic and not at all helpful to understand your experiment.	Please see changes made due to Comment-Responses in 1.10. Additional information has been added to the Methods and Supplementary, including:  -Table on spawning crosses -Edits suggested by Reviewers to improve clarity of experimental design Figure 1B. -Information on larval and juvenile heat stress
2.15	Why there is no statistical analysis on any of your data?	A multitude of new analyses have been added. Further, we have highlighted previous analyses: Previous analysis: Lines 133-136: “Generalized linear models performed in DESeq2 using Benjamini-Hochberg multiple test corrections accounting for temperature and tank replicates identified four significantly differentially abundant ASVs between these treatments (as outlined in the main text, Supplementary Fig. 1).” Results Lines 157-160: “Survival under heat stress was associated with 6- and 8-fold greater abundances of I4 and C15 and 25- and 22-fold less C15g and C12 compared to surviving juveniles in the control treatment when exposed to sediments (DESeq2 Benjamini-Hochberg multiple test correction $P_{adj} < 0.05$).” FOUR new statistical analyses sections, encompassing a range of tests: Methods Lines 440-446: (1) “Larval and juvenile survival (amongst and between crosses) and between larvae and juveniles were assessed using the non-parametric, two-sided, Wilcox.test from the package “ggpubr”⁶⁶. This non-parametric test does not assume normality in the data. These tests included the comparison of the mean values between each group of crosses at 27 and 35.5°C (two independent groups) to assess if they were statistically different at the final timepoint (Supplementary Figs. 3 and 4).” Lines 102-104: “Survival under the elevated temperature varied widely across the 25 crosses (Figs. 2a, b), with winners and losers at both the larval

and juvenile stages across the four Symbiodiniaceae treatments (Figs. 2a, b, all larval cross comparisons $p < 0.05$, Supplementary results and Figs. 2-4).”

(2)

Statistics have been run for symbiont comparisons and added to Figure 2. Supplementary Figures have also been added:

Lines 150-152:

“...inshore CU sediments, conferred the highest survival at elevated temperatures to their juvenile hosts (77.7% [\pm 3.5]) whilst suffering the lowest mean survival at the control temperature (75.4% [\pm 4.5], Fig. 2b, c; comparison between temperatures not significantly different, $p = 0.72$).”

Lines 138-140:

“Interactive effects were less prominent in juveniles infected with *D. trenchii*, where the same crosses survived well at the control and elevated temperature treatments (CUxLS, DRxBK, and CUxDR, Supplementary Fig. 3).”

Lines 140-144:

“At the control temperature, juveniles infected with *C. goreau*, the progenitor of SS1 and the most common symbiont associated across a range of Cnidaria²⁵, were the best survivors at on average >60% across all the population crosses. However, these host-symbiont pairings suffered the highest mortality at the elevated temperature treatment, highlighting the fragility of this widespread symbiotic partnership.”

Lines 118-130:

“When averaged across familial crosses, survival at elevated temperatures varied significantly in only the *C. goreau* symbiont treatment, demonstrating the conferral of increased survival of juvenile corals at elevated temperatures by the lab-evolved *C. goreau* strain SS1 compared to the wild-type *C. goreau* (Fig. 2c), similar to the tolerance provided by *Durusdinium* (also see Supplementary results). Specifically, only juveniles with *C. goreau* performed significantly worse ($p = 0.001$) at heat compared to the other symbiont treatments ($p = 0.15 - 0.72$), indicating that juveniles in both the SS1 and the sediment treatments performed as well as the “stress tolerant” *D. trenchii* at 32°C (Fig. 2c). Juveniles infected with SS1 survived approximately on-par at both temperatures with juveniles infected with *Durusdinium* (control: 82.5% [\pm 3.5] vs. 82.4% [\pm 3.3] and elevated: 76.6% [\pm 4.0] vs. 77.2% [\pm 3.3]). At the level of individual familial crosses, there were no significant differences in survival between 27 and 32°C at the juvenile stage when

examined within each symbiont treatment (all juvenile cross comparisons Wilcoxon test $p > 0.05$, Supplementary Fig. 3).”

(3)

Statistical comparison of larval vs. juvenile survival

Supplementary Lines 31 – 45:

“Comparison of survival between larvae and juveniles. To assess if more thermally tolerant larvae grew into more thermally tolerant juveniles, survival at both life-stages was compared at the final timepoint for each life-stage. It should be noted that these life-stages were measured for different lengths of time, under different symbiont states (symbiotic and aposymbiotic) and at different elevated temperature treatments, so comparisons as such should be taken with caution. At the control temperature (27°C), only three crosses differed significantly in survival, in which BKxBK, CUxBK, LSxBK larvae survived better than their juvenile counterparts ($p = 0.00042 - 0.047$; Supplementary Fig. 4).

In the elevated temperature treatments (35.5 or 32°C for larvae and juveniles, respectively), most of the crosses survived better as juveniles compared to as larvae (~50% of crosses, potentially due to lower heat), including juveniles from crosses CUxDR, CUxLS, CUxSB ($p = 0.011 - 0.0088$), DRxBK, DRxDR, DRxSB ($p = 0.00087 - 0.0024$), LSxCU, LSxLS ($p = 0.00092 - 0.001$), and SBxLS and SBxSB ($p = 0.0014 - 0.011$). Only 15% of cross comparisons resulted in larvae that survived better compared to juveniles, including BKxCU, BKxSB, CUxBk ($p = 0.00062 - 0.023$).”

(4)

Multiple statistical tests outlining differences in environmental data.

Lines 54 – 68:

“Non-parametric Wilcoxon tests were used to determine if there were statistically significant differences in the mean value of predictors (Fig. 1c-h) across each reef (pairwise comparisons, $p < 0.05$ denoted by asterisks in Fig. 1c-h). SST_av.pre varied significantly between Curd and Sand Bank 7 (Fig. 1c; $p = 0.049$), Davies ($p = 3.6e-7$), and Backnumbers ($p = 1.5e-5$). SST_av.pre did not vary between Long Sandy and Curd or Sand Bank 7 ($p = 0.39$ and 0.17) but it did vary with Davies and Backnumbers ($p = 0.001$ and $7.4e-5$). Sand Bank 7 also varied with Davies and Backnumbers ($p = 0.0027$ and 0.023). Davies and Backnumbers did not differ significantly in SST_av.pre ($p = 0.34$). SST_stdev did not differ significantly across any reef comparisons (Fig. 1d; non-parametric Wilcoxon test all $p = 0.38 - 1$). SST_av.post differed significantly between Curd reef and Long Sandy ($p = 0.02$), Sand Bank 7, ($p = 0.031$), Davies ($p = 0.0051$) and Backnumbers ($p = 0.02$), but not between any of the other reef

		combinations (Fig. 1e). SSTA_Freq_stdev differed significantly between Backnumbers and Curd ($p = 0.038$), Davies ($p = 8e-4$), Sand Bank 7 ($p = 0.055$), Long Sandy ($p = 0.032$) and Davies compared to Long Sandy ($p = 0.051$, Fig. 1g). TSA_DHW_stdev did not vary significantly between any reef combinations ($p = 0.087 - 1$, Fig. 1h).”
2.16	You use cultured Symbiodiniaceae in your experiments, you need to justify their efficacy since those in long-time culture are no more similar to those in the natural environment or in symbiosis in terms of their physiology.	This has been acknowledged and that information has been highlighted for readers: Lines 405 - 411: “To assess the resulting symbiont communities in coral juveniles, which are difficult to assign taxonomically with only light microscopy and whose growth may have been influenced by culture conditions ⁶³ , different batches of single coral juveniles were assessed using two independent next generation sequencing methods (amplicon and RNAseq) to justify the efficacy of infection of each cultured strain (see below and Supplementary results). We acknowledge that long-term culturing may have changed these strains physiologically/genetically in comparison to fresh isolates. However, we are interested in the heat tolerance of these specific cultured strains, given their potential use in restoration initiatives.”
2.17	Also, see “Limitations of Using Cultured Algae to Study Cnidarian-Algal Symbioses and Suggestions for Future Studies” by Maruyama and Weis 2020 - https://onlinelibrary.wiley.com/doi/10.1111/jpy.13102	Please see Comment-Response 2.16. This information and reference have been added. Please see reference list.
2.18	Line 103 -109 – This seems, but obvious random effects at any given point of time. Corals from warm locations do better sometimes and those from not so warm do better some times, and random crosses have better response sometimes. So, if you repeat this experiment, will you see the same cross response as you see here, or you think it might change...	It is true that this variability may seem random. However, more recent results from field tests of these crosses were consistent with laboratory tests. This information has been added: Lines 106 -109: “Although variability was high, survival responses from these laboratory tests were consistent with patterns seen in the field when a subset of crosses were outplanted onto the reef (e.g. 75 - 100% survival across all treatments for SBxLS juveniles ²¹).”
2.19	Line 113-117 – I don’t understand this, how come you say that lab-evolved strain is better? From figure 1C the survival and moreover their response is not on par with other Symbiodiniaceae. I see a significant difference in the response and as usual Durusdinium trenchii fares better. I would like to see statistical analysis done on your data, at	(1) The lab-evolved strain is better because juveniles with this strain performed as well at heat as they did at ambient. In comparison, juveniles exposed to C1 performed worse at heat compared to ambient. These additional statistical tests have been performed on this data (see Comment-Response 2.15) and summarized in Figure 2C. (2)

	least simple statistics to show if there is any significant difference within and between treatments across your crosses as well as juveniles and larvae	Moreover, we did not mean to suggest that SS1 was better than D1a, only that the provisioning of SS1 provided heat tolerance generally equal to D1a. This has been clarified. Lines 118-130: “When averaged across familial crosses, survival at elevated temperatures varied significantly in only the C. goreau symbiont treatment, demonstrating the conferral of increased survival of juvenile corals at elevated temperatures by the lab-evolved C. goreau strain SS1 compared to the wild-type C. goreau (Fig. 2c), similar to the tolerance provided by Durusdinium (also see Supplementary results). Specifically, only juveniles with C. goreau performed significantly worse ($p = 0.001$) at heat compared to the other symbiont treatments ($p = 0.15 - 0.72$), indicating that juveniles in both the SS1 and the sediment treatments performed as well as the “stress tolerant” D. trenchii at 32°C (Fig. 2c). Juveniles infected with SS1 survived approximately on-par at both temperatures with juveniles infected with Durusdinium (control: 82.5% [± 3.5] vs. 82.4% [± 3.3] and elevated: 76.6% [± 4.0] vs. 77.2% [± 3.3]). At the level of individual familial crosses, there were no significant differences in survival between 27 and 32°C at the juvenile stage when examined within each symbiont treatment (all juvenile cross comparisons Wilcoxon test $p > 0.05$, Supplementary Fig. 3).” (3) Please see Comment-Response 2.15 We have also added the requested statistical tests and new Supplementary Figures for difference in survival within and between larval and juvenile crosses using non-parametric Wilcoxon tests.
2.20	And the, in your supplementary file (predictive models) you say “the overall survival of the C1 treatment at elevated temperature was lower than other treatments (Fig. 2). SS and heat-tolerant D1a provided an almost equivalent heat protection to juveniles at 32°C although the explanatory power of the....”	Please see Comment-Response 2.19 above for a summary of the changes made.
2.21	Line 132 onwards – I don’t understand this. Is this right? – you have different Symbiodiniaceae composition in your control and high temperature treatments? If so, why? And if not, then please excuse me, I cannot understand. Maybe you can describe it in a way it can be understood easily?	This has been further clarified in the text above this section: Lines 116-118: “Juveniles from each cross were exposed to the same four symbiont treatments at control and elevated temperatures (at 27° and 32°C: Cladocopium goreau, SS1, Durusdinium trenchii, and the free-living mixture).” Lines 152-157:

		“Although juveniles were exposed to the same sample of sediments originally, uptake of symbionts from the sediments by host juveniles differed in the control and heat treatments. Symbiont communities in juveniles at the control temperature were predominantly made up of taxa from Cladocopium and Fugacium, whereas juveniles exposed to heat were dominated by these taxa in addition to the more cryptic “I” clade (see further discussion of Fugacium and “I” in Supplementary).” In Methods: Lines 403-404: “The sediment treatment was therefore a choice experiment, whereas the other treatments were not (only a single algal strain was provided).”
2.22	Line 162 – This is outdated and based on the response of 1 coral species. Corals do tolerate more than 1-1.5 °C – for example, Coelastrea aspera, corals in the Persian and Arabian Gulf and many other places...I think you use that the study by Berkelmans et al just to make your point here about selective breeding mechanism.	This information and additional references have been added in the beginning paragraphs: Lines 48-49: “In some of these locations, acquired heat tolerance has since been shown across diverse species (e.g. Coelastrea aspera⁹) and regions (e.g. the Persian Gulf¹⁰).”
2.23	Line 251 – so how will you find heat-tolerant colonies in the natural environment? I am not convinced from this work that you can find them. Also, I am bit confused. Is there any result or did you do any work on stress resistance or sensitivity of aposymbiotic larvae and juveniles? Because how do you know if the tolerance you see in your experiment is because of tolerant parents or just because of the experiment effect using cultured symbionts? Coral response, mainly to temperature stress, is relative to time and space. It also depends on acquired thermal breadths over time and on the thermal thresholds of individual coral species. Moreover, as authors mention in lines 279-284, there are so many other factors that might influence how a coral colony will react to its environment. So, a coral colony in a particular location may be not bleach this year but might bleach next year. Or once considered temperature stress resistant Porites lutea, is not so much resistant anymore as seen by its reported bleaching from various locations in past couple of years.	This information has been added: Lines 285-292: “Finally, the multitude of factors influencing the heritability of heat tolerance makes ascribing drivers of tolerance challenging⁵³. For example, holobiont tolerance is influenced by the algal and bacterial symbionts, and the acclimatory history and potential of the host and symbionts (reviewed in⁵⁴). However, the operationalization of increasing heat tolerance via genetic adaptation in at least some coral species is possible given the strong heritability of this trait (for theory see¹⁷, for quantification in corals see^{16,55}). Given the current difficulty in finding heat tolerant individuals, our method represents a step-change improvement in detection.”

	Even in Plaumbi et al, they discuss “Thus, acclimatization alone cannot be expected to completely overcome the threat to corals from widespread bleaching events, especially if the onset of high-temperature stress is abrupt and sustained. In this regard, the tempo and severity of heat anomalies will be critical for effective coral acclimatization” in addition to that the microbiome and other factors What I want to say is, we cannot be sure that to identify potential parents across latitudes that might help us make resistant juveniles in the laboratory.	
2.24	Line 255 onwards – I don’t understand this. You used mainly cultured symbionts and one modified symbiont to make stress resistant combinations. How this same as presence of heat tolerant adults in the natural environment? Figure 4 title says “Intrinsic resistance models identify hundreds of potential reefs as predicted sources of heat-tolerant adult corals able to produce offspring of optimal survival under heat stress”, this itself is contradictory to this work, because there is little chance for those adults in the natural environment to cross with each other – for example parents from north with those in central GBR. did Dixon et al and Palumbi et al conduct their experiment to search for heat-tolerant corals? Dixon et al, similar to this work, perform crosses using parents from different latitudes to show again, that it is possible to get tolerant juveniles by crossing parents, and Palumbi et al just look at how corals between warm and normal pool respond to temperature stress in one location in American Samoa.	The order in which this information is presented has been modified and further sentences have been added to aid in clarity: Lines 261 -262: “In the search for heat-tolerant corals, many reef habitats and large latitudinal clines ¹⁵ to semi-isolated backreef pools ⁵³ have been surveyed. Hundreds of reefs were identified here as potential predicted sources of heat-tolerant corals also exhibiting high heritability for heat tolerance (Fig. 4). Of the approximately 3,255 reefs within the GBR marine park boundaries, in total, approximately 251 unique reefs or closely adjacent (~7.5% of total reefs and potentially non-reefal locations) were identified as potential sources of heat tolerant parental stock for producing larval and juvenile corals capable of surviving temperatures from 32- 35.5°C.”
2.25	Line 290-292 – a. I am not convinced that lab-evolved strain confers increased survival to juvenile corals (see my previous comment). Need more statistical analysis and clear data presentation.	a. We did not mean to suggest that SS1 was better than D1a, only that the provisioning of SS1 provided heat tolerance generally equal to D1a. This has been clarified. Lines 118 – 131: “When averaged across familial crosses, survival at elevated temperatures varied significantly in only the C. goreau symbiont

	Also as mentioned before, you need to discuss about using cultured or modified symbiont in terms of their physiology and long-term association with corals.	treatment, demonstrating the conferral of increased survival of juvenile corals at elevated temperatures by the lab-evolved C. goreau strain SS1 compared to the wild-type C. goreau (Fig. 2c), similar to the tolerance provided by Durusdinium (also see Supplementary results). Specifically, only juveniles with C. goreau performed significantly worse ($p = 0.001$) at heat compared to the other symbiont treatments ($p = 0.15 - 0.72$), indicating that juveniles in both the SS1 and the sediment treatments performed as well as the “stress tolerant” D. trenchii at 32°C (Fig. 2c). Juveniles infected with SS1 survived approximately on-par at both temperatures with juveniles infected with Durusdinium (control: 82.5% [± 3.5] vs. 82.4% [± 3.3] and elevated: 76.6% [± 4.0] vs. 77.2% [± 3.3]). At the level of individual familial crosses, there were no significant differences in survival between 27 and 32°C at the juvenile stage when examined within each symbiont treatment (all juvenile cross comparisons Wilcoxon test $p > 0.05$, Supplementary Fig. 3).” b. Please see edits from 2.15, 2.19, 1.10/2.14 for additional statistics and Figure presentation. c. Please see 2.16
--	--	---

Reviewer #3 (Remarks to the Author):

3.1	This is an excellent, ground-breaking manuscript and I recommend publication following a minor revision. The authors report out on a monumental effort to increase heat tolerance via selective breeding/novel host-symbiont associations. The authors use these results to model where corals may be sourced that have inherent heat tolerance, as well as the environmental conditions of reefs that correlate with elevated heat tolerance. This is incredibly important and I hope to see it published soon.	We are very appreciative of Reviewer 3 for their comment that “this is an excellent, ground-breaking manuscript and I recommend publication following a minor revision.” and for recognizing that this was a “monumental effort” and “incredibly important.”
3.2	The authors might consider changing their title (see comments below). I think there are plenty of examples of heat tolerant corals that aren't from low latitudes.	This title has been changed to be broader and highlight the focus on the models. “Predictive models for the selection of thermally tolerant corals”
3.3	Also, the discussion seems to run out of steam and go on too long. Please tighten it up and organize it better.	Significant edits have been made throughout the manuscript, especially the Results/Discussion section. Please see the “Tracked Changed version of the draft for full details.
3.4	Line 31. a. I don't think it's a common assumption that only low latitude corals are heat tolerant. Rerword.	a.

	Maybe say it is assumed low latitude corals are more heat-tolerant or something along those lines. I just think that anyone who knows anything about coral heat tolerance does not assume that only low latitude corals are heat tolerant. b. I'd also suggest changing the title so as not to dwell on the low latitude aspect of the study. I think its worth discussing, but I don't find that aspect of the work the most important.	Reviewer 1 also commented on this sentence (Comment-Response 2.4). The edits are summarized in Response 2.4. This sentence has been changed per Reviewer 3's suggestion: Lines 45 -46: “Although it is assumed that low latitude corals are more heat-tolerant, resilient reefs of tolerance are increasingly being found.” b. The title has been changed to highlight the models conceptually, not the low latitude aspect of the study. Specifically: “Predictive models for the selection of thermally tolerant corals” The word “earlier” has been added here to show that this paradigm is changing: Lines 219-222: “Latitude was the poorest predictor of coral heat tolerance in both purebred and hybrid offspring, which conflicts with the earlier paradigm^{39,40} that warmer reefs closer to the equator produce more heat-tolerant corals^{40,41}, and that heat-tolerant corals are restricted to low latitudes.”
3.5	Reference 2 is not a good reference to describe bleaching.	Reference 2 has been changed from Baker 2001 to the more appropriate Baker et al. 2008 “Baker, A. C., Glynn, P. W. & Riegl, B. Climate change and coral reef bleaching: an ecological assessment of long-term impacts, recovery trends and future outlook. Estuar. Coast. Shelf Sci. 80, 435–471 (2008).”
3.6	Reference 5 reports on bleaching in Singapore, not Palau	Thank you for catching this. This has been changed to Singapore in the following section: Lines 43 -49: “Past exposure of corals to high temperatures over short (single generation, acclimation) and longer ecological timescales (multiple generations, adaptation) shapes corals' bleaching responses and survival during thermal anomalies. Although it is assumed that low latitude corals are more heat-tolerant, resilient reefs of tolerance are

		increasingly being found. Signatures of acclimation and adaptation have been detected globally, including in Singapore ⁶ , the Great Barrier Reef (GBR) ⁷ and the Persian Gulf ⁸ .”
3.7	Line 82. Change to 23-33oC	These values have been reversed for clarity: Lines 78- 79: “...encompassing a 23 - 33°C mean annual temperature gradient (>6° of latitude, ~900 km, Fig. 1a).”
3.8	Line 93: Insert “of” in between mixture and free-living	“Of” has been added: Lines 88- 91: “Larvae from each cross were then settled to produce juveniles (Fig. 1b), and subsequently inoculated with one of three Symbiodiniaceae species or a mixture of free-living symbionts within sediments (specifically, Cladocopium goreau , the lab-evolved “Selected Strain” derived from C. goreau - SS1 sensu ¹⁸ ...
3.9	Line 188: Start new paragraph here	A new section and paragraph break has been added here: Line 215: “The most influential predictors of survival...”

Reviewer #4 (Remarks to the Author):

4.1	I really like what Quigley and van Oppen are trying to do here.	We appreciate that Reviewer 4 finds value in the questions we are asking, and the analyses were are using.
4.2	My problem is that I can’t really assess the validity of the conclusions because I cannot work out some important sample size information.	All samples size information have now been added to the Supplementary material: Including sample sizes for the survival of larvae and juveniles (Supplementary Table 3 and 4). This information has also been added to the Figure 2 legend and in the text to direct readers to the Supplementary. Figure 1 legend: “A summary of sample sizes can be found in Supplementary Tables 3 and 4.

		A summary of the experimental design for larvae and juveniles is provided in (Fig. 1B). Samples sizes per treatment are given in Supplementary Tables 3-4.
4.3	a. How many parent colonies were collected from each reef, and b. how many of those (at each reef) provided sperm or eggs or both? c. This is important for working out how good is the evidence that the reef effects identified are actually effects of the reef, or whether they may reflect vagaries of which genotype was sampled. The follow up to this question is then how was this incorporated into the analysis? Say reef A had corals A1 and A2, and reef B had corals B1 and B2. If A1 x B1 and A2 x B2, then all the larvae and juveniles come from two sets of parents. So is each set of parents treated as a random effect nested within reef, and then all the corresponding juveniles or larvae are the individual observations nested within the parent? If not, how do the authors know they are not quantifying parent affects that may or may not reflect what holds generally on the reef that the parent is from?	a. and b. This information has now been added to the Supplementary information, including a new Supplementary Table 1. Specifically: Lines 335 -340: “Gravid Acropora tenuis colonies were collected from three reefs in the far north of the Great Barrier Reef (Long Sandy, Curd, and Sand Bank 7) and three reefs in the central GBR (Backnumbers and Davies) (Fig. 1a). Ten gravid colonies per reef were collected from Long Sandy, Curd, Sand Bank 7, and Davies, and 15 gravid colonies from Backnumbers. Not all colonies per population spawned, and the total number of maternal and paternal colonies used per cross is listed in Supplementary Table 1.” c. This is a valid concern. This address this, we performed additional statistical analyses and Supplementary Figure 6 to determine if differences in parental identity per cross contribute to differences in survival. Lines 355- 361: “To determine if differences in parental identity per cross contributed to differences in survival in the larval and juvenile life-stages, Pearson correlation coefficients (R^2) and corresponding significance values were calculated (Supplementary Fig. 6). The number of parents explained little variation in either larval or juvenile survival ($R^2 = 0.00051 - 0.04$), with only one comparison significant, although variation explained was extremely low ($R^2 = 0.04, p = 8e^{-6}$). To account for any potential influence of maternal or paternal identity, survival models (see below), incorporated both maternal and paternal identity as random effects.”

4.4	A similar argument applies to the boosted regression tree analysis. With only 5 reefs, is it really possible to distinguish the predictive power of so many environmental variables? Along the same lines, it seems like the only way to cross-validate with 10 folds would be to treat each observation of each larva or juvenile as statistically independent. Again, I would want to be reassured that the nested structure of the data is accounted for in the analysis. It is hard for me to imagine how data from 5 reefs could be used to generalize to the whole Great Barrier Reef in a regression model with so many variables. I could be wrong, but I would want to know how the authors' approach to the analysis accounts for these issues.	This is a good point. One of the strengths of BRT is their ability to handle smaller datasets by tuning the models to balance predictive power versus overfitting. To explain how this was done in more detail and validate these decisions, the following was added: (1) An extensive additional section has been added to the Methods that explains how the BRTs have been optimised using cross validation for this data to balance predictive power with small sample sizes. Section “Boosted regression trees” Lines 483 - 557 (2) Additional clarification about the level of statistical observations and the number of predictors vs. observation levels for predictive power. Lines 530 – 535: “The following levels were treated as statistically independent observations given the measurements were taken directly at these levels: each replicate well of pooled larvae and each single juvenile. Further, whilst there were only five “maternal” reefs, this dataset consisted of 25 unique familial combinations when these five maternal reefs were crossed with the same five paternal reefs, increasing the predictive power of the models given there were 3.5x more combinations of reefs compared to the seven predictor environmental variables.” (3) An additional section explaining why 10 folds cross validation is an appropriate here. Lines 505 -517: “To avoid overfitting the optimum number of trees, which is of particular concern for small datasets, K-fold cross-validation was performed with a 10-fold cv using a “random” search of the hyper-parameter space. During this process, the data was split into 10 randomly chosen testing subsets and was repeated to make 10 training sets, meaning that cv involves randomly splitting the data into 10 x 10 (nt, ⁷¹) random splits and repeated, with each step randomly omitting one test set for validation and optimal nt. The use of 10 folds is commonly used and recommended in method guidelines ⁷¹. To test the appropriateness of this parameter choice on our data, further
------------	--	---

		diagnostics were run for each model to assess if cv adequately balances the squared error loss versus the number of trees (Supplementary Fig. 7), in which the optimal number of trees should reflect where there is a decline in prediction error to a minimum value (dashed line) compared to independent training test estimates (solid line). These figures suggests that 10 randomly chosen subsets of the data (error on the test data) performed very similarly to the training dataset, with the optimum number of interactions (trees) designated in grey.” (4) New Supplementary Figure 7 has been added to show how 10 folds cross validation performs with these data.
--	--	--

Reviewers' Comments:

Reviewer #2:

Remarks to the Author:

After reading the revised version and rebuttal, I realise that the authors have put lots of efforts in improving the manuscript. New title is kind of detached from the whole thing. I am not sure how this is going to explain so much of physiology work you have included, because this is not just the story of predictive models.

While I see what reviewer want to convey to the readers through this manuscript, I still feel uneasy with certain aspects that are not clear - physiology part of this manuscript.

Here are my comments

(all page and line numbers are for the word file of the manuscript with track changes)

1. Page 5, line 336- This sentence by the authors is irrelevant here except for the case of Sediment Symbiont source -- because for all there 3 you cannot be sure if their differences can be attributed to variation in photosynthetic mechanisms and the regulation of specific genes - since they are cultured Symbionts.

This may also the reason why you see difference between those cultured Symbionts and those from sediments- both infection and response in the symbiont found in the nature are dynamic and natural.

2. This pertains to question by Reviewer 1 and also my own curiosity. How can there be mortality in Control treatment tanks. I don't understand. If you have mortality in control treatment, then how can you call it "control treatment".

The temperature you have used for control treatment are mean temperature for those corals at their locations. If what you say -- location, mother colonies etc influence how crosses will respond to temperature treatment, then all of them should be ok at control temperature - technically - because you are conducting your experiment in a controlled and very efficient system - SeaSYM - if this is not happening then there is some problem with your experiment.

In supplement, you justify this by quoting the review in MEPS done by your own group and others. The place where 30-99% Juvenile mortality is mentioned - the studies were conducted in the field. The factors effecting juvenile survival in natural environment is not comparable to a controlled environment where literally such disturbances does not exist. Hence, mortality has to be none (which I also agree cannot be achieved because of the experimental setup and no control on how life behaves) or very less compared to stress tanks.

Then you go on to say that it is because of the duration of the experiment - almost 2 month. What happens if the duration is less - say one week or 2 week - does the mortality come down?

After reading the supplementary tables 3 and 4 for larvae and juvenile respectively. it shows that larvae do better than juveniles in control

Juveniles have problem - now, maybe those variations are because of uneven sample number for each treatment and cross.

Also, I don't understand how you calculate "percent survival" for time point 1 and 2 in those tables - please elaborate

If you still insist on what you have stated, then don't say it is control treatment - it can be mean temperature stress treatment. Because no matter what justification your will give, this is a stalemate topic of discussion

3. Page 6 - Differential association of symbionts from sediments with juveniles could be coincidence and chance. You don't have enough number of juveniles in each treatment and replicate wells to show that the association you saw at different temperature treatments was just random chance and not represent reality.

If the sample number for sequencing is less, you might have missed a juvenile associated with "I" in

the control treatment. Can you be sure that the sequencing you did on the juveniles was representative of all the juveniles in that treatment and it would not have changed if there were say-> 20 juveniles for each treatment and each replicate

4. Page 7- "Symbiodiniaceae shuffling in the related species, *Acropora millepora*, increases tolerance by 1-1.5°C 20. Therefore, selective breeding mechanisms have the potential capacity to double heat tolerance"

This is outdated...there are many studies and observations after that one on *A. millepora* showing that there are other corals which have more tolerance with or without shuffling. Authors are using citation selectively to show that "selective breeding mechanisms can potentially double the tolerance - meaning unto 3 °C - because of what they observed in the GBR - "Therefore, surviving juveniles were able to withstand on average an increase of 2.9°C ± 0.08"

5. Page 11, line 691~ Authors write, "However, the operationalization of increasing heat tolerance via genetic adaptation in at least some coral species is possible given the strong heritability of this trait"

This is misleading. Those studies just show what is obvious, Please give concrete example of a study or studies that show - tolerance can be transferred between generation of corals via intervention and it has been that they can do well in natural environment. I could not find one. you mention Dixon - Dixon work just show what is obvious and already know from so many physiology experiments - "Here, we show an up-to-10-fold increase in odds of survival of coral larvae under heat stress when their parents come from a warmer lower-latitude location" but on a large scale and your own paper in *Mol Ecol* - which is similar to this own in experiment

It is not a new information that early stages of corals which arise from parents present in warmer, marginal and extreme environment are more robust.

You are just taking a big leap when you make that statement --

Figure 2 and Figure 3 - are still messy and complicated. Figure 1 - reads clear than previous version and easy to understand.

Reviewer #4:

Remarks to the Author:

Apologies to you and to the authors for my delay in getting to this.

In my review of the initial submission, I raised two main problems. One was lack of clarity with respect to sample sizes, and the other was the use of a cross-validation approach that assumes each observation of a larva or juvenile is statistically independent. I appreciate the greater degree of sample size information, and this partly addresses my concern. However, the main piece of information that I wanted is not presented: how many colonies provided eggs and sperm came from each reef? The supplementary table does not show this information, because presumably eggs and sperm from a given colony could have been used in multiple crosses. I asked this question in order to get some idea of how much the authors might be quantifying effects of individual parents versus of the population of parents on a given reef. In response, the authors did an analysis of survival against NUMBER of crosses. I don't understand how this addresses the question. The question is about whether the IDENTITY of the parents used in the crosses is important, and whether a particular parent that produces particularly vigorous or inviable offspring for example could be driving whether a "reef" appears to confer high or low survival. If the authors included parental ID as a random effect (see below) nested within reef, and they were still seeing significant reef effects, this would address my concern, but the text says in one place that parent was a random effect (lines 358-360) but then later that it was a fixed effect (434-436). So I cannot figure out if the authors can address this issue with the results they already have or not. For these analyses it would be very helpful to have tables of statistical results reporting random

effect and fixed effect estimates and their standard errors and so forth. Then there is no question about what terms are in the statistical analyses.

On the second point, I think the authors have not come to terms with what they are assuming by doing their cross-validation. I am guessing that the cross-validation did indeed treat larvae and juveniles as all statistically independent of one another, so observations from particular parent colonies/crosses are appearing in both the testing and training partition in any given CV run. If so, literally what the authors are doing is using a cross-validation that would optimize the choice of a model for predicting the responses of a new set of larvae and juveniles from the same set of colonies that they have, if they repeated their crosses with these same colonies. That is not what they are trying to do with the model in terms of prediction. What they want to do is extrapolate the results from their five reefs to what they would expect for crosses from other reefs that they don't have data from. A cross validation to optimize this should then exclude the data from a whole reef, or maybe exclude all the crosses involving mothers from one reef and fathers from the same reef or a different reef, then fit the BRT to the remaining data, and test the model against the excluded data. Then repeat excluding a different reef. Here the CV would be looking for a model that does a good job of predicting outcomes for crosses of unobserved parents from locations with a given combination of predictor variable values, which is how they want to use the model. By doing the random partitioning of the data as they have done it, the authors are substantially increasing the risk of overfitting for making predictions about reefs not in their data set.

One other thing that I noticed in the revision is that there are a lot of p-values reported for pairwise comparisons between all the different pairs of crosses. If there are 25 different crosses, this would be 300 different pairwise comparisons. There is no mention of a multiple comparisons correction being applied in most cases (multiple comparisons correction is only mentioned for the ASV analysis). The authors may have a problem here because with so many comparisons, p-values will have to be extremely small (around 0.0001 with a standard Bonferroni correction) to be deemed significant (some multiple correction methods are more restrictive than others though). An alternative way to deal with this would be to have random effects of parent ID or of cross, and test whether a model with the random effect fits better than one without. As noted above, I'm not sure if the authors already have such a model or not.

Visually, it looks almost certain that large differences between crosses are biologically real. If a random effects approach like I mentioned above is not feasible to implement, and multiple comparisons are so stringent nothing is significant, then the authors just need to take care to avoid referring to particular differences as being "significant", and will need to make an argument on the fact that the p-values are all for example skewed towards small values and therefore indicate that there are real differences, or something along those lines.

Last, I don't consider the BRT analysis to be essential to the paper, meaning I think it would be publishable without it.

Nature Communications manuscript NCOMMS-20-48044A REVIEWER COMMENTS -RESPONSES Predictive models for the selection of thermally tolerant corals based on offspring survival Authors K. M. Quigley, M. J. H. van Oppen			
	Comment	Response	Edits within manuscript or supplementary materials
2-1	After reading the revised version and rebuttal, I realise that the authors have put lots of efforts in improving the manuscript.	We thank Reviewer 2 for acknowledging the large amount of additional analyses performed in response to the Reviewers comments. This paper was substantially improved from this feedback.	NA
2-2	New title is kind of detached from the whole thing. I am not sure how this is going to explain so much of physiology work you have included, because this is not just the story of predictive models.	This specific new title was suggested by Reviewer 3. We have now further modified it to add further emphasis on the physiology data.	Additional information has been added to the title to incorporate this point: “Predictive models for the selection of thermally tolerant corals based on offspring survival”
2-3	While I see what reviewer want to convey to the readers through this manuscript, I still feel uneasy with certain aspects that are not clear - physiology part of this manuscript.	We have made substantial edits to the presentation of the physiology data and have added further supplementary results to improve the robustness and clarity of these results. Please see each comment below.	Changes have been outlined below one by one.

2-4	1. Page 5, line 336- This sentence by the authors is irrelevant here except for the case of Sediment Symbiont source -- because for all there 3 you cannot be sure if their differences can be attributed to variation in photosynthetic mechanisms and the regulation of specific genes - since they are cultured Symbionts. This may also be the reason why you see difference between those cultured Symbionts and those from sediments- both infection and response in the symbiont found in the nature are dynamic and natural.	Page 5 corresponds to line 136. Perhaps the Reviewer meant line 136 instead of 336? Line 336 is in the Methods. Given this comment, we have acknowledged that the performance of cultured symbionts may be different to free-living and have made edits accordingly.	Lines 182 - 185: “The improved performance of sediment-exposed juveniles at elevated compared to 27°C may be due to differences between cultured and free-living symbionts or due to the novel symbiont taxa sourced from this warm reef and may represent an untapped source of adaptive diversity for corals.”
2-5	2. 1) This pertains to question by Reviewer 1 and also my own curiosity. How can there be mortality in Control treatment tanks. I don't understand. If you have mortality in control treatment, then how can you call it "control treatment". The temperature you have used for control treatment are mean temperature for those corals at their locations. If what you say -- location, mother colonies etc influence how crosses will respond to temperature treatment, then	1) Earlier figures suggested the mortality at control was much higher than it was. This has been clarified. Specifically: Larval mortality in the control was minimal. -Survival of all the crosses was high in the mean stress temperature treatment [mean 96.2 % ± 0.4 Standard Error, median = 100 %, Supplementary Fig. 2].	Lines 105-107: “Mean survival of all the crosses was high in the 27°C temperature treatment (Figs. 2a, b; larvae: mean 96.2 % ± 0.4 Standard Error, median = 100 %, Supplementary Figs. 2-4), although variable, particularly for the juveniles (mean 82.02 % ± 1.8, median = 95.8 %, Supplementary Fig. 2).” We have changed “control” temperature to “27°C” treatment throughout and have added in further information about it being a “mean temperature treatment.” Supp Lines 14-21: “Elevated mortality of juveniles, even at 27°C conditions is common in corals and other organisms with mass-spawning reproductive strategies. For example, there is a well-documented drop in survival in the first few

all of them should be ok at control temperature - technically -because you are conducting your experiment in a controlled and very efficient system - SeaSYM - if this is not happening then there is some problem with your experiment. 2) In supplement, you justify this by quoting the review in MEPS done by your own group and others. The place where 30-99% Juvenile mortality is mentioned - the studies were conducted in the field. The factors effecting juvenile survival in natural environment is not comparable to a controlled environment where literally such disturbances does not exist. Hence, mortality has to be none (which I also agree cannot be achieved because of the experimental setup and no control on how life behaves) or very less compared to stress tanks. 3) Then you go on to say that it is because of the duration of the experiment - almost 2 month. What happens if the duration is less - say one week or 2	Survival was slightly lower for juveniles at control. -At the mean stress temperature treatment, survival of all the crosses was high [mean 82.02 % \pm 1.8 Standard Error, median = 95.8 %, Supplementary Fig. 2]. We believe that a mean 82.02% or median 95.8% survival of juveniles is acceptable. This is to be expected given reported background rates of mortality for corals, and in particular, K selected organisms. As you state, it could also be due to using the mean temperature of these reefs. Hence, we have changed “control” temperature to “27°C” treatment throughout and have added in further information about it being a “mean temperature treatment.” 2) This is further justified by foundational literature as well as a meta-analysis including published work from other reef regions outside of our working group.	months of juvenile life, generally accounting for around >30-99% mortality (reviewed in ¹). Hence, we propose that this mortality at 27°C temperatures is expected, especially given the long timeframe of this experiment (58 days) covering this early window of juvenile life. Survival at both temperatures were incorporated into the Boosted Regression Tree models, allowing the machine learning algorithm to learn from patterns in survival of both 27°C and elevated temperature datasets and thereby incorporates this variability.”
---	---	---

	week - does the mortality come down?	Moreover, this “background mortality” at control is factored into the BRT models, such that mortality at heat is groundtruthed against background mortality, therefore it is a control. 3) Photographs were only analysed for the first and last timepoints of the experiment. However, larval survival decreased over time according to theory, as seen in the original Figure 2a.	
2-6	After reading the supplementary tables 3 and 4 for larvae and juvenile respectively. it shows that larvae do better than juveniles in control Juveniles have problem - now, maybe those variations are because of uneven sample number for each treatment and cross. Also, I don't understand how you calculate "percent survival" for time point 1 and 2 in those tables - please elaborate	Yes, as shown in Figure 2, and Supplementary Tables 3, 4, and Supplementary Figure 4, larvae at controls do better than juveniles in control. Both larval and juvenile survival was quite high at ambient as seen in Figure 2 A and C. We appreciate that this is hard to see, so a new version of Figure 2 has been added. It is unlikely that variations are because of uneven sample number for each treatment and cross given sample	-Supplementary Figure 2 added. -We have completely revised Figure 2 as requested so this is clearer. -Supplementary Table 4 added. -Experimental design Supplementary Figure 7 added. Lines 105-107: “Mean survival of all the crosses was high in the 27°C temperature treatment (Figs. 2a, b; larvae: mean 96.2 % ± 0.4 Standard Error, median = 100 %, Supplementary Figs. 2-4), although variable, particularly for the juveniles (mean 82.02 % ± 1.8, median = 95.8 %, Supplementary Fig. 2).” Lines 444 – 462:

		sizes were very similar. To make this clearer, we have added an additional figure 2. It is more likely the lower variation is due to the symbiont treatments now. We have added more text to highlight this. The explanation on how percent survival for both life-stage (as shown in Supplementary Tables 3 and 4) was calculated has been further edited for clarity. For the calculation of survival in juveniles, further information has been added to the Methods and an additional Supplementary Figure 7 explaining the experimental design and analysis steps.	“Survival statistical analyses. Larval survival was based on the number of remaining larvae per replicate well at the final sampling time point. The final time point was chosen as approximately the time needed for which 50% of the larval crosses would reach 50% survival at the elevated temperature treatment. Survival was quantified as the average percentage of surviving larvae across each of the replicate wells as compared to the initial number of larvae added per well (Supplementary Table 3). Replicate, single juveniles per cross were tracked through time using photographs, allowing for individual-based survival analysis to be undertaken using the “survival” package 65 (Supplementary Fig. 7 for juvenile survival experimental design). Individual juveniles were assessed from photographs as either “alive” or “dead” at the final timepoint compared to initial photographs. This data was input into survival models. Survival models were performed using Kaplan-Meier methods using the survfit function; with symbiont treatment, temperature, tank replicates, cross, maternal reef, and paternal reef identities accounted for as fixed effects. As per standard output of survival models, the output included the number of censored individuals per fixed effect treatment and the total number of events (i.e. total alive individuals left at the final timepoint). Percentage survival and standard error for replicate juveniles were calculated per treatment by dividing the number of juveniles (n.event) by the total number of potential individuals to die given initial numbers (n.event + n.censor). Replication included replicate individual juveniles, within replicate wells, replicate plates, within replicate tanks per symbiont by each temperature combination (Supplementary Fig. 7, Supplementary Table 4).”
--	--	--	--

2-7	If you still insist on what you have stated, then don't say it is control treatment - it can be mean temperature stress treatment. Because no matter what justification your will give, this is a stalemate topic of discussion	We agree that this treatment should more accurately be called a “mean temperature stress treatment.” For ease of reading, we have changed “control” to “27°C” throughout the ms and have added a definition of what this treatment is.	The use of “control” has been deleted throughout and replaced with either “27°C” or “mean.” Lines 99 -103: “Larvae and juveniles were both exposed to 27°C (the approximate mean annual temperature of the five reef sites where reproductive adults were collected, Fig. 1c; this was used as a non-stress temperature) and elevated temperature treatments (larvae: 35.5°C for 56 hrs; juveniles: 32°C for 58 days, Fig. 2b, see “Methods” for justification of treatment temperatures).”
2-8	3. Page 6 - Differential association of symbionts from sediments with juveniles could be coincidence and chance. You don't have enough number of juveniles in each treatment and replicate wells to show that the association you saw at different temperature treatments was just random chance and not represent reality. If the sample number for sequencing is less, you might have missed a juvenile associated with "I" in the control treatment. Can you be sure that the sequencing you did on the juveniles was representative of all the juveniles in that treatment and it would not have	The results presented here in relation to specific symbiont taxa (I4, C15, C15g and C12) were statistically analysed using the DESeq2 package. This package therefore statistically tests the probability of detection and relative abundance of different ASVs and then applied a multiple test correction statistics (Benjamini-Hochberg) to determine if the detection is statistically significant given the total number of tests (which is reflective of biological replicates) and specifically tests against the probability of false positives. Therefore, we only discuss results and conclusions of symbiont taxa that are	Lines 172- 178: “Symbiont communities in juveniles at 27°C were predominantly made up of taxa from Cladocopium and Fugacium, whereas juveniles exposed to heat were dominated by these taxa in addition to the more cryptic “I” clade (see further discussion of Fugacium and “I” in Supplementary results). Survival under heat stress was associated with 6- and 8-fold greater abundances of I4 and C15 and 25- and 22-fold less C15g and C12 compared to surviving juveniles in the 27°C treatment when exposed to sediments (DESeq2 Benjamini-Hochberg multiple test correction $P_{adj} < 0.05$).”

	changed if there were say-> 20 juveniles for each treatment and each replicate	based on this multiple correction threshold.	
2-9	4. Page 7- "Symbiodiniaceae shuffling in the related species, Acropora millepora, increases tolerance by 1-1.5°C²⁰. Therefore, selective breeding mechanisms have the potential capacity to double heat tolerance" This is outdated...there are many studies and observations after that one on A. millepora showing that there are other corals which have more tolerance with or without shuffling. Authors are using citation selectively to show that "selective breeding mechanisms can potentially double the tolerance - meaning unto 3 °C - because of what they observed in the GBR - "Therefore, surviving juveniles were able to withstand on average an increase of 2.9°C ± 0.08"	We are trying to provide some context to the heat tolerance boost measured here. However, we recognize this reference is old. This section has been deleted.	Deleted: “For reference, the additional acquired heat tolerance via mechanisms like Symbiodiniaceae shuffling in the related species, Acropora millepora, increases tolerance by 1-1.5°C¹⁹. Therefore, selective breeding mechanisms have the potential capacity to double heat tolerance.”
2-10	5. Page 11, line 691~ Authors write, "However, the operationalization of increasing heat tolerance via genetic adaptation in at least some coral species	We do not want to mislead. We have toned down this statement.	Deleted: “However, the operationalization of increasing heat tolerance via genetic adaptation in at least some coral species is possible given the strong heritability of this trait (for theory see 17, for quantification in corals see

is possible given the strong heritability of this trait" This is misleading. Those studies just show what is obvious, Please give concrete example of a study or studies that show - tolerance can be transferred between generation of corals via intervention and it has been that they can do well in natural environment. I could not find one. you mention Dixon - Dixon work just show what is obvious and already know from so many physiology experiments - "Here, we show an up-to-10-fold increase in odds of survival of coral larvae under heat stress when their parents come from a warmer lower-latitude location" but on a large scale and your own paper in Mol Ecol - which is similar to this own in experiment It is not a new information that early stages of corals which arise from parents present in warmer, marginal and extreme environment are more robust. You are just taking a big leap when you	This statement has been deleted and replaced by the suggestion from Reviewer 1.	16,54). Given the current difficulty in finding heat tolerant individuals, our method represents a step-change improvement in detection." Added Lines 302-304: "Although physiological experiments are useful for investigating if tolerance can be transferred between generations, further work is needed to confirm if this is maintained in natural environments."
---	--	--

	make that statement --		
2-11	Figure 2 and Figure 3 - are still messy and complicated. Figure 1 - reads clear than previous version and easy to understand.	We are glad Reviewer 1 finds Figure 1 much improved. We agree that Figure 2 was still messy and complicated. We have completely reformatted this. We have also only presented the last timepoint for comparison. Specifically Figure 3 was divided into two separate figures.	Figure 2 and 3 have both been updated and replaced.
2-12	Reviewer #2 was also asked to cross-comment on responses to reviewer #1, who was not able to look at the revised version. This reviewer was asked for their feedback on the responses to points #2, #4, #8, #10 and #11 from reviewer #1. Comments from reviewer #2 are as follows:	We thank Reviewer 2 for also providing this feedback.	NA
2-13	(1) 2. Did you look at the correlation of larval thermal tolerance to recruit thermal tolerance? If not, this seems like a really important result either way! Does a cross who is more thermally	(1) This additional analysis was supplied in our first review (previously Supplementary Fig. 4, now Supplementary Fig. 8).	Supplementary Lines 33-40: “It should be noted that these life-stages were measured for different lengths of time, under different symbiont states (symbiotic and aposymbiotic) and at different elevated temperature treatments, so comparisons as such should be taken with caution.”

tolerant as a larvae predict who is thermally tolerant as a recruit? This is especially important info for restoration and I would love to see this result incorporated in a revision. It could be that this analysis was already performed, however I did not find the information easily. (2) I have same concern as the reviewer. We cannot say anything because the authors did not consider this aspect during the experiment. They do have the data but is not matching – as they say in their rebuttal and include the information in the supplement. To answer this question, every stage of the development including the recruits need to be exposed to same stress conditions and same parameters measured. Which has not been done	Given that the comparison of larvae and juveniles was not our original research question (it was asked to be included by Reviewer 2), we had not made our experimental design for this and therefore our heat treatments were not the same. That is why this additional analysis is in the supplementary with the following warning: “It should be noted that these life-stages were measured for different lengths of time, under different symbiont states (symbiotic and aposymbiotic) and at different elevated temperature treatments, so comparisons as such should be taken with caution.” (2) This additional analysis was performed (Supplementary Figure 8) as requested by Reviewer 2 but we hold the same concerns as Reviewer 1 about its legitimacy. For this reason, it has not been added to the main results.	Supplementary Lines 33-46: “Comparison of survival between larvae and juveniles. To assess if more thermally tolerant larvae grew into more thermally tolerant juveniles, survival at both life-stages was compared at the final timepoint for each life-stage (Supplementary Fig. 8). It should be noted that these life-stages were measured for different lengths of time, under different symbiont states (symbiotic and aposymbiotic) and at different elevated temperature treatments, so comparisons as such should be taken with caution. At 27°C, only cross CUxBK differed significantly in survival, in which larvae survived better than their juvenile counterparts (Wilcoxon test Bonferroni $P_{adj} = 0.019$; Supplementary Fig. 8). In the elevated temperature treatments (35.5 or 32°C for larvae and juveniles, respectively), most of the crosses survived better as juveniles compared to as larvae (5 of the 22 crosses, ~22.7%, potentially due to a lower temperature), including juveniles from crosses, DRxDR (Wilcoxon test Bonferroni $P_{adj} = 0.038$), LSxCU and LSxLS ($P_{adj} = 0.0058 - 0.041$). Only 1 of the 22 (4.5%) cross comparisons resulted in larvae that survived better compared to juveniles, including BKxSB and CUxBk ($P_{adj} = 0.027 - 0.046$) (Supplementary Fig. 5).” - Supplementary Figure 8 edited to include post-hoc test values.
---	---	---

2-14	4. Looking at the supplements it is not clear if there were culture replicates. How can you be sure that the response is not due to jar effects? At minimum this needs to be discussed. This is OK, it has been clarified in the revised version	We are glad that Reviewer 1 confirms that this has been clarified satisfactorily in the first round of review.	NA
2-15	8. Figure 2: It seems odd that the survival between the 27 and 32 was only really different for the corals infected with C1. How can this be the case? Seems like all treatments resulted in significant mortality in recruits and this seems like a major caveat to the study (This is the same comment as point 1). The grey colored larvae and recruit makes it seem like these plots are just corals from one region, consider changing color. Yes, this is a concern indeed. Since the control in the experiment did not behave as control – I did raise this point in my 2nd assessment. I would like to see how the results will be if the control did not have significant mortality and sample	Please see Comments and Responses 2-5 and 2-6.	Please see Comments and Responses 2-5 and 2-6 for all additions.

	size were not so low in some cases (I have raised this issue as well). Authors need to convince readers that they repeated this experiment multiple times and saw similar mortality in control. But then, it can also be issue with their experiment system. I don't know how they can compare the mortality they see in control to that in the natural environment (I have raised this point as well).		
2-16	10. Fig S2: I think something very strange is going on with the ITS2 data. Have you tried seeing if your ASVs that are mapping to the I and F are actually host contamination? We have seen this before when mapping to the GeoSymBio database, which I assume is what you're doing- although it is never explicitly stated (just referred to an old github). I recommend taking these data through SymPortal to see if you still see this I and F association. We have seen similarly strange data and when we reanalyzed in SymPortal these strange lineages disappear and they were actually associated with host	We are glad that Reviewer 1 confirms that the additional work we did in Reviewer 1 has been clarified satisfactorily for this comment. We agree with Reviewer 1 that coral juveniles “are known to associate with multiple symbionts at this stage, it is not strange to see such not common symbiont genera.” Reviewer 2 brings up a good point. This had been added.	Supplementary Lines 150-155: “The detection of these additional taxa could also be due to symbiont cells being present in the coelenteron of the juvenile corals and may not necessarily be in symbiosis. Further work using methods such as in situ fluorescence hybridization (FISH) would be needed to confirm if these cells are within the coral cells. Therefore, given the detection of these taxa in coral juveniles of this species previously from other sources and using other methods, we suggest that these ASVs do not represent spurious variants or host contamination.”

	contamination. An alternative is to include host sequences in the GeoSymBio database when assigning the ASVs. 11. Fig S2: I also find it strange that an ASV is shared between C1 and D1a. What does this mean? Contamination? According to me, since this analysis is in the juveniles who are known to associate with multiple symbionts at this stage, it is not strange to see such not common symbiont genera. However, it is not necessary that they are in actual symbiosis with the juveniles and may be just present in the coelenteron. So the authors need to be careful how they present it and usage of the terms. So, I think the actual symbiosis is with Cladocopium and Durusdinimum, in this case and other are just (not a contaminant per se) associated by being present in the coelenteron		
4-1	Reviewer #4 (Remarks to the Author):	We thank Reviewer 4 for taking the time to look over our work again.	NA

	Apologies to you and to the authors for my delay in getting to this.		
4-2	In my review of the initial submission, I raised two main problems. One was lack of clarity with respect to sample sizes, and the other was the use of a cross-validation approach that assumes each observation of a larva or juvenile is statistically independent.	We thank Reviewer 4 for looking over the work we have done and for providing additional feedback to improve our paper.	NA
4-3	I appreciate the greater degree of sample size information, and this partly addresses my concern. However, the main piece of information that I wanted is not presented: how many colonies provided eggs and sperm came from each reef? The supplementary table does not show this information, because presumably eggs and sperm from a given colony could have been used in multiple crosses. I asked this question in order to get some idea of how much the authors might be quantifying effects of individual parents versus of the population of parents on a given reef. In response, the authors did an analysis of survival against NUMBER of crosses. I don't understand how this addresses the question.	We are glad that Reviewer 4 believes what we have done partly addresses their concern regarding sample size of colonies. They state: "However, the main piece of information that I wanted is not presented: how many colonies provided eggs and sperm came from each reef?" The information requested was "how many colonies" and was provided in Supplementary Table 1. This analysis was not against the number of crosses, but the number of moms or dads per cross. "Correlation between the number of maternal or	Supplementary Lines 324 – 326: "Supplementary Fig. 6. Correlation between the number of maternal or paternal colonies used in each cross compared to percent survival in larvae and juveniles. R² and p-values correspond to Pearson correlation coefficients."

		paternal colonies used in each cross compared to percent survival in larvae and juveniles.” This has been clarified in the Supplementary Figure legend.	
4-4	(1) The question is about whether the IDENTITY of the parents used in the crosses is important, and whether a particular parent that produces particularly vigorous or inviable offspring for example could be driving whether a “reef” appears to confer high or low survival. If the authors included parental ID as a random effect (see below) nested within reef, and they were still seeing significant reef effects, this would address my concern, but the text says in one place that parent was a random effect (lines 358-360) but then later that it was a fixed effect (434-436). (2) So I cannot figure out if the authors can address this issue with the results they already have or not. For these analyses it would be very helpful to have tables	(1) Reviewer 4 is now requesting information about the “IDENTITY of the parents.” This is an interesting question. However, in this work, we were interested in a reef level assessment and so our experimental design did not include crosses with different sets of parents per reef. Hence, parental identity would be confounded with reef identity if we ran the test as suggested: “If the authors included parental ID as a random effect (see below) nested within reef.” This was stated in the Methods but we have now further clarified that these were bulk crosses at the level of reef for this question. We acknowledge that a particular parent that produces particularly vigorous or inviable offspring for example could be driving	(1) Lines 357-361: “Briefly, purebred and hybrid offspring were produced by pooling eggs and sperm, respectively, from colonies collected from each reef and combining egg and sperm pools with sperm and egg pools from the same colonies (purebreds: parental colonies sourced from one reef) or from colonies sourced from different reefs (hybrids: parental colonies from two reefs) (Fig. 1b, Supplementary Table 1).” Lines 118- 120: “We acknowledge that a particular parent could produce particularly vigorous or inviable offspring, influencing whether a “reef” appears to confer high or low survival given the importance of parent identity on offspring survival ^{15,23}.” Lines 374-375: “To account for any potential influence of the reef of origin of egg and sperm donors used in each cross, survival models (see below), incorporated both maternal and paternal identity as fixed effects.” Replaced familial: Lines 138:

of statistical results reporting random effect and fixed effect estimates and their standard errors and so forth. Then there is no question about what terms are in the statistical analyses.	whether a “reef” appears to confer high or low survival. We and others have shown this in previous work but it was not the main goal here. Further, the text should read as “fixed” effects for both. This has now been corrected. Also, the use of “familial” or “family” might be confusing. That has been replaced with “cross.” (2) Multiple models were run using fixed and random effects to account for reef identity, including into probability of survival models and Wilcoxon and post-hoc tests of survival. For example, survival probability models were performed using Kaplan-Meier methods; with symbiont type, temperature, tank replicates, cross, maternal and paternal reef identities accounted for as fixed effects. As suggested, we have now added glmms to verify that the survival models accounted for variance due to	“Further, when averaged across all hybrid and purebred crosses...” Lines 142-145: “At the level of individual purebred and hybrid crosses, there were no significant differences in survival between 27 and 32°C at the juvenile stage when examined within each symbiont treatment (all juvenile cross comparisons Wilcoxon test $P_{adj} > 0.05$, Supplementary Fig. 4).” (2) As suggested, glmms including a “random effects of cross” was run “and tested whether a model with the random effect fits better than one without.” Lines 470-476: “ To verify these results and calculate the variance due to fixed and random effects, a generalized linear mixed effects model was run using a negative binomial distribution in lme4 67. The interactive effect of temperature and symbiont treatment were fixed effects and replicate tank, and cross were set as random effects; without an intercept. These results corroborated those from the Wilcox.test (Fig. 2b). Model selection was performed with AIC and the log-likelihood ratio tests using the “anova” function (Supplementary Table 5 and results).” Lines 145-150: “ However, cross identity significantly explained variability in juvenile survival shown through comparisons between models with and without cross as a random factor (AIC without cross: 101102.3 compared to AIC with cross: 8808.2; log-likelihood test $P = 2.2e-16$). Therefore, although differences between individual crosses were not significant (likely due to high stringency of tests) the overall cross identity was shown as a
--	--	--

		family. Namely, a negative binomial generalized linear model using posthoc Tukey tests were performed. These additional models confirmed our previous Wilcox tests and provided traditional coefficient tables. -As requested, results are reported as random effect and fixed effect estimates and their variances and standard deviations.	significant factor through log-likelihoods, suggestive of biological differences between crosses.” Supplementary Lines 29-32: “To verify previous Wilcoxon results and calculate the variance due to fixed and random effects, a generalized linear mixed effects model was run using a negative binomial distribution in lme4 using post-hoc Tukey tests (Supplementary Tables 5-6), and corroborated previous results.” -Supplementary Tables 5 and 6 were also added.
4-5	(1) On the second point, I think the authors have not come to terms with what they are assuming by doing their cross-validation. I am guessing that the cross-validation did indeed treat larvae and juveniles as all statistically independent of one another, so observations from particular parent colonies/crosses are appearing in both the testing and training partition in any given CV run. (2) If so, literally what the authors are doing is using a cross-validation that would optimize the choice of a model for predicting the responses of a new set	(1) Yes, we can confirm that CV is treating larvae and juveniles as all statistically independent of one another (this text and Supplementary Figure 9 was added in the previous round of reviews). (2) Your question about testing the CV method used is a very good point. To test the influence of our random CV method, we have tested a subset of the data against three of the other most common CV methods and compared the resulting explanatory power and	Additional text added to the Methods regarding new tests and additional Figure added to the Supplements. Supplementary Fig. 10 has been added. Lines 548 -556: “Furthermore, we also tested if our method of CV using random partitioning influenced the downstream predictions of the models, if the model’s ability to predict was influenced (as measured by R2) or influenced by overfitting. To do this, we tested our CV method (Repeated k-fold CV) against three other CV methods (Bootstrap, k-fold, and Leave One Out) using a subset of the data (purebred larval crosses) to inspect the resulting impact on predicted survival models (Supplementary Fig. 10). The percent variation in the data explained using each CV method was almost equivalent (model goodness of fit R2 = 0.837-0.835), as were the resulting predicated survival models (Supplementary Fig. 10). There was

of larvae and juveniles from the same set of colonies that they have, if they repeated their crosses with these same colonies. That is not what they are trying to do with the model in terms of prediction. What they want to do is extrapolate the results from their five reefs to what they would expect for crosses from other reefs that they don't have data from. A cross validation to optimize this should then exclude the data from a whole reef, or maybe exclude all the crosses involving mothers from one reef and fathers from the same reef or a different reef, then fit the BRT to the remaining data, and test the model against the excluded data. Then repeat excluding a different reef. Here the CV would be looking for a model that does a good job of predicting outcomes for crosses of unobserved parents from locations with a given combination of predictor variable values, which is how they want to use the model. By doing the random partitioning of the data as they have done it, the authors are substantially increasing the risk of overfitting for making predictions about reefs not in their data set.	the subsequent downstream model predictions against them. This including a "leave one out cross validation method" which is similar to the method you describe but is a more formalized framework for the exclusion and re-testing of missing data. We found no evidence that the CV method was influencing either of these outputs. These results and a new figure have been added to the Methods section in the main manuscript and Supplementary documents, respectively.	no evidence of overfitting from our choice of random repeated k-fold CV (Supplementary Fig. 9)."
---	--	---

4-6	(1) One other thing that I noticed in the revision is that there are a lot of p-values reported for pairwise comparisons between all the different pairs of crosses. If there are 25 different crosses, this would be 300 different pairwise comparisons. There is no mention of a multiple comparisons correction being applied in most cases (multiple comparisons correction is only mentioned for the ASV analysis). The authors may have a problem here because with so many comparisons, p-values will have to be extremely small (around 0.0001 with a standard Bonferroni correction) to be deemed significant (some multiple correction methods are more restrictive than others though). (2) An alternative way to deal with this would be to have random effects of parent ID or of cross, and test whether a model with the random effect fits better than one without. As noted above, I'm not sure if the authors already have such a model or not.	(1) All tests that include pairwise comparisons have now been re-run using Bonferroni multiple test corrections to report only significant p-adjusted values supplied in the Supplementary Figures and Results. Specifically, this includes Supplementary Figures 2, 3, 4, and 5. (2) As suggested above, we have now added glmms to verify that the survival models accounted for variance due to family. Namely, a negative binomial generalized linear model using posthoc Tukey tests were performed. We also report tests to whether a model with the random effect fits better than one without.	(1) Lines 463-466: “Larval and juvenile survival (amongst and between crosses) and between larvae and juveniles were assessed using the non-parametric, two-sided, p-adjusted for multiple pairwise tests with multiple grouping variables value using the “Bonferroni” method with the Wilcox.test from the package “ggpubr”⁶⁶.” Supplementary Figures 2, 3, 4, and 5 have been added. (2) Lines 470-476: “ To verify these results and calculate the variance due to fixed and random effects, a generalized linear mixed effects model was run using a negative binomial distribution in lme4⁶⁷. The interactive effect of temperature and symbiont treatment were fixed effects and replicate tank, and cross were set as random effects; without an intercept. These results corroborated those from the Wilcox.test (Fig. 2b). Model selection was performed with AIC and the log-likelihood ratio tests using the “anova” function (Supplementary Table 5 and results).” Lines 142-150: “At the level of individual purebred and hybrid crosses, there were no significant differences in survival between 27 and 32°C at the juvenile stage when examined within each symbiont treatment (all juvenile cross comparisons Wilcoxon test $P_{adj} > 0.05$, Supplementary Fig. 4). However, cross identity significantly explained variability in juvenile survival shown through comparisons between models with and without cross as a random factor (AIC without cross: 101102.3 compared to AIC with cross: 8808.2; log-likelihood test $P = 2.2e-16$). Therefore, although differences between
------------	---	---	--

			individual crosses were not significant (likely due to high stringency of tests) the overall cross identity was shown as a significant factor through log-likelihoods, suggestive of biological differences between crosses.”
	Visually, it looks almost certain that large differences between crosses are biologically real. If a random effects approach like I mentioned above is not feasible to implement, and multiple comparisons are so stringent nothing is significant, then the authors just need to take care to avoid referring to particular differences as being “significant”, and will need to make an argument on the fact that the p-values are all for example skewed towards small values and therefore indicate that there are real differences, or something along those lines.	This information has been added.	Lines 142-150: “At the level of individual purebred and hybrid crosses, there were no significant differences in survival between 27 and 32°C at the juvenile stage when examined within each symbiont treatment (all juvenile cross comparisons Wilcoxon test $P_{adj} > 0.05$, Supplementary Fig. 4). However, cross identity significantly explained variability in juvenile survival shown through comparisons between models with and without cross as a random factor (AIC without cross: 101102.3 compared to AIC with cross: 8808.2; log-likelihood test $P = 2.2e-16$). Therefore, although differences between individual crosses were not significant (likely due to high stringency of tests) the overall cross identity was shown as a significant factor through log-likelihoods, suggestive of biological differences between crosses. ”
4-7	Last, I don’t consider the BRT analysis to be essential to the paper, meaning I think it would be publishable without it.	We thank Reviewer 4 for their support that even without the BRT analysis, our paper provides relevant information that is publishable.	NA

Reviewers' Comments:

Reviewer #2:

Remarks to the Author:

I have read the revised version of ms by Quigley and vanOppen and this is the 3rd time. This version is more clear - including the text and figures (more confusing and unclear than the text in the previous versions)

I did enjoy going through all the 3 versions (including all the suggestions and comments from other reviewers) and evolution of the ms as a result of comments and suggestions from all the reviewers and efforts from authors.

I don't have any comments and In my opinion this will be a nice contribution to coral reef science and a initial step towards identifying potential tolerant colonies

Shashank Keshavmurthy

Reviewer #4:

Remarks to the Author:

In my first review, I raised two main points. One was that it was hard to work out important sample size information. I asked the question "How many parent colonies were collected from each reef, and how many of those (at each reef) provided sperm or eggs or both?" The authors did provide the total number collected, and number providing sperm or eggs or both *to each cross*. I clarified in my second review that, while useful, it still does not indicate how many provided sperm or eggs for the experiment overall. If this information is presented in the MS, I cannot find it, since the number of colonies from a given reef used in the different crosses seems to vary. It would be easy enough to do – simply say M colonies from reef X contributed eggs to the experiment, and P contributed sperm to the experiment, but not all colonies contributed sperm or eggs to all relevant crosses involving that reef. This could be reported in a two column table with a row for each reef. As I said at the outset, this is critical for knowing how strong the evidence is that the authors are measuring reef-level effects, and this interpretation of the results forms an important basis for the BRT analysis. The authors do note in their reply that "Our experimental design did not include crosses with different sets of parents per reef," and later that "these were bulk crosses at the level of reef," and this is why they did not include parental ID as a random effect. I assume this means that all the eggs from all parents from reef A were mixed with all sperm from all parents from reef B to do the AxB cross. If this is normal practice for coral biologists when doing crosses, then fine, but the total number of colonies providing eggs and sperm from each reef is still important information for readers to judge whether reef-level effects are really being measured. If some reefs only provided 2 colonies to the whole experiment, for example, this is different from if they provided 8 in terms of the confidence a reader can have that the results are representative of corals from that reef and unlikely to be potentially skewed by one outlier parent colony in the mix. I appreciate the addition of text acknowledging that individual parents could be having strong effects on the results, but reporting these numbers would help the reader evaluate just how big an issue this might be.

My second main point in my original review was that the cross-validation was not appropriate for extrapolating from corals sampled from 5 reefs to the entire GBR. In my second review I clarified why a conventional cross-validation that treats all larvae as independent does not do this, but instead optimizes the choice of a model for predicting the responses of a new set of larvae and juveniles from the same set of colonies that they have. There is a huge risk of overfitting in using this sort of cross-validation to extrapolate out to new reefs. I suggested that the authors instead leave out whole reefs one-by-one and use the removed reef as the testing data set. My hunch is that this would lead to a larger prediction error and a simpler model being picked for extrapolation, but the only way to know is to do the analysis. Instead, the authors did three different variations on cross-validation, all of which as far as I can tell make the same problematic assumption of the original cross-validation, in which case they don't address this concern. If the authors want to make predictions about reefs not included in the data based on a model chosen by CV, they should

do the CV in a way that optimizes the model for predicting what will happen on unobserved reefs.

That said, the revision does a good job of addressing my concern about the large number of pairwise comparisons. Since there is strong support overall for variability among crosses, this supports the idea that at least some of the crosses differ from each other in their effects, even though after Bonferroni corrections it is not possible to say definitively which crosses were significantly different from which.

	Nature Communications manuscript NCOMMS-20-48044B	
	REVIEWER COMMENTS -RESPONSES	
	Predictive models for the selection of thermally tolerant corals based on offspring survival	
	Authors K. M. Quigley, M. J. H. van Oppen	
	Comment	Response/Edit
Reviewer #2 - 1	I have read the revised version of ms by Quigley and vanOppen and this is the 3rd time. This version is more clear - including the text and figures (more confusing and unclear than the text in the previous versions)	We also agree that this new version is clearer. We are very appreciative of the time you have taken to help improve this work.
#2 - 2	I did enjoy going through all the 3 versions (including all the suggestions and comments from other reviewers) and evolution of the ms as a result of comments and suggestions from all the reviewers and efforts from authors.	We also believe that the manuscript has been much improved during this review process. We sincerely thank Dr. Keshavmurthy for sticking with us during these 3 revisions!
#2 - 3	I don't have any comments and In my opinion this will be a nice contribution to coral reef science and a initial step towards identifying potential tolerant colonies Shashank Keshavmurthy	We thank Dr. Keshavmurthy for his positive assessment of our work and for his belief that this would be a positive contribution to the coral reef science literature.
Reviewer #4 - 1	In my first review, I raised two main points. One was that it was hard to work out important sample size information. I asked the question "How many parent colonies were collected from each reef, and how many of those (at each reef) provided sperm or eggs or both?" The authors did provide the total number collected, and number providing sperm or eggs or both *to each cross*. I clarified in my second review that, while useful, it still	My apologies, I thought that a "cross level" analysis was requested, not the reef level. This information has now been provided as requested as a Supplementary Table and in the text to include the number of unique colonies used per reef. The exact sentence and Table suggested has been added to the Methods and Supplementary sections, respectively,

does not indicate how many provided sperm or eggs for the experiment overall. If this information is presented in the MS, I cannot find it, since the number of colonies from a given reef used in the different crosses seems to vary. It would be easy enough to do – simply say M colonies from reef X contributed eggs to the experiment, and P contributed sperm to the experiment, but not all colonies contributed sperm or eggs to all relevant crosses involving that reef. This could be reported in a two column table with a row for each reef. As I said at the outset, this is critical for knowing how strong the evidence is that the authors are measuring reef-level effects, and this interpretation of the results forms an important basis for the BRT analysis. The authors do note in their reply that “Our experimental design did not include crosses with different sets of parents per reef,” and later that “these were bulk crosses at the level of reef,” and this is why they did not include parental ID as a random effect. I assume this means that all the eggs from all parents from reef A were mixed with all sperm from all parents from reef B to do the AxB cross. If this is normal practice for coral biologists when doing crosses, then fine, but the total number of colonies providing eggs and sperm from each reef is still important information for readers to judge whether reef-level effects are really being measured. If some reefs only provided 2 colonies to the whole experiment, for example, this is different from if they provided 8 in terms of the confidence a reader can have that the	as requested. Lines 369 – 377: “Seven, 6, 8, 7, 7 colonies from BK, DR, CU, LS, SB contributed eggs to the experiment, respectively, and 11, 12, 16, 15, and 14 colonies from those reefs contributed sperm to the experiment, but not all colonies contributed sperm or eggs to all relevant crosses involving that reef (Supplementary Table 2). We are glad that Reviewer 4 agrees with the additional text provided in Round 2 of the reviews. This information has improved the clarity and rigour of this manuscript.
---	--

	results are representative of corals from that reef and unlikely to be potentially skewed by one outlier parent colony in the mix. I appreciate the addition of text acknowledging that individual parents could be having strong effects on the results, but reporting these numbers would help the reader evaluate just how big an issue this might be.	
#4 - 2	My second main point in my original review was that the cross-validation was not appropriate for extrapolating from corals sampled from 5 reefs to the entire GBR. In my second review I clarified why a conventional cross-validation that treats all larvae as independent does not do this, but instead optimizes the choice of a model for predicting the responses of a new set of larvae and juveniles from the same set of colonies that they have. There is a huge risk of overfitting in using this sort of cross-validation to extrapolate out to new reefs. I suggested that the authors instead leave out whole reefs one-by-one and use the removed reef as the testing data set. My hunch is that this would lead to a larger prediction error and a simpler model being picked for extrapolation, but the only way to know is to do the analysis. Instead, the authors did three different variations on cross-validation, all of which as far as I can tell make the same problematic assumption of the original cross-validation, in which case they don't address this concern. If the authors want to make predictions about reefs not included in the data based on a model chosen by CV, they should do the CV in a way that optimizes the model for predicting what will happen on unobserved reefs. That said, the revision does a good job of addressing my concern about the large number of pairwise comparisons. Since there is	This is an important point to acknowledge. We have added this point and have undertaken the additional analysis suggested and included an additional Supplementary Figure, which includes using the larval purebred data as an example to “leave out whole reefs one-by-one and use the removed reef as the testing data set.” Lines 559 – 564: “Finally, it is important to note that the ability to extrapolate to all GBR reefs is likely limited by the relatively small number of reefs (n = 5) used in this study. Although all four methods of cross-validation did not show any sign of overfitting (including the drop-one method implemented across all reefs), and the models do well in predicting survival when all reefs are included (Supplementary Fig. 11), the ability of the models to predict when whole reefs are removed may be limited.”

	strong support overall for variability among crosses, this supports the idea that at least some of the crosses differ from each other in their effects, even though after Bonferroni corrections it is not possible to say definitively which crosses were significantly different from which.	
--	---	--

Reviewers' Comments:

Reviewer #4:

Remarks to the Author:

The authors have responded to my remaining two points by reporting the sample size information I requested (its omission from earlier revisions appears to have been based on a misunderstanding). The numbers are higher than I feared, which addresses my concern about how representative the sperm and eggs are of the coral population on the relevant reef.

The authors also reported a small piece of a "leave-one-whole-reef-out" analysis, and they report it in Supplementary Figure 11. At first, it did not seem like a big problem, until I realized that the scales on the y axis are completely different. Measured survival (panel a) looks to be around 75% on average, which matches the predicted survival when larvae from Curd reef are included in the analysis (panel b). However, when the larvae from Curd reef are excluded, predicted survival seems dramatically lower at around 12% (panel c). In the main text results, this result is not reported, and even in the methods, the text written by the authors does not explicitly acknowledge that, when the reef is omitted from the analysis, the model predicts a survival that is wildly different (a 6-fold difference in survival) from what is predicted when data from that reef is included in the training set. This seems to support my fear that the BRT approach, by using cross validation that takes random subsets of larvae taken from across all the study reefs, is leading to substantial overfitting and very large prediction error when it comes to extrapolating to reefs not included in the training data. Unless I really misunderstand something, this undermines the analysis reported under "Models of intrinsic resistance" in the main text, since the authors use the survival data from the 5 reefs they studied to extrapolate to survival on all the reefs on the GBR. If the one leave-one-reef-out analysis they did gets survival wrong so dramatically, then there are very strong grounds to suspect similar problems when predictions are made to other reefs.

I strongly recommend that the authors wrestle seriously and honestly with this issue. I can see two ways forward. One is to drop the BRT analysis entirely, and publish what is otherwise a very solid paper representing a lot of hard and important work.

Another option is to complete the leave-whole-reefs-out analysis, and report the differences between the BRT analysis when cross validation removes observations at random, versus when whole reefs are left out (this would require presenting results of the two types of cross-validation similarly so they can be compared), and to discuss the differences between the analysis and what they imply about the ability to infer temperature tolerance on reefs where data have not been collected yet.

Title: Predictive models for the selection of thermally tolerant corals based on offspring survival

Manuscript number: NCOMMS-20-48044C

Authors: K. M. Quigley, M. J. H. van Oppen

REVIEWER COMMENTS

Reviewer #4 (Remarks to the Author):

Comment 4- 1:

The authors have responded to my remaining two points by reporting the sample size information I requested (its omission from earlier revisions appears to have been based on a misunderstanding). The numbers are higher than I feared, which addresses my concern about how representative the sperm and eggs are of the coral population on the relevant reef.

Response 4-1:

We are pleased that our revisions have adequately addressed these concerns.

Comment 4- 2:

The authors also reported a small piece of a "leave-one-whole-reef-out" analysis, and they report it in Supplementary Figure 11. At first, it did not seem like a big problem, until I realized that the scales on the y axis are completely different. Measured survival (panel a) looks to be around 75% on average, which matches the predicted survival when larvae from Curd reef are included in the analysis (panel b). However, when the larvae from Curd reef are excluded, predicted survival seems dramatically lower at around 12% (panel c). In the main text results, this result is not reported, and even in the methods, the text written by the authors does not explicitly acknowledge that, when the reef is omitted from the analysis, the model predicts a survival that is wildly different (a 6-fold difference in survival) from what is predicted when data from that reef is included in the training set. This seems to support my fear that the BRT approach, by using cross validation that takes random subsets of larvae taken from across all the study reefs, is leading to substantial overfitting and very large prediction error when it comes to extrapolating to reefs not included in the training data. Unless I really misunderstand something, this undermines the analysis reported under "Models of intrinsic resistance" in the main text, since the authors use the

survival data from the 5 reefs they studied to extrapolate to survival on all the reefs on the GBR. If the one leave-one-reef-out analysis they did gets survival wrong so dramatically, then there are very strong grounds to suspect similar problems when predictions are made to other reefs.

Response 4- 2:

-We have now completed and present the full "leave-one-whole-reef-out" analysis as requested.

-The full "leave-one-whole-reef-out" is now presented in the main results section and we have included a new figure (Figure 6) to present these results so that it is transparent, and in the main body of the work.

-We have moved all results relating to cross validation results and methods from the Methods into the main Results, including Results previously presented for Supplementary Figures 10 and 11 and new Results of the full "leave-one-whole-reef-out" analysis.

-We have added further information to the main text as Results and moved information from the methods that explicitly acknowledges that, when the reef is omitted from the analysis, the model predicts a survival are different.

-We have moved information from the Methods and Supplementary into the main text, showing that substantial overfitting is not occurring (previously Supplementary Figure 10).

-The Results added are presented below:

Lines 300- 340:

"...For example, holobiont tolerance is influenced by the algal and bacterial symbionts, and the acclimatory history and potential of the host and symbionts (reviewed in ⁵⁵). Although physiological experiments are useful for investigating if tolerance can be transferred between generations, further work is needed to confirm if this is maintained in natural environments. Therefore, given the multitude of different factors influencing heat tolerance, it is possible that this design may not fully capture the adaptive potential of a single reef that can therefore be extrapolated to the entirety of the GBR.

To assess the influence of individual corals or specific reefs on the BRT models and ultimately, intrinsic resistance equations, we compared our Cross Validation method (Repeated k-fold CV), which takes random subsets of larvae taken from across all the study reefs, against three other CV methods (Bootstrap, k-fold, and Leave One Out), and as well as sequential reef exclusion (Fig. 6). There was no evidence of overfitting from our choice of random repeated k-fold CV, and the percent variation in the data explained using each CV method was almost equivalent (model goodness of fit $R^2 = 0.837-0.835$), as were the resulting predicated intrinsic resistance models (Fig. 6 a, b). When reefs were excluded sequentially from each dataset (larval purebreds, larval hybrids, juvenile D1 purebreds, juvenile D1 hybrids) and compared to each of the final models with all reefs included (Fig. 6

c -f), the inclusion or exclusion of any single reef did not drive the resulting models. Instead, the models were influenced by each reef and were dependent on the life-stage and symbiotic state. Specifically, purebred larval responses were mainly influenced by Davies and Backnumbers, the larval hybrids by Long Sandy, juvenile D1 purebreds by Curd, whereas juvenile D1 hybrids were not influenced by any one particular reef compared to the others. This comparison between our BRT analysis, which CV removes observations at random (Fig. 6 “All reefs”), versus when whole reefs are left out (Fig. 6, all other reefs) highlights the importance of including as many reefs as possible in the BRT models but suggests that not one reef influenced the overall model construction more than others. It is important to note that the ability to extrapolate to all GBR reefs is likely limited by the relatively small number of reefs ($n = 5$) used in this study. Although all four methods of cross-validation did not show any sign of overfitting (including the drop-one method implemented across all reefs), and the models do well in predicting survival when all reefs are included (Fig. 6 a, b, Supplementary Fig. 10 a, b), the ability of the models to predict when whole reefs are removed may be limited given when a whole reef is omitted from the analysis, the model predictions of survival are different (Supplementary Fig. 10 c). This would be expected given the exclusion of whole factors from the dataset and again suggests the importance of including all reefs in the analysis. Although the predictive power of our models does not seem to be biased towards any particular reef, we acknowledge that it is hard to capture the full diversity of biological and biophysical parameters of all the >3000 reefs with the only five reefs measured here. This is important because we recognize that any differences between these analyses due to the removal of one reef would imply that the ability to infer temperature tolerance on reefs where data have not been collected yet (i.e. naïve reefs) could vary depending on the data input into the model. The flexibility of the BRT framework however will easily incorporate new coral species, new reefs, and new experimental data, which can be used in the future to optimize these methods.”

Comment 4- 3:

I strongly recommend that the authors wrestle seriously and honestly with this issue. I can see two ways forward. One is to drop the BRT analysis entirely, and publish what is otherwise a very solid paper representing a lot of hard and important work.

Another option is to complete the leave-whole-reefs-out analysis, and report the differences between the BRT analysis when cross validation removes observations at random, versus when whole reefs are left out (this would require presenting results of the two types of cross-validation similarly so they can be compared), and to discuss the differences between the analysis and what they imply about the ability to infer temperature tolerance on reefs where data have not been collected yet.

Response 4- 3:

We have elected to do option 2 (“complete the leave-whole-reefs-out analysis”).

-We have completed the full “leave-whole-reefs-out” analysis as requested. This is now presented in the main Results section and it has been added as Figure 6 so that it is transparent, and in the main body of the work.

-We have completed the following as requested: "report the differences between the BRT analysis when cross validation removes observations at random, versus when whole reefs are left out (this would require presenting results of the two types of cross-validation similarly so they can be compared.)"

Reviewers' Comments:

Reviewer #4:

Remarks to the Author:

This version is definitely improved, with some important material moved into the main text, although the phrasing could be clearer at some points (see below), and the typical reader is still likely to miss the significance of the analysis omitting whole reefs, especially given that (1) there is no mention of it in the abstract, and (2) the two figures showing the effect of omitting whole reefs from the analysis (Fig 6 and Supplementary Figure 10) use different y-axis limits across panels that make it hard for readers to see the differences (as I did on first reading of the previous version).

For the abstract, given the results of the analyses omitting whole reefs, at a minimum I recommend toning down the claims about how many reefs harbor heat-resistant corals – for example “Our findings predict that hundreds of reefs (~7.5%) MAY BE home to corals that have high and heritable...” (lines 22-23).

Lines 330-331: “...when a whole reef is omitted from the analysis, the model predictions of survival are different (Supplementary Fig 10c)”. I suggest adding “, in some cases substantially”, or at least saying “quite different in some cases” because a several-fold prediction error for reefs withheld from analysis is actually a huge miss by the model, strongly indicating that the model is overfitted for predicting what is happening on unobserved reefs. It also seems like citing Figure 6 along with Supplementary Fig 10 would be appropriate here

Line 332: “suggests the importance of including all reefs in the analysis.” I disagree. The fact that the predictions differ by such a large amount rather suggests the importance of collecting data from more reefs to generate more stable model predictions about reefs where we do not yet have data. I suspect the authors rephrase along these lines.

Lines 336-338: “...differences between these analyses due to the removal of one reef would imply that the ability to infer temperature tolerance on reefs where data have not been collected yet (i.e., naïve reefs) could vary depending on the data input into the model.” (lines 336-338). This phrasing is not very clear, since it is stating a truism -- obviously if the data are different, the results will be different. The point is that the leave-one-reef-out analysis suggests that the data in hand do not allow as much confidence when predicting what is happening on reefs for which there are not yet any data. Instead they imply that the ability to infer temperature tolerance on reefs where data have not been collected yet may depart from predictions substantially more than expected based on model confidence intervals. The important point here is that the model's ability to predict the situation on reefs not included in the fitting is potentially poor, and this does not come across clearly in the text.

Figure 6 and Supplementary Figure 10 – Figure 6 is a really good start to illustrating what happens when reefs are excluded from the analysis. However, the panels in fig 6 (c), (d), (e), and (f), and likewise the panels in Supp Fig 10 really should have the same y-axis limits to make it easier to visually compare the survival patterns across panels –a reader casually looking at this figure might not register that the y-axis values are completely different. For example, the NoLongSandy and NoBacknumbers panels in (e) are essentially predicting very constant survival (50-51% in the former and ~35.2% in the latter) versus a ~4-fold decline in survival from (80% to 20%) in the NoCurd case. These differences would be much clearer if the y-axis limits were consistent across all panels. I also recommend that Supp Fig 10 be expanded to show corresponding sets of panels for the results when other reefs are included vs excluded (also using consistent y-axis limits so that the predicted values for a reef when data from that reef is included versus excluded can be easily seen by looking at the graph).

Reviewer #4 (Remarks to the Author):	Comment	Response
NA	This version is definitely improved, with some important material moved into the main text, although the phrasing could be clearer at some points (see below), and the typical reader is still likely to miss the significance of the analysis omitting whole reefs, especially given that (1) there is no mention of it in the abstract, and (2) the two figures showing the effect of omitting whole reefs from the analysis (Fig 6 and Supplementary Figure 10) use different y-axis limits across panels that make it hard for readers to see the differences (as I did on first reading of the previous version).	All suggestions made below have been incorporated. Please see below.
1	For the abstract, given the results of the analyses omitting whole reefs, at a minimum I recommend toning down the claims about how many reefs harbor heat-resistant corals – for example “Our findings predict that hundreds of reefs (~7.5%) MAY BE home to corals that have high and heritable...” (lines 22-23).	This has been edited as suggested. “Our findings predict hundreds of reefs (~7.5%) may be home to corals that have high and heritable heat-tolerance in habitats with high daily and annual temperature ranges and historically variable heat stress.”
2	Lines 330-331: “...when a whole reef is omitted from the analysis, the model predictions of survival are different (Supplementary Fig 10c)”. I suggest adding “, in some cases substantially”, or at least saying “quite different in some cases” because a several-fold prediction error for reefs withheld from analysis is actually a huge miss by the model, strongly indicating that the model is overfitted for predicting what is happening on unobserved reefs. It also seems like citing Figure 6 along with Supplementary Fig 10 would be appropriate here	Both of these suggestions have been added. “when whole reefs are removed may be limited given when a whole reef is omitted from the analysis, the model predictions of survival are quite different in some cases (Fig. 6, Supplementary Fig. 10 c).”
3	Line 332: “suggests the importance of including all reefs in the analysis.” I disagree. The fact that the predictions differ by such a large amount rather suggests the importance of collecting data from more reefs to generate more stable model predictions about reefs where we do not yet have data. I suspect the authors rephrase along these lines.	This has been modified as suggested: “This would be expected given the exclusion of whole factors from the dataset and suggests the importance of collecting data from more reefs to generate precise predictions about reefs where we do not yet have data.”
4	Lines 336-338: “...differences between these analyses due to the removal of one reef would imply that the ability to infer temperature tolerance on reefs where data have not been collected yet (i.e., naïve reefs) could vary depending on the data input into the model.” (lines 336-338). This phrasing is not very clear, since it is stating a truism - - obviously if the data are different, the results will be different.	We apologize if this was still unclear. We have added in the additional text outlined in “the important point here” that was suggested to further clarify this point. “Hence, the model’s ability to predict the situation on reefs not included in the fitting is potentially limited.”

	The point is that the leave-one-reef-out analysis suggests that the data in hand do not allow as much confidence when predicting what is happening on reefs for which there are not yet any data. Instead they imply that the ability to infer temperature tolerance on reefs where data have not been collected yet may depart from predictions substantially more than expected based on model confidence intervals. The important point here is that the model's ability to predict the situation on reefs not included in the fitting is potentially poor, and this does not come across clearly in the text.	
5	Figure 6 and Supplementary Figure 10 – Figure 6 is a really good start to illustrating what happens when reefs are excluded from the analysis. However, the panels in fig 6 (c), (d), (e), and (f), and likewise the panels in Supp Fig 10 really should have the same y-axis limits to make it easier to visually compare the survival patterns across panels –a reader casually looking at this figure might not register that the y-axis values are completely different. For example, the NoLongSandy and NoBacknumbers panels in (e) are essentially predicting very constant survival (50-51% in the former and ~35.2% in the latter) versus a ~4-fold decline in survival from (80% to 20%) in the NoCurd case. These differences would be much clearer if the y-axis limits were consistent across all panels.	This is a good point. I did explore making all panels on the same y-axis initially, but found it was harder to compare the fit of the predicted survival with the actual survival, which was the main goal of this exercise. This comes at the expense of comparisons across reefs. However, given comparisons of predicted vs. actual values was the main point of comparison here, I would elect to have the y-axis vary for clarity of this specific point. This is a valid point though, to highlight this and clarify this point, a further sentence in the Figure Legend has been added in the Main Document. “Please note the differences in y-axis values between groups of panels. Y-axis values within each analysis group was consistent but were allowed to vary across groups to facilitate comparisons between predicted versus actual survival estimates.”
6	I also recommend that Supp Fig 10 be expanded to show corresponding sets of panels for the results when other reefs are included vs excluded (also using consistent y-axis limits so that the predicted values for a reef when data from that reef is included versus excluded can be easily seen by looking at the graph.	This information can now be found more completely in Figure 6. Supp Fig. 10 was made in response to the first set of comments but then followed up with the more comprehensive analysis on the 4th review. The outcome of that is Fig. 6.